# An Efficient Local Search Approach for Polarized Community Discovery in Signed Networks

**Linus Aronsson**
Department of Computer Science and Engineering
Chalmers University of Technology &
University of Gothenburg
Gothenburg, Sweden
linaro@chalmers.se

**Morteza Haghir Chehreghani**
Department of Computer Science and Engineering
Chalmers University of Technology &
University of Gothenburg
Gothenburg, Sweden
morteza.chehreghani@chalmers.se

## Abstract

Signed networks, where edges are labeled as positive or negative to represent friendly or antagonistic interactions, provide a natural framework for analyzing polarization, trust, and conflict in social systems. Detecting meaningful group structures in such networks is crucial for understanding online discourse, political divisions, and trust dynamics. A key challenge is to identify communities that are internally cohesive and externally antagonistic, while allowing for neutral or unaligned vertices. In this paper, we propose a method for identifying $k$ polarized communities that addresses a major limitation of prior methods: their tendency to produce highly size-imbalanced solutions. We introduce a novel optimization objective that avoids such imbalance. In addition, it is well known that approximation algorithms based on *local search* are highly effective for clustering signed networks when neutral vertices are not allowed. We build on this idea and design the first local search algorithm that extends to the setting with neutral vertices while scaling to large networks. By connecting our approach to block-coordinate Frank-Wolfe optimization, we prove a linear convergence rate, enabled by the structure of our objective. Experiments on real-world and synthetic datasets demonstrate that our method consistently outperforms state-of-the-art baselines in solution quality, while remaining competitive in computational efficiency.

## 1 Introduction

Signed networks extend traditional graph representations by associating each edge with a positive or negative number, indicating friendly or antagonistic relationships. Originating from studies on social dynamics in the 1950s [22], signed networks introduce fundamental differences in graph structure that make many algorithms designed for unsigned networks inapplicable [42, 6, 44]. These challenges have fueled extensive research in recent years, leading to advances in signed network embeddings, signed clustering, and signed link prediction. We refer to the survey by [42] for a comprehensive review of these methods. Most relevant to this paper is the problem of signed clustering, which we split into two categories: (i) *signed network partitioning* (SNP), and (ii) *polarized community discovery* (PCD). The latter is the problem studied in this paper.

The goal of signed clustering is to identify $k$ clusters where intra-cluster similarity is maximized (predominantly positive) and inter-cluster similarity is minimized (predominantly negative). This problem has numerous real-world applications [42], particularly in social networks, where vertices represent individuals and edges capture friendly or antagonistic relationships (e.g., shared or opposing political views). Detecting conflicting groups in such networks is crucial for analyzing polarization [1, 47, 45], echo chambers [19, 17], and the spread of misinformation [41, 13, 46].

39th Conference on Neural Information Processing Systems (NeurIPS 2025).

In the SNP problem, the $k$ groups must form a partition of the vertices, meaning every vertex must be included. Spectral methods based on the signed Laplacian have been widely used to tackle this problem [27, 11, 34, 14]. Alternatively, formulating SNP explicitly as an optimization problem leads to the well-studied *correlation clustering* (CC) problem [5], which is known to be APX-hard. Consequently, numerous approximation algorithms have been developed [5, 8, 15, 2], with *local search* methods standing out for their strong performance in both clustering quality and computational efficiency [43, 10, 3, 4].

The problem formulation of PCD is identical to that of SNP, *except that the k clusters are not required to form a partition of the vertices, allowing some vertices to remain unassigned.* The goal is therefore to only find the *dense* subgraphs of polarized communities. This accounts for cases where certain vertices are neutral w.r.t. the underlying conflicting group structure. For example, in a social network with a heated political debate, many users may not engage in the dispute, and their interactions might not align with any specific faction. There is a substantial body of work addressing this problem, but most approaches focus on identifying only two communities [6, 45, 36, 16, 35, 21]. As a result, they do not easily generalize to arbitrary $k$. To our knowledge, only two works specifically tackle PCD for arbitrary $k$. [12] formulated the task as a constrained quadratic optimization problem and proposes an efficient algorithm that iteratively refines small subgraphs, avoiding the costly computation of the full adjacency matrix. [44] introduced a spectral method based on maximizing a discrete Rayleigh quotient, which extends the seminal work of [6] to accommodate arbitrary $k$. These methods are known to produce highly imbalanced communities in terms of size [21].

The main contributions of this paper are as follows:

(i) We propose a novel formulation for the PCD problem that encourages more balanced communities, addressing a key limitation of previous work that typically optimize *polarity* [44] (see Eq. 2). As demonstrated in our experiments, optimizing polarity often leads to clustering solutions with multiple empty clusters. The importance of promoting balanced communities (to avoid trivial solutions where all objects are placed in one or few clusters) is well established in the graph clustering literature [11]. In Appendix E, we expand on this, and provide some examples of practical scenarios where (reasonably) balanced communities are favorable in our context. We note that [21] also proposes an objective for PCD called $\gamma$-polarity aimed at addressing cluster imbalance; however, it is restricted to the case of $k = 2$ clusters. In contrast, our objective supports an arbitrary number of clusters $k$. Nonetheless, we compare to their method experimentally for $k = 2$, and explain how it differs conceptually in Appendix F.

(ii) Motivated by the effectiveness of local search-based approximation algorithms for CC (and many other machine learning models), we propose the first scalable local search algorithm for PCD, which explicitly allows for neutral objects.

(iii) We establish a linear convergence rate of our local search algorithm by connecting it to block-coordinate Frank-Wolfe optimization [18, 29]. This connection is made possible utilizing the specific structure of our proposed optimization objective and extending the analysis in [43].

(iv) We propose techniques that allow the local search method to scale to large networks.

(v) Finally, through extensive experiments on commonly used real-world and synthetic datasets in previous work on PCD, we show that our approach consistently outperforms state-of-the-art baselines in terms of (a) recovering ground-truth solutions and (b) finding high quality solutions of reasonable cluster size balance.

## 2 Problem Formulation

We start by introducing the relevant notation, followed by an introduction to CC, which is connected to our problem. Finally, we describe PCD, including our novel formulation of the problem.

**Notation.** Consider a signed network $G = (V, E)$, where $V$ is the set of objects and $E$ the set of edges. The weight of an edge $(i, j) \in E$ is represented by the element $A_{i,j} \in \{-1, 0, +1\}$ of an adjacency matrix $A$. The matrix $A$ is symmetric with zeros on the diagonal, which means $A_{i,j} = A_{j,i}$ and $A_{i,i} = 0$. We use $A_{i,:}$ and $A_{:,j}$ to denote row $i$ and column $j$ of $A$, respectively. While we restrict all similarities to be in $\{-1, 0, +1\}$ (for clarity), all methods presented in the paper extend to arbitrary similarities in $\mathbb{R}$. We can decompose the adjacency matrix as $A = A^+ - A^-$ where $A^+ = \max(A, 0)$ and $A^- = \max(-A, 0)$. A clustering with $k$ clusters is denoted

$S_{[k]} = \{S_1, \ldots, S_k\}$, where each $S_m \subseteq V$ is the set of objects assigned to cluster $m \in [k] = \{1, \ldots, k\}$. Let $N_{\text{intra}}^+ = \sum_{m \in [k]} \sum_{i,j \in S_m} A_{i,j}^+$ and $N_{\text{intra}}^- = \sum_{m \in [k]} \sum_{i,j \in S_m} A_{i,j}^-$ be the sum of positive and absolute negative intra-cluster similarities, respectively. Furthermore, let $N_{\text{inter}}^+ = \sum_{m \in [k]} \sum_{p \in [k] \setminus \{m\}} \sum_{i \in S_m} \sum_{j \in S_p} A_{i,j}^+$ and $N_{\text{inter}}^- = \sum_{m \in [k]} \sum_{p \in [k] \setminus \{m\}} \sum_{i \in S_m} \sum_{j \in S_p} A_{i,j}^-$ be the sum of positive and absolute negative inter-cluster similarities, respectively.

## 2.1 Correlation Clustering

We begin by noting that for CC, unlike PCD to be discussed in the next subsection, a clustering $S_{[k]}$ is a partition of $V$, meaning $V = \bigcup_{m \in [k]} S_m$ and each $S_m$ is disjoint. A notable feature of CC is its ability to automatically determine the number of clusters [7], but here we we focus on the $k$-constrained variant of CC [20] as it is most relevant to our problem. The $k$-CC problem can be defined as shown below.

**Problem 1** ($k$-CC). *Find a clustering $S_{[k]}$ that maximizes*

$$N_{\text{intra}}^+ - N_{\text{intra}}^- + N_{\text{inter}}^- - N_{\text{inter}}^+. \tag{1}$$

In other words, the goal is to find a clustering that (i) maximizes intra-cluster similarities and (ii) minimizes inter-cluster similarities. In the CC literature, it is known that maximizing certain subsets of terms in Eq. 1, such as the total number of agreements $N_{\text{intra}}^+ + N_{\text{inter}}^-$, is equivalent to maximizing the full objective [9]. However, this equivalence does not hold for PCD when neutral objects are allowed, as each of the four terms in Eq. 1 contributes uniquely to the decision of whether an object should be clustered or left neutral. We formally show this in Appendix A, and thus focus on the full objective in the next section when we introduce PCD. CC is known to be NP-hard [5], and many approximation algorithms have been proposed [5, 8, 15, 2], with *local search* methods standing out for their strong performance in both clustering quality and computational efficiency [43, 10, 3].

## 2.2 Polarized Community Discovery

For PCD, we introduce a *neutral set* $S_0$. As a result, each object in $V$ is either assigned to one of the non-neutral clusters $S_1, \ldots, S_k$ or designated as neutral by placing it in $S_0$. Consequently, a clustering $S_{[k]}$ is no longer a partition of $V$ and we have $S_0 = V \setminus \bigcup_{m \in [k]} S_m$ (although all clusters are still disjoint). Given this, the goal of PCD is to identify non-neutral clusters $S_{[k]}$ such that (i) a large value of the objective in Eq. 1 is obtained (consistent with CC), and (ii) the graph induced by the non-neutral objects is as *dense* as possible (i.e., most edge weights are $+1$ or $-1$). Any object that hinders either of these goals should be assigned to the neutral set $S_0$. This includes ambiguous objects, such as those with significant similarity to multiple clusters or those with inherently weak associations (e.g., low-degree nodes). Importantly, there exists a natural trade-off between the size and density of the non-neutral clusters: small clusters can trivially achieve high density. As we will demonstrate, our objective allows for a flexible balance of this trade-off.

In prior work, it is common to encourage the presence of neutral objects by penalizing large/sparse non-neutral clusters. This is typically done by normalizing Eq. 1 by the number of non-neutral objects, i.e.,

$$\frac{(N_{\text{intra}}^+ - N_{\text{intra}}^-) + \alpha(N_{\text{inter}}^- - N_{\text{inter}}^+)}{\sum_{m \in [k]} |S_m|}, \tag{2}$$

where $\alpha \in \mathbb{R}$ is used to balance (i) maximization of intra-cluster similarities and (ii) minimization of inter-cluster similarities. If $\alpha = 1/(k-1)$, Eq. 2 is commonly referred to as *polarity* in prior work and is a well-established objective for PCD [6, 44]. This choice of $\alpha$ was proposed in [44], based on the observation that the number of intra-similarities scale linearly with $k$, while the number of inter-similarities grow quadratically. This choice prevents inter-similarities from dominating the objective. We use this value of $\alpha$ throughout the paper unless otherwise stated.

However, as highlighted in [21] (and in our experiments), maximizing polarity often results in highly imbalanced clustering solutions (often with multiple empty clusters). In particular, *clustering solutions with the same polarity can differ significantly in terms of cluster size balance*. A concrete example illustrating this issue is provided in Appendix D. [21] proposes a new objective called $\gamma$-polarity that addresses this issue for the special case of $k = 2$. Our proposed objective is different

from $\gamma$-polarity and is applicable with any arbitrary $k$. In Appendix F, we compare $\gamma$-polarity with our proposed objective, defined in Eq. 3. We also compare to [21] in our experiments.

In this paper, we propose an alternative objective that, instead of normalizing by the number of non-neutral objects, incorporates a regularization term by subtracting the sum of squared sizes of the non-neutral clusters.

$$(N^+_{\text{intra}} - N^-_{\text{intra}}) + \alpha(N^-_{\text{inter}} - N^+_{\text{inter}}) - \beta \sum_{m \in [k]} |S_m|^2. \tag{3}$$

The third term in Eq. 3 has been previously applied to the minimum cut objective for unsigned networks [10]. In our context (i.e., for the PCD problem), the objective in Eq. 3 achieves two goals simultaneously: it penalizes the formation of (i) large/sparse and (ii) highly imbalanced non-neutral clusters. The second property is easy to see, as for a clustering with $k$ clusters and $n$ objects, the term $\sum_{m \in [k]} |S_m|^2$ is minimized when each cluster is assigned $\frac{n}{k}$ objects (i.e., the clusters are perfectly balanced). Notably, the squaring of cluster sizes is what encourages cluster size balance.

We introduce regularization as an additive term rather than a normalization for two key reasons: (i) It allows a flexible trade-off between the number of non-neutral objects and the density of the graph induced by them (controlled by the parameter $\beta \in \mathbb{R}$), which is a desirable property in this context as discussed in the beginning of this section. This possibility is absent in the existing methods that are based on Eq. 2 (polarity). (ii) It enables the development of an efficient optimization procedure based on *local search* with strong convergence guarantees. In the context of CC, local search–based approximation algorithms are known to significantly outperform other methods [43, 10].

In this paper, we develop the first scalable local search algorithm specifically tailored to the PCD setting. In Section 4, we demonstrate across a range of real-world and synthetic datasets that the advantages of local search optimization (e.g., for CC) carry over to PCD as well. We are now ready to formally state our problem, and we subsequently highlight its computational complexity in Thm. 1.

**Problem 2** ($k$-PCD). *Find a clustering $S_{[k]}$ with neutral objects $S_0 = V \setminus \bigcup_{m \in [k]} S_m$ that maximizes Eq. 3.*

**Theorem 1.** *Problem 2 (i.e., $k$-PCD) is NP-hard.*

All proofs can be found in Appendix B.

## 3 Algorithms

Thm. 1 underscores the necessity of approximate methods to solve Problem 2. In this section, we demonstrate how it can be solved using Frank-Wolfe (FW) optimization [18]. Specifically, we consider a variant called block-coordinate FW, which we begin by describing in the next subsection. After this, we establish its equivalence to a straightforward and provably efficient local search procedure. Next, we analyze the convergence rate of this approach. Following that, we propose practical enhancements to improve scalability, enabling the method to handle large problems. A detailed discussion of the impact of $\alpha$ and $\beta$ is deferred to Appendix C.

### 3.1 Block-Coordinate Frank-Wolfe Optimization

The Frank–Wolfe (FW) algorithm is one of the earliest methods for nonlinear constrained optimization [18]. In recent years, it has regained popularity, particularly in machine learning, due to its scalability [24]. In this paper, we use a variant of this method called block-coordinate FW [29]. This method yields a significantly faster optimization procedure while enjoying similar theoretical guarantees. Block-coordinate FW is applied to problems where the feasible domain can be split into blocks $\mathcal{D} = \mathcal{D}^{(1)} \times \cdots \times \mathcal{D}^{(n)} \subseteq \mathbb{R}^d$, where each $\mathcal{D}^{(i)} \subseteq \mathbb{R}^{d_i}$ is convex and compact and we have $d = \sum_{i=1}^n d_i$. Let $\boldsymbol{x}_{[n]}$ denote the concatenation of the variables $\boldsymbol{x}_i \in \mathcal{D}^{(i)}$ from all blocks $i \in [n]$. The optimization problem is then

$$\max_{\boldsymbol{x}_{[n]} \in \mathcal{D}^{(1)} \times \cdots \times \mathcal{D}^{(n)}} f(\boldsymbol{x}_{[n]}), \tag{4}$$

where $f$ is a differentiable function with an $L$-Lipschitz continuous gradient. This approach is particularly effective when optimizing $f$ w.r.t. the variables in a single block (while keeping other

**Algorithm 1** Block-coordinate Frank-Wolfe

1: Initialize $\boldsymbol{x}_{[n]}^{(0)} \in \mathcal{D}^{(1)} \times \cdots \times \mathcal{D}^{(n)}$.
2: **for** $t := 0, \ldots, T$ **do**
3:      Select a random block $i \in [n]$
4:      $\boldsymbol{x}_i^* := \arg\max_{\boldsymbol{x}_i \in \mathcal{D}^{(i)}} \boldsymbol{x}_i \cdot \nabla_i f(\boldsymbol{x}_{[n]}^{(t)})$
5:      Let $\gamma := \frac{2n}{t+2n}$ or optimize by line-search
6:      $\boldsymbol{x}_i^{(t+1)} := (1-\gamma)\boldsymbol{x}_i^{(t)} + \gamma \boldsymbol{x}_i^*$
7: **end for**

**Algorithm 2** Local Search for PCD

1: Randomly assign each object $i \in [n]$ to a cluster in $S_{[k]}$, or to the neutral set $S_0$
2: **while** not converged **do**
3:      Select object $i \in [n]$ randomly
4:      Assign object $i$ to a cluster in $S_{[k]}$, or to the neutral set $S_0$, which maximally increases our objective in Eq. 3
5: **end while**

blocks fixed) is simple and efficient. This turns out to be the case for our problem, as will be discussed in the remainder of this section. The method is outlined in Alg. 1, where $\nabla_i f(\boldsymbol{x}_{[n]})$ represents the gradient of $f(\boldsymbol{x}_{[n]})$ with respect to block $\boldsymbol{x}_i$. When the problem involves only a single block ($n = 1$), Alg. 1 reduces to the standard FW algorithm. We now show how our problem can be turned into an instance of Eq. 4. Below, we show an alternative way of writing our objective in Eq. 3.

**Proposition 1.** *Our objective in Eq. 3 can be written as*

$$\sum_{m \in [k]} \sum_{i,j \in S_m} (A_{i,j} - \beta) - \alpha \sum_{m \in [k]} \sum_{p \in [k] \setminus \{m\}} \sum_{i \in S_m} \sum_{j \in S_p} A_{i,j}. \tag{5}$$

We observe that the regularization term in Eq. 3 is equivalent to shifting the intra-cluster similarities by $-\beta$. This reformulation proves highly useful for the remainder of this section. In our context, each object $i \in [n]$ defines a block. We represent the cluster membership of object $i$ using $\boldsymbol{x}_i \in \{\boldsymbol{e}_0, \ldots, \boldsymbol{e}_k\}$, where $\boldsymbol{e}_m$ (for $m \in \{0, \ldots, k\}$) are the standard basis vectors. Each $\boldsymbol{x}_i$ is a vector of dimension $k + 1$, with index zero indicating membership in the neutral set $S_0$. Specifically, if $x_{i0} = 1$, object $i$ is assigned to $S_0$. Using this notation, we can now define our objective as follows.

$$f(\boldsymbol{x}_{[n]}) = \sum_{(i,j) \in E} \sum_{m \in [k]} x_{im} x_{jm} (A_{i,j} - \beta) - \alpha \sum_{(i,j) \in E} \sum_{m \in [k]} \sum_{p \in [k] \setminus \{m\}} x_{im} x_{jp} A_{i,j}. \tag{6}$$

Note that we do not include any terms involving $x_{i0}$, thereby excluding contributions from neutral objects, as intended. The objective in Eq. 6 remains discrete and is therefore unsuitable for FW optimization. To address this, we relax the problem to make it continuous by allowing soft cluster memberships. Specifically, each $\boldsymbol{x}_i \in \Delta^{k+1}$, where $\Delta^{k+1} = \{\boldsymbol{x} \in \mathbb{R}^{k+1} \mid x_m \geq 0, \sum_{m=0}^k x_m = 1\}$ represents the simplex of dimension $k$. With this relaxation, we can now reformulate the optimization problem as follows.

$$\max_{\boldsymbol{x}_i \in \Delta^{k+1}, \forall i \in [n]} f(\boldsymbol{x}_{[n]}) \tag{7}$$

Eq. 7 is a specific instance of the block-coordinate FW formulation described in Eq. 4 (where $f$ is non-concave). Consequently, we can apply Alg. 1 to solve this problem.

### 3.2 Equivalence to a Local Search Approach

We now show that optimizing Eq. 7 using Alg. 1 is equivalent to the local search procedure in Alg. 2. Let matrix $G \in \mathbb{R}^{n \times (k+1)}$, where element $G_{i,m} := [\nabla_i f(\boldsymbol{x}_{[n]}^{(t)})]_m$ is the gradient of $f(\boldsymbol{x}_{[n]})$ w.r.t. variable $m$ of block $i$ evaluated at $\boldsymbol{x}_{[n]}^{(t)}$. Given this, we present the following theorem.

**Theorem 2.** *If $\boldsymbol{x}_{[n]}^{(0)}$ in Alg 1 is discrete, the following hold. (a) For our problem (Eq. 7), the solution $\boldsymbol{x}_i^*$ (line 4 of Alg. 1) is the basis vector $\boldsymbol{e}_p$, where $p = \arg\max_{m \in \{0, \ldots, k\}} G_{i,m}$ and the optimal value of the step size on line 6 is $\gamma = 1$. (b) Our objective function in Eq. 6 satisfies $(\boldsymbol{x}_i^* - \boldsymbol{x}_i^{(t)}) \cdot G_{i,:} = f(\boldsymbol{x}_{[n]}^*) - f(\boldsymbol{x}_{[n]}^{(t)})$, where $\boldsymbol{x}_{[n]}^*$ is $\boldsymbol{x}_{[n]}^{(t)}$ with block $i$ modified to $\boldsymbol{x}_i^*$.*

From part (a) of Thm. 2, the current solution, $\boldsymbol{x}_{[n]}^{(t)}$, remains discrete (i.e., hard cluster assignments) at every step of Alg. 1 for all $i \in [n]$. Moreover, each step of Alg. 1 consists of placing object $i$ in

the cluster $m \in \{0, \ldots, k\}$ with maximal gradient $G_{i,m}$. By part (b) of Thm. 2, this is equivalent to placing object $i$ in the cluster that maximally improves our objective in Eq. 3. Based on this, we conclude the following corollary.

**Corollary 1.** *From Thm. 2, if $\boldsymbol{x}_{[n]}^{(0)}$ is discrete, solving the optimization problem in Eq. 7 using Alg. 1 is equivalent to executing the local search procedure described in Alg. 2.*

### 3.3 Convergence Analysis

Given Corollary 1, we now present results for the convergence rate of Alg. 2. Following the prior work on the analysis of general FW algorithms [24, 29], we begin by providing the following definitions.

**Definition 1** (FW duality gap). The FW duality gap is defined as [24]

$$g(\boldsymbol{x}_{[n]}) = \max_{\boldsymbol{s}_{[n]} \in \mathcal{D}} (\boldsymbol{s}_{[n]} - \boldsymbol{x}_{[n]}) \cdot \nabla f(\boldsymbol{x}_{[n]}), \tag{8}$$

which is zero if and only if $\boldsymbol{x}_{[n]}$ is a stationary point. Furthermore, let $\tilde{g}_t = \min_{0 \leq l \leq t-1} g(\boldsymbol{x}_{[n]}^{(l)})$ be the smallest duality gap observed in Alg. 1 up until step $t$.

**Definition 2** (Convergence rate). We say the convergence rate of Alg. 1 is at least $O(1/r_t)$ if $\mathbb{E}[\tilde{g}_t] \leq O(1/r_t)$, where $r_t$ is some expression involving only $t$ and the expectation is w.r.t. the random selection of blocks on line 3. If $n = 1$ the bound is deterministic.

The FW algorithm has been shown to converge to a stationary point of $f$ under various settings, with well-established convergence rates. We summarize a few known results below. The standard FW algorithm ($n = 1$) achieves a deterministic convergence rate of $O(1/t)$ for concave $f$ [18] and $O(1/\sqrt{t})$ for non-concave $f$ [28, 39]. For the block variant, [29] proves a convergence rate of $O(1/t)$ for concave $f$ in expectation. For non-concave $f$, [43] proves a convergence rate of $O(1/t)$ in expectation, under the assumption that $f(\boldsymbol{x}_{[n]})$ is multilinear in each block $\boldsymbol{x}_i$ including correlation clustering. We here extend the analysis of [43] to Problem 2 ($k$-PCD) using Alg. 2, described in Thm. 3. Note that their analysis cannot be applied directly to our objective function in Eq. 6 as this objective does not satisfy the multilinearity property.

**Theorem 3.** *The convergence rate of Alg. 2 is at least $n h_0 / t = O(1/t)$, where $h_0 = \sum_{(i,j) \in E} |A_{i,j}|$.*

The $O(1/t)$ convergence rate presented in Thm. 3 should be compared with the deterministic convergence rate of $O(1/\sqrt{t})$ for general non-concave functions $f$ under the standard FW method ($n = 1$) [28, 39].

### 3.4 Improving the Computational Complexity

In the previous section, we demonstrated that Alg. 2 is guaranteed to converge at the linear rate $O(1/t)$, making it highly efficient. In this section, we propose an alternative version of Alg. 2, designed to enhance the efficiency of each step $t$ while maintaining full equivalence in functionality. This ensures that the convergence analysis from the previous section still remains valid. Firstly, a naive implementation of Alg. 2 has a complexity of $O(Tk^2n^2)$, as each iteration requires $O(k^2n^2)$ to compute the full objective in Eq. 6 for every candidate cluster in order to determine the best cluster for the current object $i$. Since the number of iterations $T$ until convergence is typically larger than $n$, this approach can become computationally expensive. Part (b) of Thm. 2 offers an alternative: *instead of evaluating the full objective, we can compute the gradient $G_{i,:}$, which involves only terms related to object $i$.* Based on this, we present the following theorem.

**Theorem 4.** *Let $S_{[k]}$ be the current clustering of our local search procedure, with neutral objects $S_0 = V \setminus \bigcup_{m \in [k]} S_m$. The gradient can then be expressed as follows.*

$$G_{i,m} = 2(1 + \alpha) \sum_{j \in S_m} A_{i,j} - 2\alpha \sum_{p \in [k]} \sum_{j \in S_p} A_{i,j} - 2\beta |S_m| + 2\beta \mathbf{1}[i \in S_m] - \beta \tag{9}$$

*for all $m \in [k]$ and $G_{i,0} = 0$.*

A naive calculation of the full gradient $G_{i,:}$ for block $i$ is $O(k^2n)$. However, the specific structure of the gradient in Eq. 9 reduces the complexity to $O(kn)$, since the term $\sum_{p \in [k]} \sum_{j \in S_p} A_{i,j}$ is

---

**Algorithm 3** Local Search for PCD (efficient)

---

1: Randomly assign each object $i \in [n]$ to one of the clusters in $S_{[k]}$, or to the neutral set $S_0$.
2: Initialize $X \in \{0,1\}^{n \times k}$, with $X_{i,m} = 1$ if object $i$ belongs to cluster $m \in [k]$, and $X_{i,m} = 0$ otherwise. Neutral objects $i \in S_0$ have rows $X_{i,:}$ of zeros.
3: $M := 2AX$
4: **while** not converged **do**
5:      Select object $i \in [n]$ uniformly at random
6:      $\hat{p} :=$ current cluster of $i$
7:      $M_i := \sum_p M_{i,p}$
8:      $G_{i,p} := (1+\alpha)M_{i,p} - \alpha M_i - 2\beta|S_p| + 2\beta\mathbf{1}[i \in S_p] - \beta, \ \forall p \in [k]$ {See Eq. 9}
9:      $G_{i,0} := 0$
10:      $p^* := \arg\max_{p \in \{0,...,k\}} G_{i,p}$
11:      **if** $p^* = \hat{p}$ **then** skip to next iteration
12:      Assign object $i$ to cluster $S_{p^*}$
13:      **if** $\hat{p} \in [k]$ **then** $M_{:,\hat{p}} := M_{:,\hat{p}} - 2A_{:,i}$
14:      **if** $p^* \neq 0$ **then** $M_{:,p^*} := M_{:,p^*} + 2A_{:,i}$
15: **end while**

---

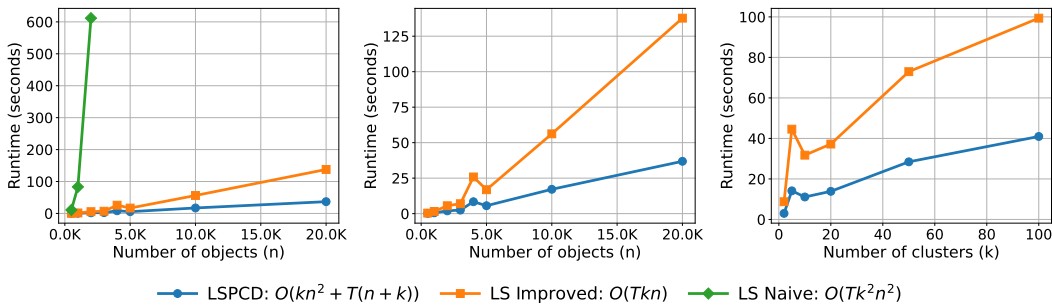

Figure 1: Comparison of runtime for the three implementations of Alg. 2 (local search) introduced in Section 3.4 by varying the graph size $n$ and the number of non-neutral clusters $k$, using data generated from the m-SSBM model. See Section 4 for a description of this dataset. The noise level is fixed at $\eta = 0.4$. When varying $n$, we fix $k = 4$; when varying $k$, we fix $n = 5000$. LSPCD corresponds to Alg. 3 and is used in all subsequent experiments because of its superior computational efficiency.

independent of the cluster $m$ and can therefore be precomputed (see Alg. 3). See the proof of Thm. 4 for further insight on this. From Thm. 2, the gradient $G_{i,m}$ represents the impact on the full objective in Eq. 6 if object $i$ is placed in cluster $m$. Thus, because $G_{i,0} = 0$, we observe that an object $i$ is made neutral if its contribution to all non-neutral clusters is currently negative. Moreover, the total complexity is now reduced to $O(Tkn)$, which is a significant improvement over the naive approach with complexity of $O(Tk^2n^2)$.

We present a third approach, shown in Alg. 3. We define a matrix $X \in \{0,1\}^{n \times k}$, with $X_{i,m} = 1$ if object $i$ belongs to cluster $m \in [k]$, and zero otherwise. Neutral objects $i \in S_0$ have rows $X_{i,:}$ of zeros. The procedure precomputes the matrix $M = 2AX$, where $M_{i,m}$ is the total similarity of object $i$ to cluster $m$. Precomputing $M$ is $O(kn^2)$, but allows gradient computation in $O(k)$ (line 8). We then have to update $M$ accordingly (lines 13 and 14), which is $O(n)$, reducing the per-iteration complexity to $O(n+k)$. The total complexity is $O(kn^2 + T(n+k))$, which improves on the $O(Tnk)$ approach because, (i) computing $M$ involves a sparse matrix product, which is highly efficient in practice, and (ii) since $T > n$, reducing per-iteration cost leads to significant practical gains.

On the largest datasets in our experiments, Alg. 3 completes in seconds or minutes, while the naive version would take hours or days. Figure 1 presents a runtime comparison of the three approaches discussed above. The method in Alg. 3 (LSPCD) achieves the best computational efficiency, significantly outperforming the naive approach in Alg. 2. Consequently, Alg. 3 is used in all subsequent experiments. In Appendix G.5, we further demonstrate the scalability of Alg. 3 to large-scale graphs.

# 4 Experiments

In this section, we present our experimental evaluation. Additional results are provided in Appendix G. We use eight publicly available real-world datasets commonly adopted in prior work on PCD [44], with dataset details included in Appendix G.1. Notably, no real-world datasets with ground-truth solutions for PCD (i.e., where neutral objects are allowed) currently exist. Consequently, prior work has relied on *polarity* (Eq. 2) as a proxy for evaluating solution quality [6, 44]. For consistency, we also report polarity scores for these datasets. Additionally, following previous work [44], we include experiments on synthetic datasets where ground-truth solutions are available. Throughout this section, we fix $\alpha$ and $\beta$ (as specified below). In Appendix C, we provide a detailed discussion on the impact of these parameters (and we investigate it experimentally in Appendix G.8). We compare our local search algorithm for PCD, named **LSPCD** (see Alg. 3), to several baseline methods, which we introduce below. The complete source code for all experiments is publicly available[1].

**Baselines**. (i) SCG [44] is a spectral method that identifies $k$ non-neutral clusters by maximizing polarity (Eq. 2) with $\alpha = 1/(k-1)$. It solves a continuous relaxation and applies one of four rounding techniques, resulting in SCG-MA, SCG-R, SCG-MO, and SCG-B. We refer to [44] for details. (ii) KOCG [12] optimizes a similar objective and formulates it as a constrained quadratic optimization problem (this optimization approach is very different from ours). It outputs a set of local minima. For comparison, we select KOCG-top-1 (the best local minimum) and KOCG-top-$r$, where $r$ is chosen such that the number of non-neutral objects is closest to SCG-MA, following [44]. (iii) BNC [11] and SPONGE [14] are spectral methods designed for SNP that do not explicitly handle neutral objects. As in [44], we apply two heuristics with these methods: (a) we treat all $k$ clusters as non-neutral, and (b) we run the methods with $k+1$ clusters and then designate the largest cluster as neutral. These variants are denoted BNC-$k$ / SPONGE-$k$ and BNC-$(k+1)$ / SPONGE-$(k+1)$, respectively. (iv) N2PC [21] introduces a framework that employs a graph neural network (GNN) to predict cluster memberships in the PCD setting. They propose $\gamma$-*polarity*, a generalization of polarity designed to encourage balanced clusters. Higher values of $\gamma$ impose stricter balance constraints, with $\gamma = 1$ recovering the standard polarity definition (Eq. 2). Since their method supports only $k = 2$ clusters, results are reported exclusively for this setting. See details about baselines in Appendix G.2.

**Metrics**. (i) Following prior work, we use *polarity* to evaluate the quality of different methods [6, 44], defined as in Eq. 2 with $\alpha = 1/(k-1)$. (ii) For datasets with available ground-truth, we measure the recovery-rate of ground-truth clusters using the F1-score, which is the precision and recall averaged over all clusters (as in [6, 44]). (iii) To evaluate the balance of a clustering solution $S_{[k]}$, we use the *imbalance factor* from [38]. Let $p_i = |S_i| / \sum_{m \in [k]} |S_m|$ be the proportion of objects in cluster $S_i$. The imbalance factor (IF) is defined as

$$\texttt{IF}(p_1, \ldots, p_k) = \frac{1}{1 - \xi} \log_2\left(\sum_{i=1}^{k} p_i^\xi\right) / \log_2(k) \in [0, 1], \tag{10}$$

where 1 indicates perfect balance and 0 indicates maximal imbalance (i.e., all objects in one cluster). For $\xi = 1$, the numerator reduces to Shannon entropy; we use $\xi = 3$ to penalize highly imbalanced solutions more strongly. The conclusions of our results are robust to changes in $\xi$ around our chosen value. Results for other values of $\xi$ are provided in Appendix G.6. In addition, Appendix G.9 presents a detailed summary of the solutions found by each method, including the number of non-neutral objects, the number of non-empty clusters, runtime, and more.

**Synthetic datasets**. In our first experiment, we evaluate how well the methods recover ground-truth clusters using synthetic networks. Following [6, 44], we employ the *modified signed stochastic block model* (m-SSBM), which was specifically designed to generate synthetic graphs with planted ground-truth communities for PCD. The m-SSBM model is parameterized by four variables: (i) $n$, the total number of nodes; (ii) $k$, the number of non-neutral clusters; (iii) $\ell$, the size of each non-neutral cluster; and (iv) $\eta \in [0, 1]$, which controls the edge probabilities. Smaller values of $\eta$ correspond to denser non-neutral clusters and lower levels of noise (see Appendix G.3 for detailed description). Here, we assume balanced ground-truth clusters of size $\ell$. In Appendix G.4, we show that our method remains robust when increasing cluster imbalance on synthetic data.

In Figure 2, we present the F1-score and polarity of various methods on different synthetic graphs generated using the m-SSBM model, across different noise levels $\eta$. For clarity, we include the

---

[1]https://github.com/Linusaronsson/NeurIPS2025-LSPCD

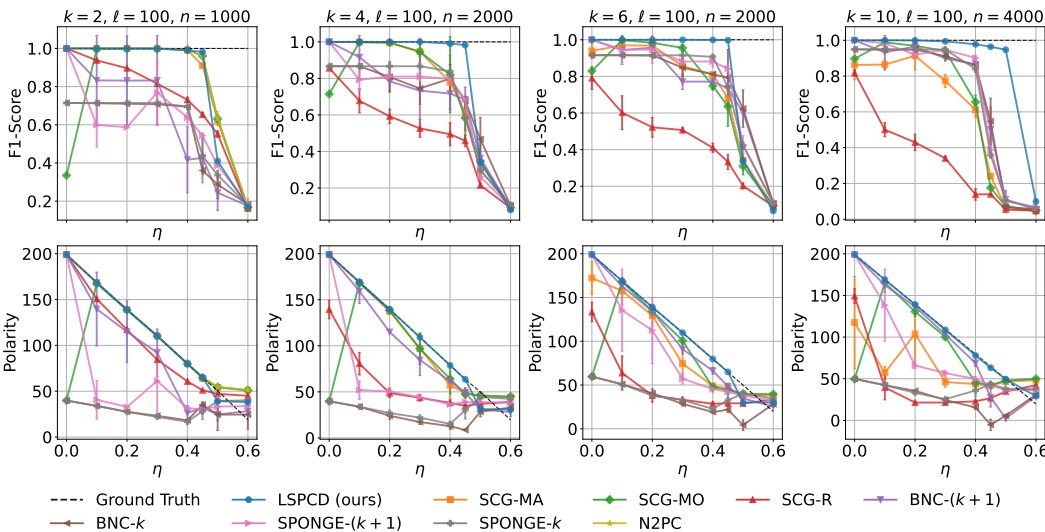

Figure 2: F1-score and polarity of different methods on synthetic graphs generated using the m-SSBM model, as the noise level $\eta$ varies. See main text below for details. See Appendix G.4 for more results.

best performing baselines (KOCG peforms poorly here). Each setting is repeated 10 times, and we report the average. We fix $\beta = 0.4$ for LSPCD (it is robust to the choice of $\beta$) and $\gamma = 1$ for N2PC. We see that the recovery rate of all methods decreases as $\eta$ increases, since the sparsity and noise level of the graph increases. For $k = 4, 6, 10$, we observe that our method significantly outperforms baseline methods, being the only method capable of recovering the ground-truth solutions for $\eta > 0.2$. For $k = 2$, we observe that our method, SCG-MA, and N2PC perform the best. However, for $k = 2$ (which N2PC is limited to), the problem is significantly more simple. Finally, we see that the ground-truth solution correlates with large polarity, justifying the use of polarity to measure solution quality for the real-world data.

**Real-world datasets**. Table 1 present results for different methods and datasets with $k = 2, 4, 6$ (we use the same values of $k$ as SCG [44]). $|E|$ denotes the number of edges with non-zero edge weight. Only seven of the eight datasets are shown (due to space limit); see Appendix G.9 for complete results. The spectral clustering methods, BNC and SPONGE, exceeded memory limits on large datasets (caused by $k$-means), indicated by dashes. We report the mean over five runs, with standard deviations included in Appendix G.9. For our method, we select the $\beta$ value that maximizes *polarity*, testing 10 values per dataset, while we fix $\alpha = 1/(k-1)$ for all methods. For each method and dataset, we report the *polarity* (POL) and the *imbalance factor* (IF).

The results show that our method is highly competitive, often the best, in polarity across all datasets. In particular, our method consistently finds solutions with large polarity, while maintaining a good cluster size balance (large imbalance factor). Additionally, our method does not impose strict balance constraints, which is beneficial since real-world clusterings are rarely perfectly balanced. Instead, it identifies high-polarity solutions with reasonable balance, making it more practical for real-world applications. Notably, in cases where baseline methods attain higher polarity, it is usually at the cost of a very low imbalance factor (which often implies one or more empty clusters). Moreover, baseline methods with an imbalance factor near 1 generally exhibit very low polarity. This observation highlights the inherent trade-off between polarity and cluster balance, which our approach balances very well.

Results for N2PC are only included for $k = 2$, as it does not support $k > 2$. We observe that increasing $\gamma$ results in more balanced clusters, which consistently leads to lower polarity. When N2PC optimizes standard polarity ($\gamma = 1$), the polarity is usually highest, but the imbalance factor is consistently very low (often near zero). This suggests that optimizing polarity alone (as SCG does) is not ideal; encouraging the algorithm to produce more balanced solutions (even at the cost of reduced polarity) generally yields solutions that align better with user expectations in practice (see Appendix E for a detailed discussion on this). This is a well-known observation in previous work on clustering of signed and unsigned graphs (beyond PCD, without neutral objects) [11]. While N2PC

Table 1: Polarity (POL) and imbalance factor (IF) for different methods and real-world datasets. $|E|$ denotes the number of edges with non-zero edge weight.

| | | BTC | | WikiV | | REF | | SD | | WikiC | | EP | | WikiP | |
|---|---|---|---|---|---|---|---|---|---|---|---|---|---|---|---|
| | $|V|$ | 6K | | 7K | | 11K | | 82K | | 116K | | 131K | | 138K | |
| | $|E|$ | 214K | | 1M | | 251K | | 500K | | 2M | | 711K | | 715K | |
| $k$ | | POL | IF | POL | IF | POL | IF | POL | IF | POL | IF | POL | IF | POL | IF |
| 2 | LSPCD (OURS) | 29.0 | 0.65 | 62.3 | 0.43 | 146.1 | 0.71 | 75.9 | 0.25 | 190.8 | 0.83 | 127.8 | 0.73 | 82.0 | 0.30 |
| | SCG-MA | 28.8 | 0.16 | 71.5 | 0.01 | 172.2 | 0.01 | 77.5 | 0.01 | 155.2 | 0.53 | 128.3 | 0.04 | 82.8 | 0.01 |
| | SCG-MO | 29.5 | 0.03 | 71.7 | 0.01 | 174.1 | 0.01 | 79.7 | 0.01 | 175.7 | 0.43 | 128.7 | 0.04 | 88.4 | 0.01 |
| | SCG-B | 21.6 | 0.99 | 37.6 | 0.04 | 116.3 | 0.03 | 61.0 | 0.05 | 129.3 | 0.64 | 156.4 | 0.04 | 46.5 | 0.04 |
| | SCG-R | 14.2 | 0.25 | 54.7 | 0.17 | 120.9 | 0.04 | 29.7 | 0.08 | 101.1 | 0.57 | 72.3 | 0.19 | 36.1 | 0.17 |
| | KOCG-TOP-1 | 1.0 | 1.00 | 7.6 | 0.72 | 11.6 | 0.64 | 2.0 | 0.79 | 5.9 | 0.84 | 8.2 | 0.60 | 3.0 | 0.79 |
| | KOCG-TOP-$r$ | 3.8 | 0.99 | 2.3 | 1.00 | 15.4 | 0.96 | 2.6 | 0.98 | 3.4 | 0.99 | 14.0 | 0.94 | 1.3 | 0.99 |
| | BNC-$(k+1)$ | -10.8 | 0.13 | -1.1 | 0.79 | -1.0 | 1.00 | — | | — | | — | | — | |
| | BNC-$k$ | 5.3 | 0.02 | 15.8 | 0.00 | 41.5 | 0.00 | — | | — | | — | | — | |
| | SPONGE-$(k+1)$ | 1.0 | 0.79 | 1.0 | 0.47 | 1.0 | 0.79 | — | | — | | — | | — | |
| | SPONGE-$k$ | 5.1 | 0.00 | 15.8 | 0.00 | 41.5 | 0.00 | — | | — | | — | | — | |
| | N2PC ($\gamma = 1$) | 29.6 | 0.02 | 71.6 | 0.00 | 173.6 | 0.01 | 81.2 | 0.00 | 172.8 | 0.46 | 169.7 | 0.00 | 87.5 | 0.00 |
| | N2PC ($\gamma = 1.2$) | 30.1 | 0.46 | 71.7 | 0.01 | 173.6 | 0.02 | 81.1 | 0.00 | 175.7 | 0.77 | 169.8 | 0.00 | 87.1 | 0.00 |
| | N2PC ($\gamma = 1.5$) | 24.4 | 1.00 | 70.0 | 0.10 | 130.3 | 0.94 | 81.8 | 0.00 | 158.2 | 0.99 | 169.9 | 0.00 | 86.6 | 0.02 |
| | N2PC ($\gamma = 1.7$) | 23.9 | 1.00 | 59.1 | 0.56 | 119.4 | 1.00 | 55.0 | 1.00 | 155.5 | 0.99 | 124.3 | 0.29 | 75.2 | 0.39 |
| | N2PC ($\gamma = 2.0$) | 24.1 | 1.00 | 40.5 | 1.00 | 118.1 | 1.00 | 52.1 | 1.00 | 142.0 | 1.00 | 76.7 | 0.99 | 48.3 | 0.96 |
| 4 | LSPCD (OURS) | 23.3 | 0.47 | 52.6 | 0.52 | 139.2 | 0.41 | 61.1 | 0.54 | 113.6 | 0.56 | 111.5 | 0.58 | 71.6 | 0.27 |
| | SCG-MA | 25.1 | 0.22 | 52.9 | 0.36 | 94.5 | 0.68 | 35.5 | 0.25 | 104.9 | 0.06 | 127.4 | 0.30 | 56.5 | 0.52 |
| | SCG-MO | 25.3 | 0.22 | 53.1 | 0.37 | 82.1 | 0.70 | 38.5 | 0.20 | 117.9 | 0.24 | 129.0 | 0.34 | 39.7 | 0.30 |
| | SCG-B | 12.4 | 0.23 | 24.8 | 0.60 | 116.2 | 0.00 | 48.3 | 0.38 | 49.8 | 0.86 | 94.4 | 0.54 | 45.7 | 0.21 |
| | SCG-R | 8.0 | 0.52 | 19.5 | 0.44 | 118.7 | 0.02 | 10.7 | 0.76 | 41.1 | 0.66 | 65.1 | 0.20 | 33.7 | 0.14 |
| | KOCG-TOP-1 | 8.4 | 0.90 | 4.5 | 0.81 | 15.0 | 0.65 | 2.6 | 0.80 | 4.5 | 0.23 | 8.9 | 0.91 | 3.1 | 0.71 |
| | KOCG-TOP-$r$ | 5.0 | 0.93 | 3.3 | 0.99 | 3.7 | 0.87 | 3.0 | 0.79 | 3.8 | 0.99 | 11.0 | 0.96 | 4.4 | 0.84 |
| | BNC-$(k+1)$ | -9.4 | 0.23 | -1.1 | 0.65 | -1.0 | 1.00 | — | | — | | — | | — | |
| | BNC-$k$ | 5.2 | 0.01 | 15.8 | 0.00 | 41.5 | 0.00 | — | | — | | — | | — | |
| | SPONGE-$(k+1)$ | 1.1 | 0.10 | 1.0 | 0.71 | 1.0 | 0.79 | — | | — | | — | | — | |
| | SPONGE-$k$ | 5.1 | 0.00 | 15.8 | 0.00 | 41.5 | 0.00 | — | | — | | — | | — | |
| 6 | LSPCD (OURS) | 20.0 | 0.49 | 46.2 | 0.56 | 137.6 | 0.33 | 57.1 | 0.43 | 96.1 | 0.53 | 103.4 | 0.47 | 58.7 | 0.54 |
| | SCG-MA | 14.6 | 0.46 | 45.5 | 0.42 | 84.9 | 0.62 | 37.8 | 0.17 | 102.6 | 0.07 | 88.8 | 0.52 | 57.5 | 0.42 |
| | SCG-MO | 15.2 | 0.46 | 47.0 | 0.41 | 55.6 | 0.72 | 34.6 | 0.29 | 111.6 | 0.22 | 129.2 | 0.26 | 41.8 | 0.24 |
| | SCG-B | 9.3 | 0.47 | 23.3 | 0.61 | 116.2 | 0.00 | 47.7 | 0.32 | 46.1 | 0.71 | 94.5 | 0.42 | 46.0 | 0.16 |
| | SCG-R | 6.9 | 0.41 | 10.4 | 0.79 | 50.3 | 0.36 | 7.9 | 0.46 | 18.3 | 0.74 | 43.3 | 0.30 | 3.3 | 0.42 |
| | KOCG-TOP-1 | 4.1 | 0.92 | 4.5 | 0.96 | 8.6 | 0.93 | 3.6 | 0.90 | 4.9 | 0.53 | 6.0 | 0.94 | 10.1 | 0.86 |
| | KOCG-TOP-$r$ | 3.6 | 0.87 | 3.1 | 0.96 | 4.0 | 0.97 | 3.3 | 0.91 | 1.5 | 0.99 | 6.8 | 0.89 | 3.6 | 0.77 |
| | BNC-$(k+1)$ | -4.2 | 0.25 | -1.1 | 0.97 | -0.8 | 0.94 | — | | — | | — | | — | |
| | BNC-$k$ | 5.2 | 0.01 | 15.8 | 0.00 | 41.5 | 0.00 | — | | — | | — | | — | |
| | SPONGE-$(k+1)$ | 1.3 | 0.15 | 1.0 | 0.86 | 1.0 | 0.92 | — | | — | | — | | — | |
| | SPONGE-$k$ | 5.1 | 0.00 | 15.8 | 0.00 | 41.5 | 0.00 | — | | — | | — | | — | |

is competitive with our method, it is limited to $k = 2$, requires tuning $\gamma$, and is significantly more complex (it requires training a graph neural network, leading to higher runtime, see Appendix G.9).

## 5   Conclusion

We proposed a novel formulation of the polarized community discovery (PCD) problem that emphasizes (reasonably) balanced communities in terms of size, addressing a key limitation of prior work, which typically optimizes *polarity* (Eq. 2) and often produces highly imbalanced clusterings. To tackle this, *we developed the first efficient and scalable local search method for PCD* and established a connection to block-coordinate Frank-Wolfe (FW) optimization. While the standard FW algorithm is known to achieve a convergence rate of $O(1/\sqrt{t})$ for general non-concave objectives [28, 39], we showed that, due to the specific structure of our objective in Eq. 6, our method achieves a significantly faster linear convergence rate of $O(1/t)$, despite the function being both non-concave and non-multilinear. Extensive experiments demonstrated that our method (LSPCD) consistently produces high-quality clusterings with reasonable cluster size balance, better aligning with practical expectations. Furthermore, we observed that the strong performance of local search algorithms in correlation clustering carried over to the PCD setting as well. Overall, our approach offers a compelling alternative in the PCD literature, both in terms of performance and simplicity (see Alg. 2). Alternative methods in the literature are significantly more complex.

## Acknowledgments

This work was partially supported by the Wallenberg AI, Autonomous Systems and Software Program (WASP) funded by the Knut and Alice Wallenberg Foundation.

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

## A   Comparing CC and PCD

The following proposition presents alternative objectives equivalent to maximizing Eq. 1 (i.e., solving the $k$-CC problem, see Problem 1). While this is known in the CC literature [9], we include a complete summary here to better motivate our problem formulation of PCD.

**Proposition 2.** *Problem 1 is equivalent to finding a clustering $S_{[k]}$ that maximizes any one of the four objectives in Eqs. 11-14 (i.e., they share all local maxima).*

$$N_{intra}^+ + N_{inter}^- \tag{11}$$

$$-N_{intra}^- - N_{inter}^+ \tag{12}$$

$$N_{intra}^+ - N_{intra}^- \tag{13}$$

$$N_{inter}^- - N_{inter}^+ \tag{14}$$

*Furthermore, maximizing any other combination of the four terms is not equivalent to Problem 1.*

All proofs can be found in Appendix B. The four formulations of CC shown in Prop. 2 respectively correspond to (i) maximizing agreements (Eq. 11), (ii) minimizing disagreements (Eq. 12), (iii) maximizing intra-cluster similarities (Eq. 13), and (iv) minimizing inter-cluster similarities (Eq. 14). Finally, we can combine all these notions into one single objective (i.e., Eq. 1). CC is an NP-hard problem, leading to the development of numerous approximation algorithms. Existing approximation algorithms maximize one of the five expressions discussed above [7], leading to differences in clustering performance, computational complexity and theoretical performance guarantees.

We now explain why Eq. 1, which incorporates all relevant terms, must be considered when neutral objects are allowed. Much prior work on PCD also optimize all terms, but often without providing a detailed justification for this choice. The next proposition provides such an intuition.

**Proposition 3.** *A clustering $S_{[k]}$ with neutral objects $S_0 = V \setminus \bigcup_{m \in [k]} S_m$ that maximizes one of the objectives in Eq. 1 or Eqs. 11-14 is not guaranteed to maximize any of the other objectives[2].*

From Prop. 3, we conclude that each term in Eq. 1 provides unique information when neutral objects are allowed, unlike the standard CC problem, where the different objectives are equivalent, as outlined by Prop. 2. This makes Eq. 1 the most reasonable objective for optimization in this context, as it effectively balances all contributing terms. Moreover, since each term captures unique aspects of the PCD problem, it may be beneficial to *weight them differently* to achieve an optimal trade-off.

## B   Proofs

**Theorem 1.** *Problem 2 (i.e., $k$-PCD) is NP-hard.*

*Proof.* Fix $\alpha, \beta \in \mathbb{R}$ to any values. Assume that we know which objects in $V$ should be assigned to the neutral set $S_0$ in the optimal solution to the $k$-PCD problem. Then, let $V' = V \setminus S_0$ and let $E'$ be the set of edges between objects in $V'$. Since no object in $V'$ should be neutral, the problem reduces to finding a partition of $V'$ that maximizes Eq. 3. We rewrite our objective in Eq. 3 as

$$
\begin{aligned}
(N_{intra}^+ - N_{intra}^-) &+ \alpha(N_{inter}^- - N_{inter}^+) - \beta \sum_{m \in [k]} |S_m|^2 = \\
&= \sum_{m \in [k]} \sum_{i,j \in S_m} A_{i,j} - \alpha \sum_{m \in [k]} \sum_{p \in [k] \setminus \{m\}} \sum_{\substack{i \in S_m \\ j \in S_p}} A_{i,j} - \beta \sum_{m \in [k]} |S_m|^2 \\
&= \sum_{m \in [k]} \sum_{i,j \in S_m} (A_{i,j} - \beta) - \alpha \sum_{m \in [k]} \sum_{p \in [k] \setminus \{m\}} \sum_{\substack{i \in S_m \\ j \in S_p}} A_{i,j}.
\end{aligned}
\tag{15}
$$

The second equality follows from Prop. 1. Defining $c_{\text{sim}} := \sum_{(i,j) \in E'} A_{i,j}$, we obtain:

---

[2]Unless $k = 2$, in which case Eq. 11 and Eq. 1 are equivalent as established in [6].

$$\sum_{m\in[k]}\sum_{\substack{p\in[k]\setminus\{m\}}}\sum_{\substack{i\in S_m \\ j\in S_p}} A_{i,j} = c_{\text{sim}} - \sum_{m\in[k]}\sum_{i,j\in S_m} A_{i,j}. \tag{16}$$

Substituting this into Eq. 15 and simplifying:

$$\sum_{m\in[k]}\sum_{i,j\in S_m}(A_{i,j}-\beta) - \alpha\sum_{m\in[k]}\sum_{p\in[k]\setminus\{m\}}\sum_{\substack{i\in S_m \\ j\in S_p}} A_{i,j} =$$

$$= \sum_{m\in[k]}\sum_{i,j\in S_m}(A_{i,j}-\beta) - \alpha\Big(c_{\text{sim}} - \sum_{m\in[k]}\sum_{i,j\in S_m} A_{i,j}\Big) \tag{17}$$

$$= (1+\alpha)\sum_{m\in[k]}\sum_{i,j\in S_m}(A_{i,j}-\beta) - \alpha c_{\text{sim}}.$$

Defining $A'_{i,j} = (1+\alpha)(A_{i,j}-\beta)$, we observe that since $c_{\text{sim}}$ is a constant across clustering solutions, the problem reduces to finding a partition of $V'$ that maximizes $\sum_{m\in[k]}\sum_{i,j\in S_m} A'_{i,j}$. This is equivalent to the *max correlation* objective (Eq. 13) applied to the transformed adjacency matrix $A'$. By Prop. 2, this objective is equivalent to the $k$-CC problem (Problem 1). Thus, solving the $k$-PCD problem requires solving the $k$-CC problem on the instance $G' = (V', E')$, meaning $k$-PCD is at least as hard as $k$-CC. Since correlation clustering is NP-hard [5, 20], we conclude that $k$-PCD is also NP-hard. $\qquad\square$

**Proposition 1.** *Our objective in Eq. 3 can be written as*

$$\sum_{m\in[k]}\sum_{i,j\in S_m}(A_{i,j}-\beta) - \alpha\sum_{m\in[k]}\sum_{p\in[k]\setminus\{m\}}\sum_{i\in S_m}\sum_{j\in S_p} A_{i,j}. \tag{5}$$

*Proof.* We have

$$(N_{\text{intra}}^+ - N_{\text{intra}}^-) + \alpha(N_{\text{inter}}^- - N_{\text{inter}}^+) - \beta\sum_{m\in[k]}|S_m|^2 =$$

$$= \sum_{m\in[k]}\sum_{i,j\in S_m} A_{i,j} - \alpha\sum_{m\in[k]}\sum_{p\in[k]\setminus\{m\}}\sum_{\substack{i\in S_m \\ j\in S_p}} A_{i,j} - \beta\sum_{m\in[k]}|S_m|^2$$

$$= \sum_{m\in[k]}\sum_{i,j\in S_m} A_{i,j} - \alpha\sum_{m\in[k]}\sum_{p\in[k]\setminus\{m\}}\sum_{\substack{i\in S_m \\ j\in S_p}} A_{i,j} - \beta\sum_{m\in[k]}\sum_{i,j\in S_m} 1 \tag{18}$$

$$= \sum_{m\in[k]}\sum_{i,j\in S_m} A_{i,j} - \alpha\sum_{m\in[k]}\sum_{p\in[k]\setminus\{m\}}\sum_{\substack{i\in S_m \\ j\in S_p}} A_{i,j} - \sum_{m\in[k]}\sum_{i,j\in S_m} \beta$$

$$= \sum_{m\in[k]}\sum_{i,j\in S_m}(A_{i,j}-\beta) - \alpha\sum_{m\in[k]}\sum_{p\in[k]\setminus\{m\}}\sum_{\substack{i\in S_m \\ j\in S_p}} A_{i,j}.$$

The second equality (line 3) holds because the number of pairs of objects inside cluster $m$ is $|S_m|^2$. A similar regularization is established in [10] for the minimum cut objective, where it is shown that optimizing this minimum cut objective regularized with $-\beta\sum_{m\in[k]}|S_m|^2$ is equivalent to optimizing the max correlation objective (Eq. 13) with similarities shifted by $\beta$. However, their result specifically considers the full network partitioning of unsigned networks, where the initial pairwise similarities are assumed non-negative. Moreover, they use a different regularization in practice: they shift the pairwise similarities so that the sum of the rows and columns of the similarity matrix becomes zero. $\qquad\square$

**Theorem 2.** *If $x_{[n]}^{(0)}$ in Alg 1 is discrete, the following hold. (a) For our problem (Eq. 7), the solution $x_i^*$ (line 4 of Alg. 1) is the basis vector $e_p$, where $p = \arg\max_{m\in\{0,\dots,k\}} G_{i,m}$ and the optimal value of the step size on line 6 is $\gamma = 1$. (b) Our objective function in Eq. 6 satisfies $(x_i^* - x_i^{(t)})\cdot G_{i,:} = f(x_{[n]}^*) - f(x_{[n]}^{(t)})$, where $x_{[n]}^*$ is $x_{[n]}^{(t)}$ with block $i$ modified to $x_i^*$.*

*Proof.* We begin by writing our objective function $f(\boldsymbol{x}_{[n]})$ in Eq. 6 as follows.

$$
\begin{aligned}
f(\boldsymbol{x}_{[n]}) &= \sum_{(i,j)\in E}\sum_{m\in[k]} x_{im}x_{jm}(A_{i,j}-\beta) - \alpha\sum_{(i,j)\in E}\sum_{m\in[k]}\sum_{p\in[k]\backslash\{m\}} x_{im}x_{jp}A_{i,j}\\
&= -\sum_{i\in[n]}\sum_{m\in[k]} x_{im}^2\beta + \sum_{\substack{(i,j)\in E\\i\neq j}}\sum_{m\in[k]} x_{im}x_{jm}(A_{i,j}-\beta) - \alpha\sum_{(i,j)\in E}\sum_{m\in[k]}\sum_{p\in[k]\backslash\{m\}} x_{im}x_{jp}A_{i,j}\\
&= -\sum_{i\in[n]}\sum_{m\in[k]} x_{im}\beta + \sum_{\substack{(i,j)\in E\\i\neq j}}\sum_{m\in[k]} x_{im}x_{jm}(A_{i,j}-\beta) - \alpha\sum_{(i,j)\in E}\sum_{m\in[k]}\sum_{p\in[k]\backslash\{m\}} x_{im}x_{jp}A_{i,j}.
\end{aligned}
\tag{19}
$$

In the second equality, we separate out the terms for $i = j$ and use that $A_{i,i} = 0$. In the third equality, we consider that $\boldsymbol{x}_{[n]}$ is a discrete solution. This makes the first term linear instead of being quadratic w.r.t. $x_{im}$, which is a crucial step in proving the theorem. Let $f(\boldsymbol{x}_i)$ denote $f(\boldsymbol{x}_{[n]})$ when treating all blocks other than $\boldsymbol{x}_i$ as constants. Then,

$$
\begin{aligned}
f(\boldsymbol{x}_i) &= -\sum_{m\in[k]} x_{im}\beta + 2\sum_{j\in[n]\backslash\{i\}}\sum_{m\in[k]} x_{im}x_{jm}(A_{i,j}-\beta) - 2\alpha\sum_{j\in[n]\backslash\{i\}}\sum_{m\in[k]}\sum_{p\in[k]\backslash\{m\}} x_{im}x_{jp}A_{i,j} + C\\
&= \sum_{m\in[k]} x_{im}\underbrace{\left(-\beta + 2\sum_{j\in[n]\backslash\{i\}}\left(x_{jm}(A_{i,j}-\beta) - \alpha\sum_{p\in[k]\backslash\{m\}} x_{jp}A_{i,j}\right)\right)}_{c_{im}} + C,
\end{aligned}
\tag{20}
$$

where $C$ denotes terms independent of $\boldsymbol{x}_i$. Define $\boldsymbol{c}_i \in \mathbb{R}^{k+1}$ with elements

$$
c_{im} := -\beta + 2\sum_{j\in[n]\backslash\{i\}}\left(x_{jm}(A_{i,j}-\beta) - \alpha\sum_{p\in[k]\backslash\{m\}} x_{jp}A_{i,j}\right), \quad \text{for } m \in [k], \quad c_{i0} := 0. \tag{21}
$$

Then, we obtain

$$
f(\boldsymbol{x}_i) = \sum_{m\in\{0,\ldots,k\}} x_{im}c_{im} + C = \boldsymbol{x}_i^T\boldsymbol{c}_i + C. \tag{22}
$$

Eq. 22 clearly illustrates that the contribution of the neutral component (index zero) of each $x_{im}$ is not included in the total objective (since $c_{i0} = 0$). From Eq. 22, the gradient of $f(\boldsymbol{x}_{[n]})$ w.r.t. $\boldsymbol{x}_i$ is

$$
\nabla_i f(\boldsymbol{x}_{[n]}) = \boldsymbol{c}_i. \tag{23}
$$

Let $\boldsymbol{c}_i^{(t)} = \nabla_i f(\boldsymbol{x}_{[n]}^{(t)})$ be the gradient of $f(\boldsymbol{x}_{[n]})$ evaluated at the current solution $\boldsymbol{x}_{[n]}^{(t)}$ (defined as in Eq. 21). The optimization problem on line 4 of Algorithm 1 is

$$
\boldsymbol{x}_i^* = \arg\max_{\boldsymbol{x}_i\in\Delta^{k+1}} \boldsymbol{x}_i^T\boldsymbol{c}_i^{(t)}. \tag{24}
$$

Since Eq. 24 is a linear program over the simplex $\Delta^{k+1}$, the optimal solution is obtained by setting $x_{im}^* = 1$ for $m = \arg\max_{m\in\{0,\ldots,k\}} c_{im}^{(t)}$ and $x_{ip}^* = 0$ for all $p \neq m$. This proves the first statement of part (a) of the theorem.

Next, we note that the difference $f(\boldsymbol{x}_{[n]}^*) - f(\boldsymbol{x}_{[n]}^{(t)})$ simplifies to $f(\boldsymbol{x}_i^*) - f(\boldsymbol{x}_i^{(t)})$ (where $f(\boldsymbol{x}_i)$ is defined in Eq. 20), since only the terms involving the variables in block $i$ change between $\boldsymbol{x}_{[n]}^*$ and $\boldsymbol{x}_{[n]}^{(t)}$. Therefore, we can derive the following.

$$\begin{aligned}
f(\boldsymbol{x}_{[n]}^*) - f(\boldsymbol{x}_{[n]}^{(t)}) &= f(\boldsymbol{x}_i^*) - f(\boldsymbol{x}_i^{(t)}) \\
&= ((\boldsymbol{x}_i^*)^T \boldsymbol{c}_i^* + C) - ((\boldsymbol{x}_i^{(t)})^T \boldsymbol{c}_i^{(t)} + C) \\
&= ((\boldsymbol{x}_i^*)^T \boldsymbol{c}_i^{(t)} + C) - ((\boldsymbol{x}_i^{(t)})^T \boldsymbol{c}_i^{(t)} + C) \\
&= ((\boldsymbol{x}_i^*)^T - (\boldsymbol{x}_i^{(t)})^T) \boldsymbol{c}_i^{(t)} \\
&= (\boldsymbol{x}_i^* - \boldsymbol{x}_i^{(t)}) \cdot \nabla_i f(\boldsymbol{x}_{[n]}^{(t)})
\end{aligned} \tag{25}$$

Here, $\boldsymbol{c}_i^*$ is defined as in Eq. 21 w.r.t. $\boldsymbol{x}_{[n]}^*$. Since $\boldsymbol{x}_{[n]}^*$ and $\boldsymbol{x}_{[n]}^{(t)}$ differ only in block $i$, and neither $\boldsymbol{c}_i^*$ nor $\boldsymbol{c}_i^{(t)}$ depend on the variables in block $i$, it follows that $\boldsymbol{c}_i^* = \boldsymbol{c}_i^{(t)}$, justifying the third equality. In Eq. 19, we assume that $\boldsymbol{x}_{[n]}$ is discrete. To ensure this property holds throughout, we require that both $\boldsymbol{x}_{[n]}^*$ and $\boldsymbol{x}_{[n]}^{(t)}$ remain discrete for all $t \in \{0, \dots, T\}$.

First, by assumption in the theorem, $\boldsymbol{x}_{[n]}^{(0)}$ is discrete. From part (a), we know that $\boldsymbol{x}_i^*$ is discrete, implying $\boldsymbol{x}_{[n]}^*$ is discrete as long as $\boldsymbol{x}_{[n]}^{(t)}$ is discrete. Furthermore, from Eq. 25, the optimal solution $\boldsymbol{x}_i^*$ in line 4 of Algorithm 1 maximally increases the objective, which ensures the optimal step size in line 6 is $\gamma = 1$ (proving the second statement of part (a)). Consequently, $\boldsymbol{x}_i^{(t+1)}$ remains discrete. By induction, this guarantees that $\boldsymbol{x}_{[n]}^{(t)}$ is discrete for all $t$, ensuring Eq. 25 holds for all $t \in \{0, \dots, T\}$. This completes the proof of part (b) of the theorem.

$\square$

**Theorem 3.** *The convergence rate of Alg. 2 is at least $nh_0/t = O(1/t)$, where $h_0 = \sum_{(i,j)\in E} |A_{i,j}|$.*

*Proof.* From Definition 1, we have that, in our case, the FW duality gap is defined as

$$g(\boldsymbol{x}_{[n]}) := \max_{\boldsymbol{s}_i \in \Delta^{k+1}, \forall i \in [n]} (\boldsymbol{s}_{[n]} - \boldsymbol{x}_{[n]}) \cdot \nabla f(\boldsymbol{x}_{[n]}). \tag{26}$$

Then, we recall that

$$\tilde{g}_t = \min_{0 \le l \le t-1} g(\boldsymbol{x}_{[n]}^{(l)}) \tag{27}$$

is the smallest duality gap observed in Alg. 1 up until step $t$. As established by [29] for general domains, the FW duality gap can be decomposed as follows.

$$\begin{aligned}
g(\boldsymbol{x}_{[n]}) &:= \max_{\boldsymbol{s}_i \in \Delta^{k+1}, \forall i \in [n]} (\boldsymbol{s}_{[n]} - \boldsymbol{x}_{[n]}) \cdot \nabla f(\boldsymbol{x}_{[n]}) \\
&= \max_{\boldsymbol{s}_i \in \Delta^{k+1}, \forall i \in [n]} \sum_{i \in [n]} (\boldsymbol{s}_i - \boldsymbol{x}_i) \cdot \nabla_i f(\boldsymbol{x}_{[n]}) \\
&= \sum_{i \in [n]} \underbrace{\max_{\boldsymbol{s}_i \in \Delta^{k+1}} (\boldsymbol{s}_i - \boldsymbol{x}_i) \cdot \nabla_i f(\boldsymbol{x}_{[n]})}_{:=g_i(\boldsymbol{x}_{[n]})}
\end{aligned} \tag{28}$$

Let $g_i(\boldsymbol{x}_{[n]}) := \max_{\boldsymbol{s}_i \in \Delta^{k+1}} (\boldsymbol{s}_i - \boldsymbol{x}_i) \cdot \nabla_i f(\boldsymbol{x}_{[n]})$ be the duality gap related to block $i$. We have that the FW duality gap is the sum of the gaps from each block: $g(\boldsymbol{x}_{[n]}) = \sum_{i \in [n]} g_i(\boldsymbol{x}_{[n]})$.

From Definition 2 (convergence rate), in order to prove the stated convergence rate, we need to show that $\mathbb{E}[\tilde{g}_t] \le nh_0/t$. The structure of our proof is similar to the proof of Theorem 2 in [43]. However, here we adapt it to our problem and make the proof more rigorous (including correction of a mistake in the proof by [43]). A key difference is that our objective in Eq. 6 is not multilinear in the blocks $i$. Then, as shown in Eq. 19 of Thm. 2, the first quadratic term can be transformed into a linear one by assuming a discrete solution (which we showed holds at every step $t$).

In Alg. 1, a block $i \in [n]$ is chosen uniformly at random (on line 3). Therefore, we have

$$\mathbb{E}[g_i(\boldsymbol{x}_{[n]}^{(t)})|\boldsymbol{x}_{[n]}^{(t)}] = \sum_{i \in [n]} P(i \text{ is selected}) g_i(\boldsymbol{x}_{[n]}^{(t)})$$

$$= \sum_{i \in [n]} \frac{1}{n} g_i(\boldsymbol{x}_{[n]}^{(t)})$$

$$= \frac{1}{n} \sum_{i \in [n]} \max_{\boldsymbol{s}_i^{(t)} \in \Delta^{k+1}} (\boldsymbol{s}_i - \boldsymbol{x}_i^{(t)}) \cdot \nabla_i f(\boldsymbol{x}_{[n]}^{(t)}) \quad (29)$$

$$= \frac{1}{n} g(\boldsymbol{x}_{[n]}^{(t)}).$$

We now take an expectation w.r.t. $\boldsymbol{x}_{[n]}^{(t)}$ on both sides and obtain

$$\mathbb{E}[\mathbb{E}[g_i(\boldsymbol{x}_{[n]}^{(t)})|\boldsymbol{x}_{[n]}^{(t)}]] = \frac{1}{n} \mathbb{E}[g(\boldsymbol{x}_{[n]}^{(t)})]$$

$$= \mathbb{E}[g_i(\boldsymbol{x}_{[n]}^{(t)})], \quad (30)$$

where the last equality follows from the Law of Total Expectation (i.e., that $\mathbb{E}_Y[\mathbb{E}_X[X|Y]] = \mathbb{E}_X[X]$, where $X$ and $Y$ are random variables). We therefore have that $\frac{1}{n}\mathbb{E}[g(\boldsymbol{x}_{[n]}^{(t)})] = \mathbb{E}[g_i(\boldsymbol{x}_{[n]}^{(t)})]$, where the expectation is w.r.t. all randomly chosen blocks $i$ before step $t$. Now, from Thm. 2 we have that our objective satisfies

$$f(\boldsymbol{x}_{[n]}^{(t+1)}) - f(\boldsymbol{x}_{[n]}^{(t)}) = g_i(\boldsymbol{x}_{[n]}^{(t)}) = \max_{\boldsymbol{s}_i^{(t)} \in \Delta^{k+1}} (\boldsymbol{s}_i - \boldsymbol{x}_i^{(t)}) \cdot \nabla_i f(\boldsymbol{x}_{[n]}^{(t)}). \quad (31)$$

Then, we have

$$\frac{1}{n} \sum_{t=0}^{T-1} \mathbb{E}[g(\boldsymbol{x}^{(t)})] = \sum_{t=0}^{T-1} \mathbb{E}[g_i(\boldsymbol{x}_{[n]}^{(t)})]$$

$$= \sum_{t=0}^{T-1} \mathbb{E}[f(\boldsymbol{x}_{[n]}^{(t+1)}) - f(\boldsymbol{x}_{[n]}^{(t)})] \quad (32)$$

$$= \mathbb{E}[f(\boldsymbol{x}_{[n]}^{(T)})] - f(\boldsymbol{x}_{[n]}^{(0)})$$

$$\leq OPT - f(\boldsymbol{x}_{[n]}^{(0)}),$$

where the third equality is due to the *telescoping rule* and $OPT$ is the objective value of the optimal clustering solution to Problem 1 ($k$-PCD). On the other hand, we have

$$\frac{1}{n} \sum_{t=0}^{T-1} \mathbb{E}[g(\boldsymbol{x}^{(t)})] \geq \frac{T}{n} \mathbb{E}[\tilde{g}_T], \quad (33)$$

where $\tilde{g}_t$ is defined as in Eq. 27 (the smallest gap observed until step $t$). Therefore,

$$\frac{T}{n} \mathbb{E}[\tilde{g}_T] \leq OPT - f(\boldsymbol{x}^{(0)})$$

$$\Rightarrow \mathbb{E}[\tilde{g}_T] \leq \frac{n(OPT - f(\boldsymbol{x}^{(0)}))}{T}. \quad (34)$$

The value of $OPT$ depends on the particular instance. In order to obtain an instance-independent bound, we use that $OPT - f(\boldsymbol{x}^{(0)}) \leq \sum_{(i,j) \in E} |A_{i,j}|$ resulting in

$$\mathbb{E}[\tilde{g}_T] \leq \frac{n \sum_{(i,j) \in E} |A_{i,j}|}{T}. \quad (35)$$

which we aimed to show since it holds for any $T$. □

**Corollary 1.** *From Thm. 2, if $\boldsymbol{x}_{[n]}^{(0)}$ is discrete, solving the optimization problem in Eq. 7 using Alg. 1 is equivalent to executing the local search procedure described in Alg. 2.*

*Proof.* From part (a) of Thm. 2, the current solution, $\boldsymbol{x}_{[n]}^{(t)}$, remains discrete (i.e., hard cluster assignments) at every step of Alg. 1 for all $i \in [n]$. Moreover, each step of Alg. 1 consists of placing object $i$ in the cluster $m \in \{0, \dots, k\}$ with maximal gradient $G_{i,m}$. By part (b) of Thm. 2, this is equivalent to placing object $i$ in the cluster that maximally improves our objective in Eq. 3. □

**Theorem 4.** *Let $S_{[k]}$ be the current clustering of our local search procedure, with neutral objects $S_0 = V \setminus \bigcup_{m \in [k]} S_m$. The gradient can then be expressed as follows.*

$$G_{i,m} = 2(1+\alpha) \sum_{j \in S_m} A_{i,j} - 2\alpha \sum_{p \in [k]} \sum_{j \in S_p} A_{i,j} - 2\beta|S_m| + 2\beta\mathbf{1}[i \in S_m] - \beta \tag{9}$$

*for all $m \in [k]$ and $G_{i,0} = 0$.*

*Proof.* From Thm. 2, we recall that since the current solution $\boldsymbol{x}_{[n]}^{(t)}$ always remains discrete, our objective can be written as

$$f(\boldsymbol{x}_{[n]}^{(t)}) = -\sum_{i \in [n]} \sum_{m \in [k]} x_{im}^{(t)}\beta + \sum_{\substack{(i,j) \in E \\ i \neq j}} \sum_{m \in [k]} x_{im}^{(t)} x_{jm}^{(t)}(A_{i,j} - \beta) - \alpha \sum_{(i,j) \in E} \sum_{m \in [k]} \sum_{p \in [k] \setminus \{m\}} x_{im}^{(t)} x_{jp}^{(t)} A_{i,j}. \tag{36}$$

We let $f(\boldsymbol{x}_i^{(t)})$ denote $f(\boldsymbol{x}_{[n]}^{(t)})$ when treating all blocks other than $\boldsymbol{x}_i^{(t)}$ as constants. Then,

$$f(\boldsymbol{x}_i^{(t)}) = \sum_{m \in [k]} x_{im}^{(t)} \underbrace{\left( -\beta + 2 \sum_{j \in [n] \setminus \{i\}} x_{jm}^{(t)}(A_{i,j} - \beta) - 2\alpha \sum_{j \in [n] \setminus \{i\}} \sum_{p \in [k] \setminus \{m\}} x_{jp}^{(t)} A_{i,j} \right)}_{c_{im}} + C. \tag{37}$$

Therefore, we have

$$G_{i,m} := [\nabla_i f(\boldsymbol{x}_{[n]}^{(t)})]_m = c_{im}, \quad \text{for } m \in [k]. \tag{38}$$

This holds because neither $c_{im}$ nor $C$ depend on $x_{im}^{(t)}$. Furthermore, since $x_{i0}$ does not show up in Eq. 37, everything in Eq. 37 is a constant w.r.t. $x_{i0}$. We therefore have

$$G_{i,0} := [\nabla_i f(\boldsymbol{x}_{[n]}^{(t)})]_0 = 0. \tag{39}$$

By noting that $\boldsymbol{x}^{(t)}$ is discrete, we can rewrite $c_{im}$ for $m \in [k]$ as follows.

$$
\begin{aligned}
c_{im} &= -\beta + 2 \sum_{j \in [n] \setminus \{i\}} x_{jm}^{(t)}(A_{i,j} - \beta) - 2\alpha \sum_{j \in [n] \setminus \{i\}} \sum_{p \in [k] \setminus \{m\}} x_{jp}^{(t)} A_{i,j} \\
&= -\beta + 2 \sum_{j \in [n] \setminus \{i\}} x_{jm}^{(t)} A_{i,j} - 2 \sum_{j \in [n] \setminus \{i\}} \beta - 2\alpha \sum_{j \in [n] \setminus \{i\}} \sum_{p \in [k] \setminus \{m\}} x_{jp}^{(t)} A_{i,j} \\
&= -\beta + 2 \sum_{j \in S_m \setminus \{i\}} A_{i,j} - 2 \sum_{j \in S_m \setminus \{i\}} \beta - 2\alpha \sum_{p \in [k] \setminus \{m\}} \sum_{j \in S_p \setminus \{i\}} A_{i,j} \\
&= -\beta + 2 \sum_{j \in S_m \setminus \{i\}} A_{i,j} - 2|S_m|\beta + 2\beta\mathbf{1}[i \in S_m] - 2\alpha \sum_{p \in [k] \setminus \{m\}} \sum_{j \in S_p \setminus \{i\}} A_{i,j}.
\end{aligned}
\tag{40}
$$

In the last equality we use $-2\sum_{j\in S_m\setminus\{i\}}\beta = -2|S_m|\beta + 2\beta\mathbf{1}[i \in S_m]$. We note that computing the final expression in Eq. 40 is $O(k^2n)$, due to the last term. However, by noting that $\sum_{p\in[k]\setminus\{m\}}\sum_{j\in S_p\setminus\{i\}} A_{i,j} = \sum_{j\notin S_0} A_{i,j} - \sum_{j\in S_m\setminus\{i\}} A_{i,j}$ we can derive the following.

$$
\begin{aligned}
c_{im} &= -\beta + 2\sum_{j\in S_m\setminus\{i\}} A_{i,j} - 2|S_m|\beta + 2\beta\mathbf{1}[i \in S_m] - 2\alpha\sum_{p\in[k]\setminus\{m\}}\sum_{j\in S_p\setminus\{i\}} A_{i,j} \\
&= -\beta + 2\sum_{j\in S_m\setminus\{i\}} A_{i,j} - 2|S_m|\beta + 2\beta\mathbf{1}[i \in S_m] - 2\alpha\Big(\sum_{j\notin S_0} A_{i,j} - \sum_{j\in S_m\setminus\{i\}} A_{i,j}\Big) \\
&= -\beta + 2\sum_{j\in S_m\setminus\{i\}} (A_{i,j} + \alpha A_{i,j}) - 2|S_m|\beta + 2\beta\mathbf{1}[i \in S_m] - 2\alpha\sum_{j\notin S_0} A_{i,j} \\
&= -\beta + 2(1+\alpha)\sum_{j\in S_m\setminus\{i\}} A_{i,j} - 2|S_m|\beta + 2\beta\mathbf{1}[i \in S_m] - 2\alpha\sum_{j\notin S_0} A_{i,j}
\end{aligned}
\tag{41}
$$

Then, we note that $2\sum_{j\notin S_0} A_{i,j} = 2\sum_{p\in[k]}\sum_{j\in S_p} A_{i,j} = \sum_{p\in[k]} M_{i,p}$ (sum of all similarities from object $i$ to all non-neutral objects) and that $A_{i,i} = 0$ (by assumption). This proves the statement of the theorem. The expression in Eq. 9 can be computed in $O(kn)$ since $\sum_{p\in[k]}\sum_{j\in S_p} A_{i,j}$ is a constant w.r.t. different clusters $m \in [k]$, and can therefore be precomputed (see Alg. 3). It may appear reasonable to remove the term $\sum_{p\in[k]}\sum_{j\in S_p} A_{i,j}$, since it is constant with respect to the cluster $m \in [k]$. However, this is invalid because neutral objects are allowed. Specifically, if $c_{im} < 0$ for all $m \in [k]$, assigning object $i$ to the neutral set is optimal, as $c_{i0} = 0$. The term $\sum_{p\in[k]}\sum_{j\in S_p} A_{i,j}$ must therefore be retained, as it affects whether an object is assigned to the neutral set. $\qquad\square$

**Proposition 2.** *Problem 1 is equivalent to finding a clustering $S_{[k]}$ that maximizes any one of the four objectives in Eqs. 11-14 (i.e., they share all local maxima).*

$$
N_{intra}^+ + N_{inter}^- \tag{11}
$$

$$
-N_{intra}^- - N_{inter}^+ \tag{12}
$$

$$
N_{intra}^+ - N_{intra}^- \tag{13}
$$

$$
N_{inter}^- - N_{inter}^+ \tag{14}
$$

*Furthermore, maximizing any other combination of the four terms is not equivalent to Problem 1.*

*Proof.* We begin by defining the following quantities, which are constants w.r.t. different clustering solutions for the $k$-CC problem.

$$
c_{\text{sim}} := \sum_{(i,j)\in E} A_{i,j}. \tag{42}
$$

$$
c_{\text{abs}} := \sum_{(i,j)\in E} |A_{i,j}|. \tag{43}
$$

The five objectives can be written as follows.

$$f^{\text{full}} := N_{\text{intra}}^{+} - N_{\text{intra}}^{-} + N_{\text{inter}}^{-} - N_{\text{inter}}^{+} = \sum_{m \in [k]} \sum_{i,j \in S_m} A_{i,j} - \sum_{m \in [k]} \sum_{p \in [k] \setminus \{m\}} \sum_{\substack{i \in S_m \\ j \in S_p}} A_{i,j}$$

$$f^{\text{MaxAgree}} := N_{\text{intra}}^{+} + N_{\text{inter}}^{-}$$
$$= \sum_{m \in [k]} \sum_{i,j \in S_m} A_{i,j}^{+} - \sum_{m \in [k]} \sum_{p \in [k] \setminus \{m\}} \sum_{\substack{i \in S_m \\ j \in S_p}} A_{i,j}^{-}$$
$$= \frac{1}{2} \sum_{m \in [k]} \sum_{i,j \in S_m} (|A_{i,j}| + A_{i,j}) - \frac{1}{2} \sum_{m \in [k]} \sum_{p \in [k] \setminus \{m\}} \sum_{\substack{i \in S_m \\ j \in S_p}} (|A_{i,j}| - A_{i,j})$$

$$f^{\text{MinDisagree}} := N_{\text{intra}}^{-} + N_{\text{inter}}^{+}$$
$$= \sum_{m \in [k]} \sum_{i,j \in S_m} A_{i,j}^{-} + \sum_{m \in [k]} \sum_{p \in [k] \setminus \{m\}} \sum_{\substack{i \in S_m \\ j \in S_p}} A_{i,j}^{+}$$
$$= \frac{1}{2} \sum_{m \in [k]} \sum_{i,j \in S_m} (|A_{i,j}| - A_{i,j}) + \frac{1}{2} \sum_{m \in [k]} \sum_{p \in [k] \setminus \{m\}} \sum_{\substack{i \in S_m \\ j \in S_p}} (|A_{i,j}| + A_{i,j})$$

$$f^{\text{MaxCorr}} := N_{\text{intra}}^{+} - N_{\text{intra}}^{-} = \sum_{m \in [k]} \sum_{i,j \in S_m} A_{i,j}$$

$$f^{\text{MinCut}} := -N_{\text{inter}}^{-} + N_{\text{inter}}^{+} = \sum_{m \in [k]} \sum_{p \in [k] \setminus \{m\}} \sum_{\substack{i \in S_m \\ j \in S_p}} A_{i,j}$$

$$(44)$$

Given this, we observe the following connection between the objectives.

$$f^{\text{MaxAgree}} = \frac{1}{2} \sum_{m \in [k]} \sum_{i,j \in S_m} (|A_{i,j}| + A_{i,j}) - \frac{1}{2} \sum_{m \in [k]} \sum_{p \in [k] \setminus \{m\}} \sum_{\substack{i \in S_m \\ j \in S_p}} (|A_{i,j}| - A_{i,j})$$
$$= \frac{1}{2} f^{\text{full}} + \frac{1}{2} c_{\text{abs}}$$
$$= -f^{\text{MinDisagree}} + c_{\text{abs}}$$
$$= \frac{1}{2} f^{\text{MaxCorr}} - \frac{1}{2} f^{\text{MinCut}} + \frac{1}{2} c_{\text{abs}}$$
$$= f^{\text{MaxCorr}} - \frac{1}{2} c_{\text{sim}} + \frac{1}{2} c_{\text{abs}}$$
$$= -f^{\text{MinCut}} + \frac{1}{2} c_{\text{sim}} + \frac{1}{2} c_{\text{abs}}$$

$$(45)$$

The above establishes that they are all equal up to constants. We prove the last statement of the proposition by counterexample. We consider the graph $V = \{1, 2, 3\}$ with edge weights $A_{1,2} = +1$, $A_{2,3} = +1$, $A_{1,3} = -1$. The possible clustering solutions (partitions) are:

$$S^{(1)} = \{\{1, 2, 3\}\}, \quad S^{(2)} = \{\{1, 2\}, \{3\}\},$$
$$S^{(3)} = \{\{1, 3\}, \{2\}\}, \quad S^{(4)} = \{\{2, 3\}, \{1\}\}, \quad S^{(5)} = \{\{1\}, \{2\}, \{3\}\}.$$

In Table 2, we list all linear combinations of the terms $N_{\text{intra}}^{+}, -N_{\text{intra}}^{-}, N_{\text{inter}}^{-}, -N_{\text{inter}}^{+}$ evaluated on each of the five clustering solutions. Expectedly (from the first part of the proposition), the five objectives $f^{\text{full}}, f^{\text{MaxAgree}}, f^{\text{MinDisagree}}, f^{\text{MaxCorr}}, f^{\text{MinCut}}$ all produce the same ranking of these solutions. In contrast, every other combination of the terms ranks at least one solution differently compared to these five. This proves the last statement of the proposition.

$$\square$$

Table 2: All sums of $N_{\text{intra}}^+$, $-N_{\text{intra}}^-$, $N_{\text{inter}}^-$, $-N_{\text{inter}}^+$ and their values on the five partitions $S^{(m)}$. In parentheses we indicate the known name when the combination corresponds to one of the five standard correlation-clustering objectives (or its negative).

| IDX | COMBINATION | $S^{(1)}$ | $S^{(2)}$ | $S^{(3)}$ | $S^{(4)}$ | $S^{(5)}$ |
|---|---|---|---|---|---|---|
| 1 | $N_{\text{INTRA}}^+$ | 2 | 1 | 0 | 1 | 0 |
| 2 | $-N_{\text{INTRA}}^-$ | -1 | 0 | -1 | 0 | 0 |
| 3 | $N_{\text{INTER}}^-$ | 0 | 1 | 0 | 1 | 1 |
| 4 | $-N_{\text{INTER}}^+$ | 0 | -1 | -2 | -1 | -2 |
| 5 | $N_{\text{INTRA}}^+ - N_{\text{INTRA}}^-$ ($f^{\text{MAXCORR}}$) | 1 | 1 | -1 | 1 | 0 |
| 6 | $N_{\text{INTRA}}^+ + N_{\text{INTER}}^-$ ($f^{\text{MAXAGREE}}$) | 2 | 2 | 0 | 2 | 1 |
| 7 | $N_{\text{INTRA}}^+ - N_{\text{INTER}}^+$ | 2 | 0 | -2 | 0 | -2 |
| 8 | $-N_{\text{INTRA}}^- + N_{\text{INTER}}^-$ | -1 | 1 | -1 | 1 | 1 |
| 9 | $-N_{\text{INTRA}}^- - N_{\text{INTER}}^+$ ($-f^{\text{MINDISAGREE}}$) | -1 | -1 | -3 | -1 | -2 |
| 10 | $N_{\text{INTER}}^- - N_{\text{INTER}}^+$ ($-f^{\text{MINCUT}}$) | 0 | 0 | -2 | 0 | -1 |
| 11 | $N_{\text{INTRA}}^+ - N_{\text{INTRA}}^- + N_{\text{INTER}}^-$ | 1 | 2 | -1 | 2 | 1 |
| 12 | $N_{\text{INTRA}}^+ - N_{\text{INTRA}}^- - N_{\text{INTER}}^+$ | 1 | 0 | -3 | 0 | -2 |
| 13 | $N_{\text{INTRA}}^+ + N_{\text{INTER}}^- - N_{\text{INTER}}^+$ | 2 | 1 | -2 | 1 | -1 |
| 14 | $-N_{\text{INTRA}}^- + N_{\text{INTER}}^- - N_{\text{INTER}}^+$ | -1 | 0 | -3 | 0 | -1 |
| 15 | $N_{\text{INTRA}}^+ - N_{\text{INTRA}}^- + N_{\text{INTER}}^- - N_{\text{INTER}}^+$ ($f^{\text{FULL}}$) | 1 | 1 | -3 | 1 | -1 |

**Proposition 3.** *A clustering $S_{[k]}$ with neutral objects $S_0 = V \setminus \bigcup_{m \in [k]} S_m$ that maximizes one of the objectives in Eq. 1 or Eqs. 11-14 is not guaranteed to maximize any of the other objectives*[3].

*Proof.* If an object transitions from neutral to non-neutral, it may introduce agreements (positive intra-cluster or negative inter-cluster similarities) and/or disagreements (negative intra-cluster or positive inter-cluster similarities). An exception is objects with zero degree (zero similarity to all others), which can be assigned as neutral or non-neutral without affecting any of the five objectives. Thus, we only consider non-zero degree objects in the remainder of the proof.

The *max agreement* objective (Eq. 11) considers only agreements. Making an object non-neutral either increases or maintains the objective but never decreases it, ensuring all objects become non-neutral. Conversely, the *min disagreement* objective (Eq. 12) considers only disagreements. Making an object non-neutral either decreases or maintains the objective but never improves it, ensuring all objects remain neutral.

Now, consider a clustering with $k$ non-neutral clusters, where all intra-cluster similarities are $+1$ and all inter-cluster similarities are $-1$. If an unassigned object $i \in V$ has similarity $+1$ to all others, the *max correlation* objective (Eq. 13) assigns it to the largest non-neutral cluster, while the *minimum cut* objective (Eq. 14) keeps it neutral. For $k > 2$, the full objective (Eq. 1) places $i$ in the largest non-neutral cluster if its size exceeds the sum of all others; otherwise, it remains neutral. Conversely, if object $i$ has similarity $-1$ to all others, *max correlation* keeps it neutral, whereas *minimum cut* assigns it to the smallest non-neutral cluster. For $k > 2$, the full objective may assign $i$ as neutral or non-neutral depending on cluster sizes.

Therefore, we conclude that *max agreement* and *min disagreement* differ fundamentally, always assigning all objects as non-neutral or neutral, respectively. Furthermore, from the two counterexamples above, *max correlation*, *minimum cut*, and the full objective are not equivalent and none of them guarantee that all objects are either neutral or non-neutral in all cases (meaning they are all different from *max agreement* and *min disagreement* in general).

For $k = 2$, the full objective always increases (or remains constant) when an object is made non-neutral, aligning it with *max agreement*. To see this, consider a clustering with $k = 2$ non-neutral clusters, and let $M_{i,m} = \sum_{j \in S_m} A_{i,j}$ be the total similarity of object $i$ to cluster $m$. The impact on the objective when assigning $i$ to $m$ is $M_{i,m} - M_{i,p}$, where $p$ is the other cluster. Since this difference is always positive when $i$ is placed in its most similar cluster, assigning $i$ as non-neutral

---

[3]Unless $k = 2$, in which case Eq. 11 and Eq. 1 are equivalent as established in [6].

always improves the objective. Then, since all objects are non-neutral, the problem is equivalent to the $k$-CC problem where we know *max agreement* and the full objective are equivalent (from Prop. 2). For $k > 2$, this reasoning no longer holds, as contributions from other clusters can outweigh the within-cluster similarity to the most similar cluster (i.e., making the total contribution negative), potentially making neutrality optimal. However, we note that in our final objective (Eq. 3), when $\alpha$ and $\beta$ are involved, all terms will contribute with unique information even for $k = 2$.

$\square$

## C   Impact of $\alpha$ and $\beta$

In this section, we analyze the impact of $\alpha$ and $\beta$ in Eq. 3. In Appendix G.8 we investigate their impact experimentally. We begin by stating the following proposition.

**Proposition 4.** *(a) There exists a $\xi_1 < 0$ such that for any $\beta \leq \xi_1$, there is a clustering solution maximizing Eq. 3 where all the objects are assigned to a single non-neutral cluster. (b) Conversely, there exists a $\xi_2 > 0$ such that for any $\beta \geq \xi_2$, there is a clustering solution maximizing Eq. 3 where all the objects are neutral.*

*Proof.* By examining the gradient in Eq. 9, we observe that the dominant term involving $\beta$ is $-\beta|S_m|$. Consequently, making $\beta$ large and negative *increases* the incentive to assign objects to non-neutral clusters. Moreover, since $-\beta|S_m|$ scales with cluster size, the local search procedure will favor placing an object $i$ in the largest non-neutral cluster. If $\beta$ is sufficiently large and negative, this term will completely dominate the objective, ensuring that no object is assigned to the neutral set (as the contribution to all non-neutral clusters remains positive). Ultimately, all objects will be placed in the largest non-neutral cluster.

Similarly, if $\beta$ is made very large and positive, $-\beta|S_m|$ will eventually dominate the objective, making the contribution to every non-neutral cluster negative for all objects. As a result, all objects will be assigned to the neutral set. $\square$

From Prop. 4, we understand the extreme cases of $\beta$: (a) a small negative $\beta$ results in a maximally imbalanced non-neutral clustering (i.e., all objects in one non-neutral cluster), while (b) a large positive $\beta$ makes all objects neutral. For intermediate $\beta \in [\xi_1, \xi_2]$, we analyze the gradient in Eq. 9. Increasing $\beta$ strictly reduces the contribution of object $i$ to each cluster $m \in [k]$, but since the term $-2\beta|S_m|$ scales with cluster size, larger clusters become less favorable, promoting balance. If $\beta$ is large enough, it forces $G_{i,m} < 0$ for all $m \in [k]$, making neutrality optimal for object $i$. Note that this is more likely for low-degree objects, implying that high-degree objects (with clear cluster assignment) are more likely to remain non-neutral, resulting in dense non-neutral clusters. Consequently, increasing $\beta$ leads to smaller (i.e., more neutral objects) and denser non-neutral clusters, while maintaining balanced, as desired.

The parameter $\alpha$ has been studied in prior work [12, 44]. From Eq. 3, $\alpha$ balances maximizing intra-similarities and minimizing inter-similarities, which translates to a trade-off between cohesion within clusters and separation between them. A heuristic choice of $\alpha = 1/(k-1)$ was proposed in [44], based on the observation that the number of intra-similarities scale linearly with $k$, while the number of inter-similarities grow quadratically. This choice prevents inter-similarities from dominating the objective. Finally, the term $-\alpha\phi_i$ indicates that $\alpha$ influences whether object $i$ becomes neutral, underscoring the need to account for inter-similarities in the objective (as suggested in Section 2.2).

## D   Limitations of Polarity

To illustrate the limitation of polarity (Eq 2), we refer to Example 2 from [21], which considers a signed graph with 12 objects: $\{A, B, C, D, E, F, G, H, I, J, K, L\}$. The sign of each similarity can be found in Figure 3 of [21]. The study evaluates the following three clustering solutions:

- $S^{(1)} = \{\{A, B, C, D\}, \{E, F, G, H\}\}$
- $S^{(2)} = \{\{A, B, C, D\}, \{E, F, G, H, I, J, K, L\}\}$
- $S^{(3)} = \{\emptyset, \{E, F, G, H, I, J, K, L\}\}$

The polarity values for these solutions are: $\text{Polarity}(S^{(1)}) = (20 + 10)/8 = 3.75$, $\text{Polarity}(S^{(2)}) = (38 + 6)/12 = 3.67$, and $\text{Polarity}(S^{(3)}) = (30 + 0)/8 = 3.75$.

Although $S^{(1)}$ and $S^{(3)}$ achieve the same polarity score, $S^{(1)}$ is significantly more balanced, making it the more reasonable choice. In contrast, evaluating our objective (Eq. 3) for the same solutions, we obtain $S^{(1)} : (20 + 10) - (4^2 + 4^2) = -2$, $S^{(2)} : (38 + 6) - (4^2 + 8^2) = -36$, and $S^{(3)} : (30 + 0) - 8^2 = -34$.

Our objective function still identifies $S^{(2)}$ as the worst solution (consistent with polarity), but it strongly favors $S^{(1)}$ over $S^{(3)}$ due to its better balance. Here, we assume $\alpha = 1/(k - 1)$ (which is 1 since $k = 2$) for consistency with polarity, and $\beta = 1$.

# E   Motivation for Discovering Balanced Communities

From an optimization perspective, the graph-clustering literature recognizes that objectives that simply maximize intra-cluster similarity or minimize inter-cluster similarity often degenerate into trivial solutions in which all objects are assigned to a single (or very few) clusters. A classic instance is the *minimum cut*, which minimizes inter-cluster similarity yet exhibits this pathology on both signed and unsigned graphs. The usual remedy is to normalize the objective by a quantity that reflects cluster size or degree, giving rise to alternative measures such as the (signed) *ratio cut* and the *normalized cut* [11, 10]. Crucially, such normalization can yield solutions whose value with respect to the unnormalized criterion is worse, while better coinciding with the true underlying clusters. In this work we tackle an analogous limitation in the context of PCD, where neutral objects are permitted.

We next motivate—through a few illustrative examples—why clusterings that are more balanced in size are frequently better aligned with ground-truth clusterings observed in real-world settings.

As discussed in [21], many social environments—from online forums and political systems to scientific institutions—can benefit from balanced community structures in signed networks. Such balance helps ensure that diverse viewpoints or specializations are sufficiently represented and reduces the chance that one perspective overwhelmingly dominates. This, in turn, promotes constructive debate, encourages critical thinking, and helps broaden individuals' perspectives—ultimately limiting echo chambers and mitigating the spread of misinformation. Below, we provide a few concrete examples.

In market research and product development, balanced communities can provide deeper insights into consumer preferences. By identifying groups of comparable size (rather than one massive consumer segment overshadowing niche but meaningful ones), organizations can more effectively tailor products or marketing strategies to each community, thereby predicting market trends more accurately.

Academic Research Networks can also profit from balanced structures. Imagine a signed network in a university, where positive edges link researchers working on similar topics (e.g., physics, computer science, mathematics), and negative edges indicate differing research domains. Because universities aim to cover a broad range of subjects—each with enough faculty to maintain healthy research output—a balanced partition of the signed network (i.e., each discipline having a solid, non-negligible presence) offers a better representation of the university's diverse pursuits. In such a setting, neutral nodes might be faculty primarily engaged in administrative roles or other staff members not actively involved in research.

Another example is social networks where communities are formed around specific interests or activities (e.g., online book clubs, gaming groups). These communities grow naturally to a size where interaction remains high, and members can still engage meaningfully with each other, without the network becoming too large or fragmented, i.e., the size of the group grows until it reaches a point where communication remains efficient and personal connections are preserved. As the group grows beyond a certain size, it may split into smaller subgroups, keeping the communities balanced in size.

In all these scenarios, balanced community detection ensures that no single faction's needs or views go unnoticed. This inclusivity promotes more representative outcomes, strengthens consensus-building, and leads to more robust decisions or insights.

# F  Comparison with $\gamma$-polarity Objective from [21]

The $\gamma$-polarity objective introduced in [21] is defined specifically for the case of $k = 2$ clusters. In this setting, we have only two non-neutral groups: $S_1$ and $S_2$. Let $s_{\max} = \max(|S_1|, |S_2|)$ and $s_{\min} = \min(|S_1|, |S_2|)$. The $\gamma$-polarity is then defined as

$$\frac{N_{\text{intra}}^+ - N_{\text{intra}}^- + N_{\text{inter}}^- - N_{\text{inter}}^+}{(s_{\max} - s_{\min})\gamma + 2s_{\min}}. \tag{46}$$

This formulation is similar to the polarity objective in Eq. 2, which instead uses the denominator $|S_0| + |S_1|$. Like polarity, $\gamma$-polarity penalizes large or imbalanced clusters through a normalization term related to cluster sizes. However, the specific form of the denominator in Eq. 46 additionally enforces balance between $S_1$ and $S_2$.

In contrast, our objective promotes balanced clustering through an additive regularization term which supports an arbitrary number of clusters $k$. A key advantage of additive regularization is its simplicity: it effectively corresponds to shifting intra-cluster similarities by $-\beta$ (see Prop. 1). This straightforward modification enables us to establish a linear convergence rate (Thm. 3) and significantly improves computational efficiency (see Section 3.4). In comparison, obtaining similar theoretical guarantees and efficiency via local search on polarity-based objectives, including $\gamma$-polarity, is likely infeasible or substantially more complex.

In summary, our objective facilitates the derivation of a local search algorithm for the PCD problem that scales efficiently to large graphs.

# G  Experiments: More Details and Further Results

All experiments were conducted locally on a single machine with an Intel Core i9-10850k CPU and 64 GB of RAM.

## G.1  Datasets

Following [44], we consider the following widely studied real-world signed networks. **WoW-EP8** (`W8`) [25] represents interactions among authors in the 8th EU Parliament legislature, where edge signs indicate collaboration or competition. **Bitcoin** (`BTC`) [31] is a trust-distrust network of users trading on the Bitcoin OTC platform. **WikiVot** (`WikiV`) [31] records positive and negative votes for Wikipedia admin elections. **Referendum** (`REF`) [30] captures tweets about the 2016 Italian constitutional referendum, with edge signs indicating whether users share the same stance. **Slashdot** (`SD`) [31] is a friend-foe network from the Slashdot Zoo feature. **WikiCon** (`WikiC`) [26] tracks positive and negative interactions between users editing English Wikipedia. **Epinions** (`EP`) [31] represents the trust-distrust relationships in the Epinions online social network. **WikiPol** (`WikiP`) [33] captures interactions among users editing Wikipedia pages on political topics.

## G.2  Baselines

For SCG, we use the public implementation from [40]. For KOCG, we use the public implementation from [32] with default hyperparameters: $\alpha = 1/(k-1)$, $\beta = 50$ (note that the purpose of this $\beta$ differs from the one used in our paper), and $\ell = 5000$. For the spectral methods SPONGE and BNC, we use the public implementations from [37]. Following [44], for SPONGE, we evaluate both the *unnormalized* and *symmetric normalized* versions and report results for the best-performing method. For N2PC [21], we use default parameters for training their framework (based on GNN). We use the public implementation provided by the authors.

## G.3  Description of Synthetic Datasets

We employ the *modified signed stochastic block model* (m-SSBM), which was specifically designed to generated synthetic graphs with planted ground-truth communities for PCD. The m-SSBM model is parameterized by four variables: (i) $n$, the total number of nodes; (ii) $k$, the number of non-neutral clusters; (iii) $\ell$, the size of each non-neutral cluster; and (iv) $\eta \in [0, 1]$, which controls the edge probabilities. Edges within the same cluster are positive with probability $1 - \eta$, and negative or absent

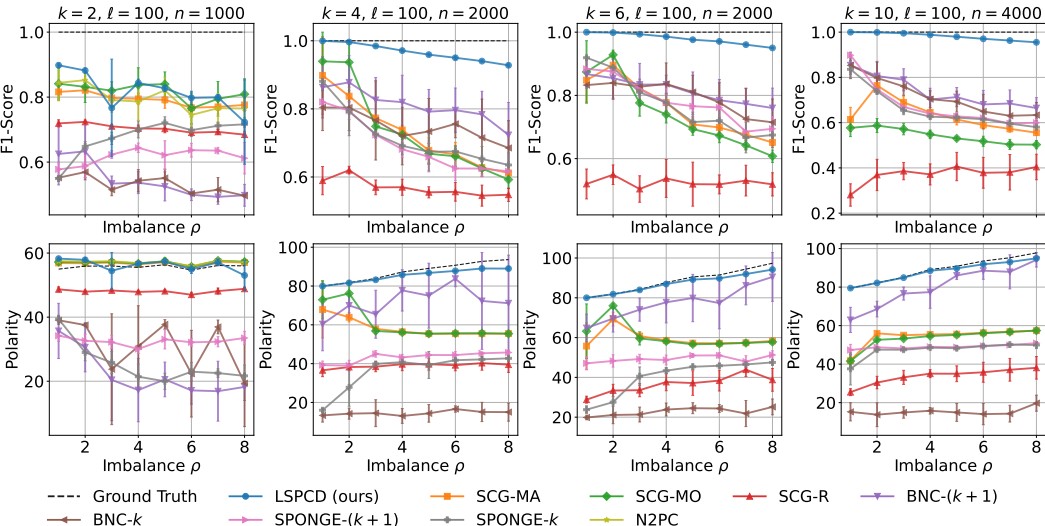

Figure 3: F1-score and polarity of different methods on synthetic graphs generated using the m-SSBM model, as the size ratio parameter $\rho$ varies. A larger value of $\rho$ means the ground-truth non-neutral clusters are more imbalanced. The noise level is fixed to $\eta = 0.4$.

with probability $\eta/2$. Conversely, edges between different clusters are negative with probability $1 - \eta$, and positive or absent with probability $\eta/2$. All other edges are positive or negative with equal probability $\min(\eta, 1/2)$.

Smaller values of $\eta$ correspond to denser non-neutral clusters and lower levels of noise. In other words, $\eta$ controls both sparsity of the graph and noise (i.e., flipping sign of edge weights).

### G.4 Results on Synthetic Datasets with Imbalanced Clusters

We now evaluate the performance of different methods on synthetic data with imbalanced cluster sizes. We use the same m-SSBM model described in the main text (and previous subsection). To control the degree of imbalance among the $k$ planted groups, we follow the approach of [23] and introduce a *group size ratio* parameter $\rho \geq 1$. When $\rho = 1$, each group has the same size $\ell$, resulting in a balanced partition. For $\rho > 1$, the group sizes follow a geometric progression: the smallest group has size $s$, the next has size $s \cdot \rho^{1/(k-1)}$, and so on, such that the largest group is $s \cdot \rho$. This construction ensures that the ratio between the largest and smallest group sizes is exactly $\rho$, while the total number of non-neutral nodes remains approximately $k\ell$. Any remaining nodes (i.e., when $n > \sum_{i=1}^{k} |C_i|$) are assigned as *neutral nodes*, and their edges are sampled from a neutral distribution. **In short**, a larger value of $\rho$ means more imbalanced ground-truth non-neutral clusters.

The results are presented in Figure 3. The noise level is fixed to $\eta = 0.4$. We observe that our method remains robust as the cluster size imbalance increases, while the performance of baseline methods deteriorates more rapidly. This suggests that our approach does not rely on strict balance assumptions and is capable of effectively recovering imbalanced ground-truth clusterings.

### G.5 Runtime Comparison on Large Scale Synthetic Data

To further highlight the scalability of our approach, we present additional experiments on large synthetic datasets generated using the m-SSBM model described in the paper, containing up to 250,000 objects and 150 million edges with non-zero edge weight. Our method is capable of handling even larger datasets; the only constraint was the memory required to store the matrix of pairwise relations on the machine used for our experiments, a limitation that could be easily overcome by using machines with larger memory capacities. We compare LSPCD with the strongest baseline, SCG with min-angle rounding (SCG-MA), and report the F1-score (ground-truth cluster recovery) and runtime in seconds. Other baselines yield lower F1-scores, and most are also slower in runtime. We fix $\eta = 0.45$, $k = 6$ and $\ell = 500$.

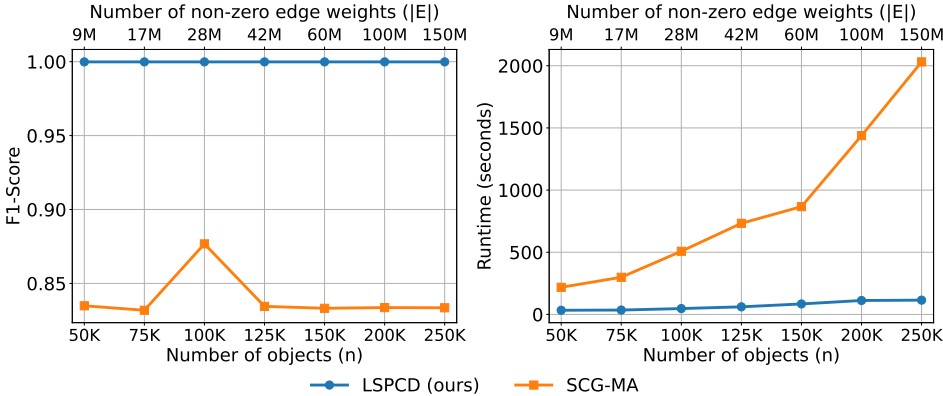

Figure 4: Runtime comparison on large-scale synthetic datasets generated using the m-SSBM model. LSPCD consistently achieves higher F1-scores than SCG-MA while requiring less runtime, demonstrating superior scalability and efficiency. We fixed $\eta = 0.45$, $k = 6$, and $\ell = 500$.

Table 3: The imbalance factor (IF) for three different values of $\xi = 1, 3, 4$.

| $k$ | Method | REF $\xi=1$ | $\xi=3$ | $\xi=4$ | SD $\xi=1$ | $\xi=3$ | $\xi=4$ | WikiC $\xi=1$ | $\xi=3$ | $\xi=4$ | EP $\xi=1$ | $\xi=3$ | $\xi=4$ | WikiP $\xi=1$ | $\xi=3$ | $\xi=4$ |
|---|---|---|---|---|---|---|---|---|---|---|---|---|---|---|---|---|
| 2 | LSPCD (ours) | 0.88 | 0.78 | 0.66 | 0.50 | 0.31 | 0.22 | 0.93 | 0.87 | 0.79 | 0.89 | 0.80 | 0.68 | 0.56 | 0.37 | 0.27 |
|   | SCG-MA | 0.05 | 0.02 | 0.01 | 0.06 | 0.02 | 0.01 | 0.77 | 0.62 | 0.48 | 0.13 | 0.05 | 0.04 | 0.06 | 0.02 | 0.01 |
|   | SCG-MO | 0.05 | 0.02 | 0.01 | 0.04 | 0.01 | 0.01 | 0.69 | 0.51 | 0.39 | 0.13 | 0.05 | 0.03 | 0.03 | 0.01 | 0.01 |
|   | SCG-B | 0.11 | 0.04 | 0.03 | 0.15 | 0.06 | 0.04 | 0.84 | 0.72 | 0.59 | 0.14 | 0.06 | 0.04 | 0.13 | 0.05 | 0.04 |
| 4 | LSPCD (ours) | 0.58 | 0.46 | 0.38 | 0.74 | 0.61 | 0.50 | 0.81 | 0.65 | 0.50 | 0.78 | 0.65 | 0.54 | 0.55 | 0.34 | 0.24 |
|   | SCG-MA | 0.75 | 0.71 | 0.66 | 0.51 | 0.31 | 0.22 | 0.18 | 0.08 | 0.05 | 0.50 | 0.36 | 0.27 | 0.64 | 0.55 | 0.49 |
|   | SCG-MO | 0.75 | 0.72 | 0.68 | 0.41 | 0.25 | 0.17 | 0.50 | 0.30 | 0.21 | 0.54 | 0.40 | 0.30 | 0.46 | 0.35 | 0.28 |
|   | SCG-B | 0.00 | 0.00 | 0.00 | 0.55 | 0.43 | 0.35 | 0.95 | 0.90 | 0.82 | 0.68 | 0.60 | 0.51 | 0.34 | 0.25 | 0.19 |
| 6 | LSPCD (ours) | 0.49 | 0.37 | 0.31 | 0.66 | 0.50 | 0.40 | 0.80 | 0.62 | 0.48 | 0.71 | 0.55 | 0.43 | 0.76 | 0.61 | 0.49 |
|   | SCG-MA | 0.71 | 0.66 | 0.60 | 0.35 | 0.21 | 0.15 | 0.22 | 0.09 | 0.06 | 0.63 | 0.55 | 0.50 | 0.53 | 0.45 | 0.40 |
|   | SCG-MO | 0.78 | 0.74 | 0.71 | 0.56 | 0.37 | 0.26 | 0.50 | 0.28 | 0.19 | 0.42 | 0.31 | 0.24 | 0.36 | 0.27 | 0.22 |
|   | SCG-B | 0.00 | 0.00 | 0.00 | 0.45 | 0.36 | 0.29 | 0.83 | 0.76 | 0.67 | 0.53 | 0.46 | 0.39 | 0.26 | 0.19 | 0.14 |

The results are shown in Figure 4. Our method consistently outperforms SCG in terms of F1 score while also being more efficient, demonstrating its ability to scale to larger graphs.

### G.6 Impact of $\xi$ on Imbalance Factor

Evaluating unsupervised learning methods in the absence of ground truth is inherently challenging, and there is often no universally or uniquely accepted metric or criterion. As a result, it is common practice to report multiple metrics, each reflecting a different objective or perspective. In this study, for real-world datasets, we report both polarity and imbalance factor. As shown in our extensive experiments, our method is the only one that consistently performs well w.r.t. both metrics across various datasets.

Given the difficulty of evaluation in such settings, no single fixed value of $\xi$ can be considered optimal. In the absence of prior preference, a reasonable approach is to examine multiple values of $\xi$ and analyze how different methods perform under each. We experimented with a range of values and found that the conclusions remain stable. As mentioned in the main paper, we chose $\xi = 3$ since it provides a sharper distinction between solutions with different degrees of balance, compared to $\xi = 1$, which would have been the natural choice otherwise as it corresponds to Shannon entropy. To further demonstrate robustness, we now also report results for $\xi = 1, 3, 4$ for a subset of methods (see Table 3). Across all tested values, the relative ranking of methods remains consistent, and our conclusions are unchanged (again when SCG is more balanced, it usually yields low polarity). Results are consistent for even larger values of $\xi$.

### G.7 Aspects to Assess Solution Quality

Below, we present 11 evaluation criteria used in our experiments on real-world data to shed light on how the clustering solutions generated by each method differ. In the context of PCD (where ground-truth solutions are not available), evaluating clustering quality is inherently subjective: each aspect listed below highlights a distinct facet of the solution. Generally, there is a trade-off among these aspects, and the objective is to achieve a good balance between them. Below, we define the aspects used in our analysis. We have defined these aspects such that a larger number is better (apart from runtime).

Let $N = \sum_{m \in [k]} |S_m|$ denote the number of non-neutral objects, and let $N_{\text{nz}} = N_{\text{intra}}^+ + N_{\text{intra}}^- + N_{\text{inter}}^- + N_{\text{inter}}^+$ represent the number of non-zero similarities between non-neutral objects.

- SIZE $= N$: The total number of non-neutral objects.
- IF: The imbalance factor introduced in the main text.
- POL: *Polarity*, as defined in the main paper (Eq. 2).
- K: The number of non-empty non-neutral clusters.
- MAC: *Mean Average Cohesion*, quantifying the density of positive intra-cluster similarities, defined as
$$\text{MAC} = \frac{1}{k} \sum_{m \in [k]} \frac{1}{|S_m|(|S_m| - 1)} \sum_{i,j \in S_m} A_{i,j}^+.$$
  Its range is $[0, 1]$, where higher values indicate stronger cohesion within clusters.
- MAO: *Mean Average Opposition*, measuring the density of negative inter-cluster similarities, defined as
$$\text{MAO} = \frac{1}{k(k-1)} \sum_{\substack{m \in [k] \\ p \in [k] \setminus \{m\}}} \frac{1}{|S_m||S_p|} \sum_{\substack{i \in S_m \\ j \in S_p}} A_{i,j}^-.$$
  Its range is $[0, 1]$, where higher values indicate stronger opposition between clusters.
- CC+: Measures the fraction of intra-cluster similarities that are positive minus those that are negative, defined as
$$\text{CC+} = \frac{N_{\text{intra}}^+ - N_{\text{intra}}^-}{N_{\text{intra}}^+ + N_{\text{intra}}^-}.$$
  Its range is $[-1, 1]$, where $-1$ indicates that all non-zero intra-cluster similarities are negative, and $+1$ indicates that all are positive.
- CC–: Measures the fraction of inter-cluster similarities that are negative minus those that are positive, defined as
$$\text{CC-} = \frac{N_{\text{inter}}^- - N_{\text{inter}}^+}{N_{\text{inter}}^- + N_{\text{inter}}^+}.$$
  Its range is $[-1, 1]$, where $-1$ indicates that all non-zero inter-cluster similarities are positive, and $+1$ indicates that all are negative.
- DENS: The proportion of non-zero similarities among non-neutral objects, defined as
$$\text{DENS} = \frac{N_{\text{nz}}}{N(N-1)}.$$
  Its range is $[0, 1]$, with higher values indicating denser connectivity.
- ISO: *Isolation*, measuring the separation between non-neutral and neutral objects, defined as
$$\text{ISO} = \frac{N_{\text{nz}}}{N_{\text{nz}} + \sum_{i \in S_0} \sum_{j \notin S_0} |A_{i,j}|}.$$
  Its range is $[0, 1]$, where ISO $= 1$ means non-neutral objects are fully isolated from neutral ones, meaning no non-zero edges exist between them (which is ideal).
- TIME (s): Runtime of the corresponding method in seconds.

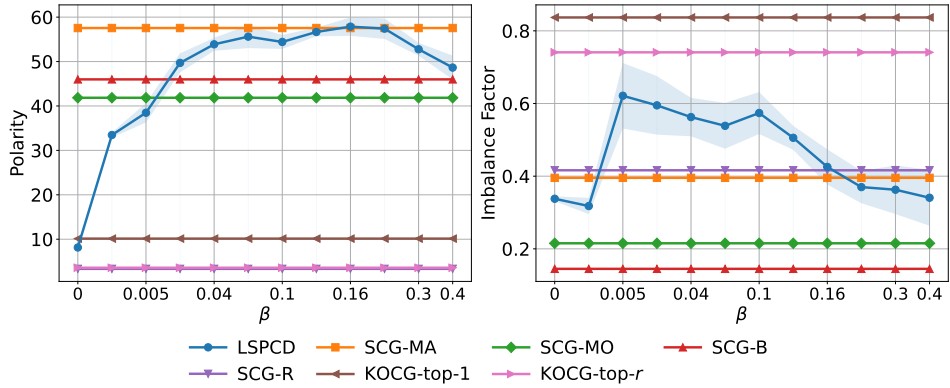

Figure 5: Impact of $\beta$ on the WikiPol dataset with $k = 6$.

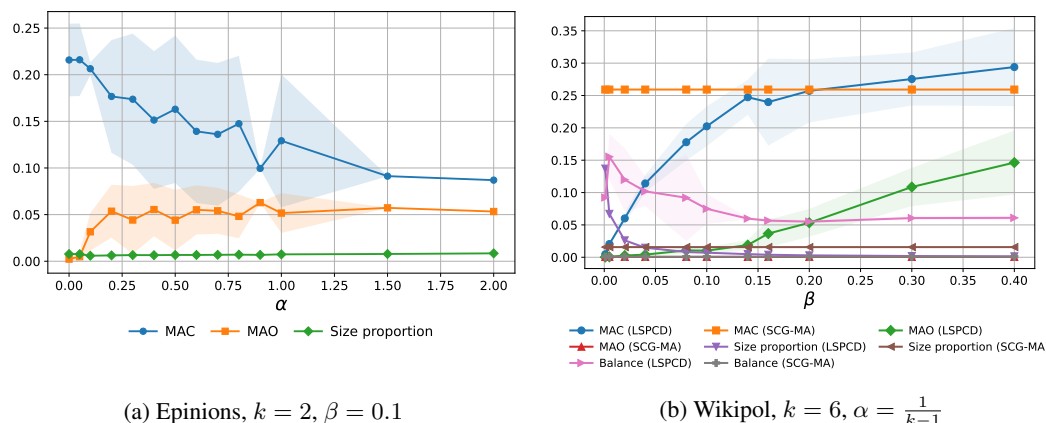

(a) Epinions, $k = 2$, $\beta = 0.1$

(b) Wikipol, $k = 6$, $\alpha = \frac{1}{k-1}$

Figure 6: Investigation of the impact of $\alpha$ and $\beta$ for LSPCD (Alg. 3).

### G.8 Varying $\alpha$ and $\beta$

Figure 5 shows the effect of varying $\beta$. Very small or large $\beta$ values lead to poorer polarity, as they produce clustering solutions with too many or too few non-neutral objects, respectively. In contrast, intermediate $\beta$ values consistently yields competitive polarity with the best performing methods, while being more balanced.

In Figure 6, we illustrate the impact of varying $\alpha$ and $\beta$. According to Figure 6a, increasing $\alpha$ naturally balances intra-cluster cohesion and inter-cluster opposition. The size proportion, defined as the fraction of non-neutral objects in $V$, remains constant as $\alpha$ varies. Figure 6b shows that increasing $\beta$ monotonically reduces the number of non-neutral objects, leading to denser clusters, as indicated by improved MAC and MAO scores. Notably, balance remains stable across different $\beta$ values, unlike the baseline SCG-MA.

### G.9 Results on Real-World Datasets

Tables 4-11 present detailed analyses for all datasets. See Appendix G.7 for a description of the 11 aspects reported. Firstly, we observe that our method exhibits low standard deviation, indicating robustness to the initial random solution. It consistently finds high-polarity solutions while maintaining better balance than its main competitor, SCG. Unlike some baselines such as KOCG and SPONGE, our method does not enforce excessive balance, ensuring solutions remain both high in polarity and reasonably balanced.

In terms of runtime, our method is efficient and competitive with the baselines. It is also consistently ranked among the best in both DENS and ISO. Unlike SCG, our method always identifies $k$ non-empty non-neutral clusters, which we argue is a significant limitation of SCG.

While SCG generally achieves higher MAC values, this is largely due to its tendency to produce highly imbalanced solutions, often with singleton clusters. Since small or singleton clusters trivially yield high average cohesion (values close to 1), they can disproportionately inflate the overall MAC score.

Finally, our method performs comparably or better than SCG in CC+ and significantly outperforms it in CC- in most cases. This highlights another limitation of SCG: it often includes more positive similarities between clusters than negative ones, as reflected by negative CC- values. Some baselines produce either overly large or overly small non-neutral clusters (based on SIZE), whereas our method consistently finds solutions with a reasonable number of non-neutral objects (which consequently leads to a good balance of the other aspects), similar to SCG. However, we note that we can easily adjust the number of non-neutral objects by adjusting $\beta$, as discussed in the paper.

Table 4: Detailed results for the **WoW-EP8** dataset. LSPCD (avg) and LSPCD (std) respectively indicate the mean and standard deviation across five runs of our method with different seeds.

| $k$ | | SIZE | IF | POL | K | MAC | MAO | CC+ | CC- | DENS | ISO | TIME (s) |
|---|---|---|---|---|---|---|---|---|---|---|---|---|
| 2 | LSPCD (AVG) | 586 | 0.176 | 223.406 | 2 | 0.261 | 0.098 | 0.757 | 0.509 | 0.511 | 0.769 | 0.249 |
| | LSPCD (STD) | 0 | 0.0 | 0.0 | 0 | 0.0 | 0.0 | 0.0 | 0.0 | 0.0 | 0.0 | 0.034 |
| | N2PC ($\gamma = 1$) | 527 | 0.0 | 236.626 | 1 | 0.519 | 0.0 | 0.765 | 0.0 | 0.588 | 0.723 | 26.916 |
| | N2PC ($\gamma = 1.2$) | 519 | 0.004 | 236.543 | 2 | 0.763 | 0.151 | 0.77 | 1.0 | 0.593 | 0.709 | 28.426 |
| | N2PC ($\gamma = 1.5$) | 535 | 0.053 | 233.701 | 2 | 0.268 | 0.132 | 0.767 | 0.715 | 0.571 | 0.724 | 27.133 |
| | N2PC ($\gamma = 1.7$) | 552 | 0.129 | 227.609 | 2 | 0.267 | 0.109 | 0.767 | 0.584 | 0.542 | 0.73 | 33.419 |
| | N2PC ($\gamma = 2.0$) | 571 | 0.235 | 217.398 | 2 | 0.27 | 0.082 | 0.769 | 0.48 | 0.504 | 0.726 | 26.603 |
| | SCG-MA | 527 | 0.0 | 236.55 | 1 | 0.52 | 0.0 | 0.762 | 0.0 | 0.59 | 0.725 | 1.025 |
| | SCG-MO | 517 | 0.0 | 236.592 | 1 | 0.527 | 0.0 | 0.769 | 0.0 | 0.596 | 0.708 | 1.026 |
| | SCG-B | 583 | 0.049 | 200.604 | 2 | 0.274 | 0.094 | 0.697 | -0.369 | 0.514 | 0.767 | 4.657 |
| | SCG-R | 513 | 0.03 | 214.616 | 2 | 0.272 | 0.104 | 0.761 | 0.451 | 0.552 | 0.654 | 0.517 |
| | KOCG-TOP-1 | 16 | 0.967 | 13.0 | 2 | 0.986 | 0.889 | 0.965 | 0.778 | 1.0 | 0.019 | — |
| | KOCG-TOP-$r$ | 527 | 0.989 | 12.964 | 2 | 0.467 | 0.11 | 0.658 | -0.598 | 0.559 | 0.691 | — |
| | BNC-($k + 1$) | 3 | 0.792 | -0.667 | 2 | 0.5 | 0.0 | -1.0 | 0.0 | 0.333 | 0.007 | 0.99 |
| | BNC-$k$ | 790 | 0.005 | 184.63 | 2 | 0.152 | 0.049 | 0.628 | 1.0 | 0.372 | 1.0 | 0.537 |
| | SPONGE-($k + 1$) | 375 | 0.425 | 87.957 | 2 | 0.204 | 0.095 | 0.738 | 0.318 | 0.343 | 0.346 | 0.581 |
| | SPONGE-$k$ | 790 | 0.236 | 191.38 | 2 | 0.191 | 0.093 | 0.696 | 0.15 | 0.372 | 1.0 | 0.572 |
| 4 | LSPCD (AVG) | 590 | 0.095 | 218.458 | 4 | 0.156 | 0.082 | 0.76 | 0.426 | 0.506 | 0.772 | 0.317 |
| | LSPCD (STD) | 1 | 0.001 | 0.142 | 0 | 0.003 | 0.002 | 0.001 | 0.016 | 0.0 | 0.002 | 0.044 |
| | SCG-MA | 599 | 0.138 | 205.137 | 4 | 0.58 | 0.13 | 0.762 | -0.32 | 0.527 | 0.822 | 1.198 |
| | SCG-MO | 568 | 0.102 | 213.211 | 4 | 0.827 | 0.141 | 0.77 | -0.349 | 0.55 | 0.776 | 1.176 |
| | SCG-B | 615 | 0.0 | 211.561 | 1 | 0.421 | 0.0 | 0.693 | 0.0 | 0.498 | 0.822 | 6.233 |
| | SCG-R | 503 | 0.02 | 214.623 | 4 | 0.211 | 0.056 | 0.762 | 0.747 | 0.564 | 0.644 | 1.131 |
| | KOCG-TOP-1 | 31 | 0.962 | 9.054 | 4 | 0.962 | 0.668 | 0.944 | 0.339 | 0.978 | 0.039 | — |
| | KOCG-TOP-$r$ | 599 | 0.987 | 7.393 | 4 | 0.436 | 0.099 | 0.692 | -0.618 | 0.516 | 0.806 | — |
| | BNC-($k + 1$) | 8 | 0.698 | -0.25 | 4 | 0.5 | 0.0 | -1.0 | 0.0 | 0.036 | 0.003 | 0.62 |
| | BNC-$k$ | 790 | 0.011 | 185.305 | 4 | 0.327 | 0.03 | 0.632 | 1.0 | 0.372 | 1.0 | 0.594 |
| | SPONGE-($k + 1$) | 485 | 0.929 | 53.823 | 4 | 0.448 | 0.06 | 0.886 | -0.53 | 0.33 | 0.433 | 0.593 |
| | SPONGE-$k$ | 790 | 0.869 | 71.162 | 4 | 0.383 | 0.08 | 0.803 | -0.499 | 0.372 | 1.0 | 0.61 |
| 6 | LSPCD (AVG) | 591 | 0.075 | 217.344 | 6 | 0.142 | 0.071 | 0.761 | 0.414 | 0.505 | 0.773 | 0.269 |
| | LSPCD (STD) | 0 | 0.001 | 0.142 | 0 | 0.006 | 0.002 | 0.0 | 0.009 | 0.0 | 0.001 | 0.015 |
| | SCG-MA | 598 | 0.106 | 207.299 | 6 | 0.785 | 0.113 | 0.763 | -0.339 | 0.527 | 0.819 | 1.226 |
| | SCG-MO | 591 | 0.112 | 205.796 | 6 | 0.756 | 0.171 | 0.77 | -0.321 | 0.534 | 0.811 | 1.378 |
| | SCG-B | 615 | 0.0 | 211.561 | 1 | 0.421 | 0.0 | 0.693 | 0.0 | 0.498 | 0.822 | 7.512 |
| | SCG-R | 744 | 0.016 | 201.172 | 5 | 0.323 | 0.178 | 0.669 | 0.313 | 0.41 | 0.978 | 1.295 |
| | KOCG-TOP-1 | 42 | 0.994 | 7.905 | 6 | 0.992 | 0.605 | 0.984 | 0.303 | 0.934 | 0.053 | — |
| | KOCG-TOP-$r$ | 598 | 0.976 | 9.109 | 6 | 0.472 | 0.095 | 0.73 | -0.637 | 0.522 | 0.812 | — |
| | BNC-($k + 1$) | 10 | 0.859 | -0.2 | 6 | 0.5 | 0.0 | -1.0 | 0.0 | 0.022 | 0.002 | 0.977 |
| | BNC-$k$ | 790 | 0.009 | 185.198 | 6 | 0.552 | 0.016 | 0.632 | 1.0 | 0.372 | 1.0 | 0.61 |
| | SPONGE-($k + 1$) | 572 | 0.899 | 47.762 | 6 | 0.537 | 0.065 | 0.893 | -0.511 | 0.326 | 0.536 | 0.607 |
| | SPONGE-$k$ | 790 | 0.878 | 57.868 | 6 | 0.53 | 0.075 | 0.834 | -0.529 | 0.372 | 1.0 | 0.599 |

Table 5: Detailed results for the **Bitcoin** dataset. LSPCD (avg) and LSPCD (std) respectively indicate the mean and standard deviation across five runs of our method with different seeds.

| $k$ | | SIZE | IF | POL | K | MAC | MAO | CC+ | CC- | DENS | ISO | TIME (s) |
|---|---|---|---|---|---|---|---|---|---|---|---|---|
| 2 | LSPCD (AVG) | 155 | 0.648 | 29.022 | 2 | 0.2 | 0.143 | 0.94 | 0.969 | 0.199 | 0.211 | 2.203 |
| | LSPCD (STD) | 0 | 0.0 | 0.013 | 0 | 0.0 | 0.0 | 0.003 | 0.001 | 0.001 | 0.001 | 0.161 |
| | N2PC ($\gamma = 1$) | 134 | 0.016 | 29.642 | 2 | 0.617 | 0.195 | 0.909 | 0.733 | 0.246 | 0.184 | 11.318 |
| | N2PC ($\gamma = 1.2$) | 164 | 0.46 | 30.146 | 2 | 0.238 | 0.141 | 0.91 | 0.964 | 0.201 | 0.222 | 12.26 |
| | N2PC ($\gamma = 1.5$) | 64 | 1.0 | 24.375 | 2 | 0.263 | 0.517 | 0.941 | 0.992 | 0.397 | 0.133 | 12.818 |
| | N2PC ($\gamma = 1.7$) | 70 | 1.0 | 23.857 | 2 | 0.255 | 0.452 | 0.876 | 0.993 | 0.364 | 0.136 | 13.461 |
| | N2PC ($\gamma = 2.0$) | 68 | 1.0 | 24.147 | 2 | 0.258 | 0.478 | 0.871 | 0.993 | 0.38 | 0.139 | 12.834 |
| | SCG-MA | 179 | 0.163 | 28.838 | 2 | 0.298 | 0.068 | 0.906 | 0.873 | 0.179 | 0.216 | 0.122 |
| | SCG-MO | 138 | 0.032 | 29.522 | 2 | 0.114 | 0.147 | 0.91 | 0.778 | 0.237 | 0.184 | 0.123 |
| | SCG-B | 40 | 0.995 | 21.65 | 2 | 0.248 | 0.87 | 0.956 | 1.0 | 0.56 | 0.201 | 2.243 |
| | SCG-R | 842 | 0.249 | 14.24 | 2 | 0.022 | 0.009 | 0.908 | 0.812 | 0.019 | 0.394 | 1.024 |
| | KOCG-TOP-1 | 2 | 1.0 | 1.0 | 2 | 1.0 | 1.0 | 0.0 | 1.0 | 1.0 | 1.0 | 0.004 |
| | KOCG-TOP-$r$ | 179 | 0.992 | 3.754 | 2 | 0.063 | 0.055 | 0.266 | 0.182 | 0.094 | 0.165 | — |
| | BNC-($k + 1$) | 50 | 0.134 | -10.76 | 2 | 0.058 | 0.0 | -0.516 | 0.0 | 0.425 | 0.581 | 0.341 |
| | BNC-$k$ | 5881 | 0.017 | 5.268 | 2 | 0.059 | 0.001 | 0.721 | 0.694 | 0.001 | 1.0 | 0.202 |
| | SPONGE-($k + 1$) | 6 | 0.792 | 1.0 | 2 | 0.667 | 0.0 | 1.0 | 0.0 | 0.2 | 1.0 | 1.116 |
| | SPONGE-$k$ | 5881 | 0.001 | 5.092 | 2 | 0.501 | 0.0 | 0.697 | 0.0 | 0.001 | 1.0 | 2.138 |
| 4 | LSPCD (AVG) | 217 | 0.47 | 23.333 | 4 | 0.182 | 0.12 | 0.929 | 0.815 | 0.143 | 0.256 | 2.117 |
| | LSPCD (STD) | 13 | 0.064 | 0.765 | 0 | 0.027 | 0.029 | 0.004 | 0.123 | 0.011 | 0.009 | 0.408 |
| | SCG-MA | 176 | 0.223 | 25.121 | 4 | 0.488 | 0.07 | 0.914 | -0.44 | 0.172 | 0.2 | 0.302 |
| | SCG-MO | 180 | 0.218 | 25.252 | 4 | 0.487 | 0.07 | 0.91 | -0.431 | 0.169 | 0.205 | 0.348 |
| | SCG-B | 216 | 0.233 | 12.401 | 4 | 0.473 | 0.052 | 0.865 | 0.296 | 0.076 | 0.176 | 6.377 |
| | SCG-R | 450 | 0.518 | 8.033 | 4 | 0.036 | 0.008 | 0.92 | -0.627 | 0.033 | 0.238 | 1.237 |
| | KOCG-TOP-1 | 26 | 0.905 | 8.41 | 4 | 0.859 | 0.621 | 1.0 | 0.653 | 0.738 | 0.112 | — |
| | KOCG-TOP-$r$ | 176 | 0.931 | 5.034 | 4 | 0.136 | 0.041 | 0.856 | -0.246 | 0.113 | 0.157 | — |
| | BNC-($k + 1$) | 58 | 0.227 | -9.414 | 4 | 0.112 | 0.0 | -0.516 | 0.0 | 0.32 | 0.576 | 0.185 |
| | BNC-$k$ | 5881 | 0.01 | 5.208 | 4 | 0.029 | 0.0 | 0.721 | 0.67 | 0.001 | 1.0 | 0.178 |
| | SPONGE-($k + 1$) | 71 | 0.096 | 1.099 | 4 | 0.754 | 0.0 | 1.0 | 0.0 | 0.016 | 0.443 | 3.364 |
| | SPONGE-$k$ | 5881 | 0.001 | 5.092 | 4 | 0.75 | 0.0 | 0.697 | 0.0 | 0.001 | 1.0 | 2.797 |
| 6 | LSPCD (AVG) | 194 | 0.494 | 20.031 | 6 | 0.251 | 0.143 | 0.948 | 0.646 | 0.155 | 0.231 | 2.73 |
| | LSPCD (STD) | 23 | 0.153 | 1.827 | 0 | 0.046 | 0.053 | 0.007 | 0.304 | 0.019 | 0.015 | 0.468 |
| | SCG-MA | 430 | 0.457 | 14.568 | 6 | 0.536 | 0.021 | 0.931 | -0.355 | 0.055 | 0.301 | 0.448 |
| | SCG-MO | 412 | 0.464 | 15.165 | 6 | 0.571 | 0.028 | 0.929 | -0.337 | 0.058 | 0.3 | 0.477 |
| | SCG-B | 326 | 0.472 | 9.321 | 6 | 0.313 | 0.009 | 0.866 | -0.421 | 0.053 | 0.222 | 10.509 |
| | SCG-R | 860 | 0.407 | 6.861 | 6 | 0.038 | 0.006 | 0.941 | -0.529 | 0.017 | 0.367 | 2.125 |
| | KOCG-TOP-1 | 28 | 0.921 | 4.071 | 6 | 0.867 | 0.338 | 1.0 | 0.197 | 0.537 | 0.055 | — |
| | KOCG-TOP-$r$ | 430 | 0.867 | 3.601 | 6 | 0.077 | 0.013 | 0.88 | -0.405 | 0.043 | 0.26 | — |
| | BNC-($k + 1$) | 224 | 0.255 | -4.239 | 6 | 0.075 | 0.001 | -0.622 | 0.958 | 0.033 | 0.394 | 0.286 |
| | BNC-$k$ | 5881 | 0.009 | 5.197 | 6 | 0.075 | 0.0 | 0.722 | 0.657 | 0.001 | 1.0 | 0.194 |
| | SPONGE-($k + 1$) | 222 | 0.147 | 1.252 | 6 | 0.622 | 0.0 | 1.0 | 0.0 | 0.006 | 0.401 | 1.959 |
| | SPONGE-$k$ | 5881 | 0.005 | 5.085 | 6 | 0.563 | 0.0 | 0.696 | -1.0 | 0.001 | 1.0 | 2.473 |

Table 6: Detailed results for the **WikiVot** dataset. LSPCD (avg) and LSPCD (std) respectively indicate the mean and standard deviation across five runs of our method with different seeds.

| $k$ | | SIZE | IF | POL | K | MAC | MAO | CC+ | CC- | DENS | ISO | TIME (s) |
|---|---|---|---|---|---|---|---|---|---|---|---|---|
| 2 | LSPCD (AVG) | 1278 | 0.428 | 62.322 | 2 | 0.038 | 0.015 | 0.831 | 0.673 | 0.06 | 0.548 | 2.457 |
| | LSPCD (STD) | 4 | 0.002 | 0.003 | 0 | 0.0 | 0.0 | 0.0 | 0.003 | 0.0 | 0.001 | 0.276 |
| | N2PC ($\gamma = 1$) | 712 | 0.0 | 71.635 | 1 | 0.11 | 0.0 | 0.852 | 0.0 | 0.118 | 0.399 | 24.313 |
| | N2PC ($\gamma = 1.2$) | 759 | 0.006 | 71.663 | 2 | 0.052 | 0.093 | 0.847 | 0.785 | 0.112 | 0.418 | 24.251 |
| | N2PC ($\gamma = 1.5$) | 760 | 0.096 | 70.016 | 2 | 0.054 | 0.029 | 0.852 | 0.753 | 0.109 | 0.412 | 22.745 |
| | N2PC ($\gamma = 1.7$) | 923 | 0.562 | 59.142 | 2 | 0.056 | 0.017 | 0.844 | 0.685 | 0.077 | 0.426 | 23.51 |
| | N2PC ($\gamma = 2.0$) | 1190 | 1.0 | 40.509 | 2 | 0.064 | 0.012 | 0.849 | 0.651 | 0.042 | 0.395 | 39.069 |
| | SCG-MA | 813 | 0.008 | 71.476 | 2 | 0.048 | 0.079 | 0.846 | 0.671 | 0.104 | 0.436 | 1.537 |
| | SCG-MO | 748 | 0.009 | 71.733 | 2 | 0.052 | 0.082 | 0.854 | 0.671 | 0.113 | 0.411 | 1.432 |
| | SCG-B | 414 | 0.037 | 37.589 | 2 | 0.054 | 0.033 | 0.756 | 0.776 | 0.12 | 0.221 | 7.501 |
| | SCG-R | 1100 | 0.174 | 54.693 | 2 | 0.032 | 0.013 | 0.83 | 0.618 | 0.061 | 0.444 | 0.781 |
| | KOCG-TOP-1 | 10 | 0.717 | 7.6 | 2 | 0.905 | 0.857 | 1.0 | 1.0 | 0.844 | 0.012 | — |
| | KOCG-TOP-$r$ | 813 | 0.999 | 2.312 | 2 | 0.047 | 0.022 | 0.427 | -0.337 | 0.066 | 0.297 | — |
| | BNC-$(k+1)$ | 9 | 0.792 | -1.111 | 2 | 0.0 | 0.0 | 0.0 | 0.0 | 0.139 | 1.0 | 0.721 |
| | BNC-$k$ | 7115 | 0.003 | 15.794 | 2 | 0.002 | 0.0 | 0.558 | 0.0 | 0.004 | 1.0 | 0.49 |
| | SPONGE-$(k+1)$ | 10 | 0.472 | 1.0 | 2 | 0.571 | 0.0 | 1.0 | 0.0 | 0.111 | 1.0 | 1.602 |
| | SPONGE-$k$ | 7115 | 0.003 | 15.794 | 2 | 0.057 | 0.0 | 0.558 | 0.0 | 0.004 | 1.0 | 0.977 |
| 4 | LSPCD (AVG) | 1089 | 0.519 | 52.605 | 4 | 0.073 | 0.013 | 0.856 | -0.045 | 0.072 | 0.489 | 4.381 |
| | LSPCD (STD) | 149 | 0.229 | 6.003 | 0 | 0.026 | 0.002 | 0.004 | 0.427 | 0.011 | 0.04 | 1.69 |
| | SCG-MA | 1142 | 0.361 | 52.945 | 4 | 0.081 | 0.018 | 0.849 | -0.618 | 0.069 | 0.506 | 2.042 |
| | SCG-MO | 1059 | 0.374 | 53.07 | 4 | 0.089 | 0.022 | 0.858 | -0.692 | 0.073 | 0.474 | 1.986 |
| | SCG-B | 790 | 0.598 | 24.782 | 4 | 0.091 | 0.014 | 0.774 | -0.718 | 0.077 | 0.342 | 18.286 |
| | SCG-R | 1524 | 0.437 | 19.524 | 4 | 0.031 | 0.008 | 0.813 | -0.68 | 0.043 | 0.549 | 3.074 |
| | KOCG-TOP-1 | 33 | 0.811 | 4.525 | 4 | 0.845 | 0.086 | 0.933 | -0.609 | 0.576 | 0.03 | — |
| | KOCG-TOP-$r$ | 1142 | 0.99 | 3.288 | 4 | 0.055 | 0.011 | 0.719 | -0.618 | 0.059 | 0.44 | — |
| | BNC-$(k+1)$ | 15 | 0.651 | -1.067 | 4 | 0.0 | 0.0 | -1.0 | 0.0 | 0.076 | 1.0 | 0.527 |
| | BNC-$k$ | 7115 | 0.001 | 15.794 | 4 | 0.001 | 0.0 | 0.558 | 0.0 | 0.004 | 1.0 | 0.533 |
| | SPONGE-$(k+1)$ | 12 | 0.712 | 1.0 | 4 | 0.8 | 0.0 | 1.0 | 0.0 | 0.091 | 1.0 | 2.327 |
| | SPONGE-$k$ | 7115 | 0.003 | 15.794 | 4 | 0.156 | 0.0 | 0.558 | 0.0 | 0.004 | 1.0 | 1.522 |
| 6 | LSPCD (AVG) | 534 | 0.563 | 46.179 | 6 | 0.143 | 0.029 | 0.896 | -0.314 | 0.133 | 0.287 | 5.292 |
| | LSPCD (STD) | 46 | 0.08 | 2.177 | 0 | 0.015 | 0.002 | 0.004 | 0.07 | 0.012 | 0.008 | 0.931 |
| | SCG-MA | 1355 | 0.421 | 45.494 | 6 | 0.064 | 0.023 | 0.849 | -0.647 | 0.056 | 0.564 | 2.24 |
| | SCG-MO | 1226 | 0.409 | 47.013 | 6 | 0.073 | 0.024 | 0.859 | -0.683 | 0.063 | 0.526 | 2.178 |
| | SCG-B | 941 | 0.605 | 23.332 | 6 | 0.121 | 0.018 | 0.78 | -0.735 | 0.065 | 0.369 | 29.072 |
| | SCG-R | 1501 | 0.786 | 10.433 | 6 | 0.039 | 0.008 | 0.817 | -0.734 | 0.044 | 0.542 | 3.475 |
| | KOCG-TOP-1 | 40 | 0.963 | 4.52 | 6 | 0.894 | 0.227 | 0.981 | -0.188 | 0.564 | 0.033 | — |
| | KOCG-TOP-$r$ | 1355 | 0.962 | 3.132 | 6 | 0.051 | 0.009 | 0.73 | -0.62 | 0.05 | 0.506 | — |
| | BNC-$(k+1)$ | 13 | 0.974 | -1.077 | 6 | 0.0 | 0.0 | -1.0 | 0.0 | 0.09 | 1.0 | 0.966 |
| | BNC-$k$ | 7115 | 0.002 | 15.794 | 6 | 0.001 | 0.0 | 0.558 | 0.0 | 0.004 | 1.0 | 0.546 |
| | SPONGE-$(k+1)$ | 20 | 0.859 | 1.0 | 6 | 0.644 | 0.0 | 1.0 | 0.0 | 0.053 | 1.0 | 1.791 |
| | SPONGE-$k$ | 7115 | 0.003 | 15.794 | 6 | 0.434 | 0.0 | 0.558 | 0.0 | 0.004 | 1.0 | 1.66 |

Table 7: Detailed results for the **Referendum** dataset. LSPCD (avg) and LSPCD (std) respectively indicate the mean and standard deviation across five runs of our method with different seeds.

| $k$ | | SIZE | IF | POL | K | MAC | MAO | CC+ | CC- | DENS | ISO | TIME (s) |
|---|---|---|---|---|---|---|---|---|---|---|---|---|
| 2 | LSPCD (AVG) | 915 | 0.71 | 146.109 | 2 | 0.279 | 0.014 | 1.0 | 0.114 | 0.17 | 0.353 | 3.376 |
| | LSPCD (STD) | 1 | 0.002 | 0.092 | 0 | 0.001 | 0.0 | 0.0 | 0.017 | 0.001 | 0.001 | 0.19 |
| | N2PC ($\gamma = 1$) | 692 | 0.013 | 173.604 | 2 | 0.542 | 0.297 | 1.0 | 0.573 | 0.253 | 0.359 | 62.32 |
| | N2PC ($\gamma = 1.2$) | 651 | 0.02 | 173.634 | 2 | 0.401 | 0.254 | 1.0 | 0.571 | 0.27 | 0.343 | 61.685 |
| | N2PC ($\gamma = 1.5$) | 918 | 0.944 | 130.261 | 2 | 0.253 | 0.011 | 1.0 | 0.24 | 0.149 | 0.331 | 70.204 |
| | N2PC ($\gamma = 1.7$) | 976 | 1.0 | 119.4 | 2 | 0.241 | 0.01 | 1.0 | 0.209 | 0.129 | 0.326 | 77.728 |
| | N2PC ($\gamma = 2.0$) | 992 | 1.0 | 118.099 | 2 | 0.236 | 0.009 | 1.0 | 0.169 | 0.126 | 0.327 | 80.728 |
| | SCG-MA | 824 | 0.013 | 172.206 | 2 | 0.455 | 0.247 | 1.0 | 0.558 | 0.211 | 0.409 | 1.863 |
| | SCG-MO | 673 | 0.013 | 174.083 | 2 | 0.546 | 0.3 | 1.0 | 0.571 | 0.261 | 0.352 | 1.108 |
| | SCG-B | 1158 | 0.03 | 116.252 | 2 | 0.176 | 0.068 | 1.0 | 0.58 | 0.101 | 0.396 | 23.313 |
| | SCG-R | 1550 | 0.037 | 120.85 | 2 | 0.095 | 0.04 | 1.0 | 0.529 | 0.079 | 0.492 | 4.657 |
| | KOCG-TOP-1 | 15 | 0.637 | 11.6 | 2 | 0.973 | 0.659 | 1.0 | 1.0 | 0.829 | 0.007 | — |
| | KOCG-TOP-$r$ | 824 | 0.961 | 15.425 | 2 | 0.057 | 0.018 | 0.705 | -0.317 | 0.065 | 0.169 | — |
| | BNC-$(k+1)$ | 4 | 1.0 | -1.0 | 2 | 0.0 | 0.0 | -1.0 | 0.0 | 0.333 | 0.286 | 1.929 |
| | BNC-$k$ | 10884 | 0.0 | 41.495 | 2 | 0.002 | 0.0 | 0.898 | -1.0 | 0.004 | 1.0 | 1.114 |
| | SPONGE-$(k+1)$ | 6 | 0.792 | 1.0 | 2 | 0.667 | 0.0 | 1.0 | 0.0 | 0.2 | 1.0 | 6.754 |
| | SPONGE-$k$ | 10884 | 0.0 | 41.495 | 2 | 0.502 | 0.0 | 0.898 | 0.0 | 0.004 | 1.0 | 6.889 |
| 4 | LSPCD (AVG) | 1065 | 0.412 | 139.163 | 4 | 0.196 | 0.043 | 1.0 | 0.056 | 0.145 | 0.394 | 3.724 |
| | LSPCD (STD) | 1 | 0.001 | 0.037 | 0 | 0.0 | 0.0 | 0.0 | 0.001 | 0.0 | 0.0 | 0.392 |
| | SCG-MA | 1713 | 0.679 | 94.544 | 4 | 0.124 | 0.048 | 1.0 | -0.693 | 0.081 | 0.512 | 6.809 |
| | SCG-MO | 1658 | 0.698 | 82.139 | 4 | 0.142 | 0.054 | 1.0 | -0.767 | 0.084 | 0.502 | 3.863 |
| | SCG-B | 1142 | 0.0 | 116.233 | 1 | 0.102 | 0.0 | 1.0 | 0.0 | 0.102 | 0.398 | 60.02 |
| | SCG-R | 1514 | 0.02 | 118.706 | 4 | 0.174 | 0.019 | 1.0 | 0.432 | 0.08 | 0.479 | 2.545 |
| | KOCG-TOP-1 | 53 | 0.648 | 14.956 | 4 | 0.85 | 0.297 | 1.0 | -0.363 | 0.615 | 0.024 | — |
| | KOCG-TOP-$r$ | 1713 | 0.87 | 3.711 | 4 | 0.065 | 0.003 | 0.885 | -0.862 | 0.052 | 0.363 | — |
| | BNC-$(k+1)$ | 8 | 1.0 | -1.0 | 4 | 0.0 | 0.0 | -1.0 | 0.0 | 0.143 | 0.25 | 1.125 |
| | BNC-$k$ | 10884 | 0.001 | 41.495 | 4 | 0.001 | 0.0 | 0.898 | -0.429 | 0.004 | 1.0 | 1.129 |
| | SPONGE-$(k+1)$ | 18 | 0.792 | 1.0 | 4 | 0.452 | 0.0 | 1.0 | 0.0 | 0.059 | 1.0 | 5.042 |
| | SPONGE-$k$ | 10884 | 0.002 | 41.495 | 4 | 0.156 | 0.0 | 0.898 | 0.0 | 0.004 | 1.0 | 6.327 |
| 6 | LSPCD (AVG) | 1021 | 0.329 | 137.627 | 6 | 0.176 | 0.028 | 1.0 | 0.04 | 0.15 | 0.379 | 5.461 |
| | LSPCD (STD) | 1 | 0.001 | 0.131 | 0 | 0.001 | 0.0 | 0.0 | 0.001 | 0.0 | 0.0 | 2.813 |
| | SCG-MA | 1945 | 0.624 | 84.933 | 5 | 0.107 | 0.033 | 1.0 | -0.771 | 0.069 | 0.56 | 8.225 |
| | SCG-MO | 2469 | 0.723 | 55.571 | 5 | 0.16 | 0.003 | 1.0 | -0.853 | 0.049 | 0.629 | 6.925 |
| | SCG-B | 1142 | 0.0 | 116.233 | 1 | 0.102 | 0.0 | 1.0 | 0.0 | 0.102 | 0.398 | 98.669 |
| | SCG-R | 1660 | 0.359 | 50.258 | 6 | 0.08 | 0.038 | 0.986 | -0.756 | 0.052 | 0.356 | 5.075 |
| | KOCG-TOP-1 | 81 | 0.929 | 8.622 | 6 | 0.923 | 0.088 | 1.0 | -0.673 | 0.536 | 0.032 | — |
| | KOCG-TOP-$r$ | 1945 | 0.974 | 4.037 | 6 | 0.061 | 0.003 | 0.917 | -0.876 | 0.053 | 0.442 | — |
| | BNC-$(k+1)$ | 12 | 0.938 | -0.833 | 6 | 0.222 | 0.0 | -0.714 | 0.0 | 0.106 | 0.25 | 1.923 |
| | BNC-$k$ | 10884 | 0.001 | 41.495 | 6 | 0.056 | 0.0 | 0.898 | -0.2 | 0.004 | 1.0 | 1.155 |
| | SPONGE-$(k+1)$ | 18 | 0.92 | 1.0 | 6 | 0.667 | 0.0 | 1.0 | 0.0 | 0.059 | 1.0 | 11.664 |
| | SPONGE-$k$ | 10884 | 0.001 | 41.495 | 6 | 0.501 | 0.0 | 0.898 | 0.0 | 0.004 | 1.0 | 9.346 |

Table 8: Detailed results for the **Slashdot** dataset. LSPCD (avg) and LSPCD (std) respectively indicate the mean and standard deviation across five runs of our method with different seeds.

| $k$ | | SIZE | IF | POL | K | MAC | MAO | CC+ | CC- | DENS | ISO | TIME (s) |
|---|---|---|---|---|---|---|---|---|---|---|---|---|
| 2 | LSPCD (AVG) | 235 | 0.251 | 75.903 | 2 | 0.207 | 0.055 | 0.969 | 0.836 | 0.337 | 0.167 | 29.957 |
| | LSPCD (STD) | 0 | 0.004 | 0.095 | 0 | 0.0 | 0.001 | 0.0 | 0.003 | 0.001 | 0.0 | 3.151 |
| | N2PC ($\gamma = 1$) | 205 | 0.0 | 81.239 | 1 | 0.403 | 0.0 | 0.979 | 0.0 | 0.407 | 0.165 | 156.653 |
| | N2PC ($\gamma = 1.2$) | 205 | 0.0 | 81.141 | 1 | 0.403 | 0.0 | 0.975 | 0.0 | 0.408 | 0.167 | 154.593 |
| | N2PC ($\gamma = 1.5$) | 191 | 0.0 | 81.77 | 1 | 0.435 | 0.0 | 0.977 | 0.0 | 0.441 | 0.175 | 161.828 |
| | N2PC ($\gamma = 1.7$) | 342 | 0.996 | 55.041 | 2 | 0.297 | 0.026 | 0.969 | 0.677 | 0.171 | 0.175 | 219.089 |
| | N2PC ($\gamma = 2.0$) | 404 | 1.0 | 52.069 | 2 | 0.248 | 0.019 | 0.97 | 0.64 | 0.137 | 0.18 | 254.945 |
| | SCG-MA | 307 | 0.014 | 77.485 | 2 | 0.63 | 0.123 | 0.968 | 0.923 | 0.262 | 0.152 | 3.316 |
| | SCG-MO | 234 | 0.009 | 79.692 | 2 | 0.674 | 0.137 | 0.973 | 0.882 | 0.352 | 0.145 | 2.654 |
| | SCG-B | 289 | 0.045 | 60.962 | 2 | 0.145 | 0.056 | 0.98 | -0.005 | 0.221 | 0.205 | 287.233 |
| | SCG-R | 3033 | 0.075 | 29.706 | 2 | 0.007 | 0.007 | 0.872 | 0.635 | 0.011 | 0.216 | 25.778 |
| | KOCG-TOP-1 | 3 | 0.792 | 2.0 | 2 | 1.0 | 1.0 | 1.0 | 1.0 | 1.0 | 0.005 | — |
| | KOCG-TOP-$r$ | 307 | 0.981 | 2.612 | 2 | 0.028 | 0.03 | 0.159 | 0.182 | 0.05 | 0.037 | — |
| 4 | LSPCD (AVG) | 380 | 0.54 | 61.089 | 4 | 0.212 | 0.087 | 0.966 | 0.492 | 0.192 | 0.189 | 37.751 |
| | LSPCD (STD) | 2 | 0.005 | 0.251 | 0 | 0.01 | 0.003 | 0.0 | 0.004 | 0.002 | 0.001 | 3.013 |
| | SCG-MA | 2552 | 0.246 | 35.53 | 4 | 0.159 | 0.012 | 0.862 | -0.431 | 0.026 | 0.269 | 16.032 |
| | SCG-MO | 2111 | 0.195 | 38.534 | 4 | 0.181 | 0.051 | 0.876 | -0.657 | 0.03 | 0.24 | 21.868 |
| | SCG-B | 410 | 0.38 | 48.306 | 4 | 0.287 | 0.101 | 0.973 | -0.491 | 0.128 | 0.199 | 814.654 |
| | SCG-R | 3853 | 0.762 | 10.749 | 4 | 0.01 | 0.002 | 0.877 | -0.34 | 0.008 | 0.227 | 27.234 |
| | KOCG-TOP-1 | 23 | 0.805 | 2.609 | 4 | 0.453 | 0.172 | 1.0 | 0.643 | 0.206 | 0.009 | — |
| | KOCG-TOP-$r$ | 2552 | 0.789 | 2.973 | 4 | 0.013 | 0.003 | 0.627 | -0.477 | 0.012 | 0.16 | — |
| 6 | LSPCD (AVG) | 272 | 0.431 | 57.075 | 6 | 0.306 | 0.083 | 0.982 | 0.423 | 0.251 | 0.156 | 64.95 |
| | LSPCD (STD) | 30 | 0.088 | 2.428 | 0 | 0.041 | 0.013 | 0.001 | 0.134 | 0.032 | 0.014 | 24.494 |
| | SCG-MA | 2343 | 0.171 | 37.849 | 5 | 0.35 | 0.026 | 0.868 | -0.701 | 0.028 | 0.256 | 32.081 |
| | SCG-MO | 2504 | 0.293 | 34.649 | 6 | 0.212 | 0.063 | 0.876 | -0.421 | 0.026 | 0.265 | 26.278 |
| | SCG-B | 420 | 0.317 | 47.676 | 3 | 0.254 | 0.005 | 0.971 | -0.481 | 0.124 | 0.191 | 1408.849 |
| | SCG-R | 9661 | 0.457 | 7.906 | 6 | 0.02 | 0.002 | 0.814 | -0.43 | 0.004 | 0.433 | 73.943 |
| | KOCG-TOP-1 | 48 | 0.899 | 3.583 | 6 | 0.65 | 0.079 | 0.978 | -0.166 | 0.216 | 0.016 | — |
| | KOCG-TOP-$r$ | 2343 | 0.911 | 3.28 | 6 | 0.021 | 0.003 | 0.722 | -0.54 | 0.014 | 0.164 | — |

Table 9: Detailed results for the **WikiCon** dataset. LSPCD (avg) and LSPCD (std) respectively indicate the mean and standard deviation across five runs of our method with different seeds.

| $k$ | | SIZE | IF | POL | K | MAC | MAO | CC+ | CC- | DENS | ISO | TIME (s) |
|---|---|---|---|---|---|---|---|---|---|---|---|---|
| 2 | LSPCD (AVG) | 1876 | 0.825 | 190.8 | 2 | 0.055 | 0.128 | 0.871 | 0.997 | 0.108 | 0.242 | 72.368 |
| | LSPCD (STD) | 0 | 0.0 | 0.019 | 0 | 0.0 | 0.0 | 0.0 | 0.0 | 0.0 | 0.0 | 17.051 |
| | N2PC ($\gamma = 1$) | 2471 | 0.463 | 172.805 | 2 | 0.035 | 0.093 | 0.829 | 1.0 | 0.078 | 0.223 | 268.183 |
| | N2PC ($\gamma = 1.2$) | 2770 | 0.773 | 175.713 | 2 | 0.037 | 0.075 | 0.836 | 1.0 | 0.069 | 0.242 | 321.913 |
| | N2PC ($\gamma = 1.5$) | 2788 | 0.986 | 158.235 | 2 | 0.044 | 0.067 | 0.857 | 0.999 | 0.061 | 0.238 | 698.413 |
| | N2PC ($\gamma = 1.7$) | 2926 | 0.991 | 155.484 | 2 | 0.042 | 0.063 | 0.851 | 0.999 | 0.057 | 0.241 | 714.811 |
| | N2PC ($\gamma = 2.0$) | 2938 | 1.0 | 142.048 | 2 | 0.044 | 0.057 | 0.841 | 0.999 | 0.052 | 0.224 | 758.571 |
| | SCG-MA | 8903 | 0.53 | 155.215 | 2 | 0.008 | 0.026 | 0.81 | 0.998 | 0.019 | 0.473 | 69.196 |
| | SCG-MO | 2442 | 0.431 | 175.654 | 2 | 0.036 | 0.094 | 0.839 | 0.999 | 0.08 | 0.22 | 19.316 |
| | SCG-B | 502 | 0.638 | 129.335 | 2 | 0.117 | 0.387 | 0.816 | 0.926 | 0.295 | 0.142 | 1314.819 |
| | SCG-R | 12669 | 0.571 | 101.138 | 2 | 0.004 | 0.011 | 0.798 | 0.997 | 0.009 | 0.441 | 169.997 |
| | KOCG-TOP-1 | 14 | 0.842 | 5.857 | 2 | 0.803 | 0.289 | 0.85 | 0.368 | 0.648 | 0.002 | — |
| | KOCG-TOP-$r$ | 8903 | 0.986 | 3.417 | 2 | 0.007 | 0.007 | 0.0 | 0.056 | 0.014 | 0.327 | — |
| 4 | LSPCD (AVG) | 2288 | 0.556 | 113.637 | 4 | 0.033 | 0.051 | 0.869 | 0.936 | 0.086 | 0.23 | 88.288 |
| | LSPCD (STD) | 40 | 0.034 | 2.613 | 0 | 0.008 | 0.005 | 0.002 | 0.058 | 0.001 | 0.009 | 20.523 |
| | SCG-MA | 4852 | 0.058 | 104.937 | 4 | 0.042 | 0.104 | 0.82 | 0.577 | 0.027 | 0.241 | 139.274 |
| | SCG-MO | 1943 | 0.238 | 117.935 | 4 | 0.063 | 0.117 | 0.848 | 0.533 | 0.086 | 0.163 | 69.637 |
| | SCG-B | 1700 | 0.856 | 49.824 | 4 | 0.12 | 0.032 | 0.768 | 0.268 | 0.07 | 0.174 | 3792.395 |
| | SCG-R | 7174 | 0.655 | 41.125 | 4 | 0.006 | 0.012 | 0.836 | 0.308 | 0.015 | 0.293 | 131.226 |
| | KOCG-TOP-1 | 57 | 0.231 | 4.456 | 4 | 0.708 | 0.502 | 0.75 | 0.891 | 0.253 | 0.027 | — |
| | KOCG-TOP-$r$ | 4852 | 0.987 | 3.821 | 4 | 0.014 | 0.011 | 0.213 | -0.075 | 0.024 | 0.199 | — |
| 6 | LSPCD (AVG) | 2394 | 0.527 | 96.085 | 6 | 0.049 | 0.039 | 0.873 | 0.847 | 0.08 | 0.233 | 69.593 |
| | LSPCD (STD) | 207 | 0.091 | 4.787 | 0 | 0.025 | 0.007 | 0.004 | 0.112 | 0.006 | 0.019 | 9.767 |
| | SCG-MA | 4827 | 0.071 | 102.611 | 6 | 0.009 | 0.044 | 0.821 | 0.622 | 0.028 | 0.243 | 145.295 |
| | SCG-MO | 2016 | 0.215 | 111.578 | 6 | 0.06 | 0.06 | 0.848 | 0.685 | 0.079 | 0.159 | 76.205 |
| | SCG-B | 1924 | 0.709 | 46.069 | 6 | 0.084 | 0.015 | 0.771 | 0.291 | 0.061 | 0.174 | 6125.694 |
| | SCG-R | 12909 | 0.739 | 18.278 | 6 | 0.011 | 0.004 | 0.788 | 0.135 | 0.009 | 0.463 | 175.294 |
| | KOCG-TOP-1 | 50 | 0.533 | 4.904 | 6 | 0.765 | 0.476 | 0.962 | 0.633 | 0.505 | 0.007 | — |
| | KOCG-TOP-$r$ | 4827 | 0.991 | 1.522 | 6 | 0.016 | 0.009 | 0.286 | -0.209 | 0.023 | 0.2 | — |

Table 10: Detailed results for the **Epinions** dataset. LSPCD (avg) and LSPCD (std) respectively indicate the mean and standard deviation across five runs of our method with different seeds.

| $k$ | | SIZE | IF | POL | K | MAC | MAO | CC+ | CC- | DENS | ISO | TIME (s) |
|---|---|---|---|---|---|---|---|---|---|---|---|---|
| 2 | LSPCD (AVG) | 2188 | 0.73 | 127.784 | 2 | 0.12 | 0.013 | 0.907 | 0.74 | 0.066 | 0.351 | 59.119 |
| | LSPCD (STD) | 4 | 0.004 | 0.181 | 0 | 0.001 | 0.0 | 0.002 | 0.002 | 0.0 | 0.0 | 3.692 |
| | N2PC ($\gamma = 1$) | 274 | 0.0 | 169.701 | 1 | 0.622 | 0.0 | 0.999 | 0.0 | 0.622 | 0.595 | 286.264 |
| | N2PC ($\gamma = 1.2$) | 265 | 0.0 | 169.834 | 1 | 0.644 | 0.0 | 0.999 | 0.0 | 0.644 | 0.578 | 252.594 |
| | N2PC ($\gamma = 1.5$) | 273 | 0.0 | 169.853 | 1 | 0.625 | 0.0 | 0.999 | 0.0 | 0.625 | 0.595 | 238.315 |
| | N2PC ($\gamma = 1.7$) | 1038 | 0.29 | 124.285 | 2 | 0.079 | 0.033 | 0.943 | 0.96 | 0.127 | 0.238 | 354.089 |
| | N2PC ($\gamma = 2.0$) | 2386 | 0.993 | 76.66 | 2 | 0.053 | 0.008 | 0.916 | 0.953 | 0.035 | 0.281 | 407.709 |
| | SCG-MA | 1234 | 0.041 | 128.316 | 2 | 0.088 | 0.114 | 0.906 | 0.739 | 0.116 | 0.246 | 34.752 |
| | SCG-MO | 1017 | 0.039 | 128.722 | 2 | 0.099 | 0.138 | 0.91 | 0.713 | 0.14 | 0.22 | 25.471 |
| | SCG-B | 253 | 0.043 | 156.379 | 2 | 0.419 | 0.205 | 0.999 | 1.0 | 0.621 | 0.501 | 822.236 |
| | SCG-R | 4396 | 0.187 | 72.282 | 2 | 0.01 | 0.007 | 0.891 | 0.766 | 0.019 | 0.363 | 12.119 |
| | KOCG-TOP-1 | 12 | 0.596 | 8.167 | 2 | 0.708 | 0.815 | 1.0 | 0.833 | 0.803 | 0.007 | — |
| | KOCG-TOP-$r$ | 1234 | 0.944 | 14.036 | 2 | 0.054 | 0.022 | 0.5 | -0.245 | 0.064 | 0.16 | — |
| 4 | LSPCD (AVG) | 2120 | 0.582 | 111.544 | 4 | 0.124 | 0.016 | 0.932 | 0.408 | 0.065 | 0.341 | 65.489 |
| | LSPCD (STD) | 129 | 0.124 | 7.5 | 0 | 0.021 | 0.003 | 0.002 | 0.312 | 0.004 | 0.006 | 2.958 |
| | SCG-MA | 1576 | 0.297 | 127.432 | 3 | 0.416 | 0.001 | 0.928 | -0.714 | 0.09 | 0.285 | 42.784 |
| | SCG-MO | 1373 | 0.337 | 128.951 | 3 | 0.438 | 0.001 | 0.934 | -0.635 | 0.103 | 0.264 | 34.407 |
| | SCG-B | 868 | 0.544 | 94.43 | 3 | 0.405 | 0.0 | 0.926 | -0.411 | 0.119 | 0.226 | 2169.558 |
| | SCG-R | 1872 | 0.201 | 65.124 | 4 | 0.152 | 0.033 | 0.928 | -0.801 | 0.044 | 0.23 | 49.068 |
| | KOCG-TOP-1 | 28 | 0.912 | 8.905 | 4 | 0.865 | 0.62 | 0.953 | 0.582 | 0.81 | 0.011 | — |
| | KOCG-TOP-$r$ | 1576 | 0.956 | 11.001 | 4 | 0.071 | 0.01 | 0.768 | -0.63 | 0.06 | 0.202 | — |
| 6 | LSPCD (AVG) | 2660 | 0.473 | 103.375 | 6 | 0.088 | 0.014 | 0.929 | 0.324 | 0.05 | 0.373 | 107.579 |
| | LSPCD (STD) | 153 | 0.051 | 3.637 | 0 | 0.009 | 0.004 | 0.002 | 0.142 | 0.004 | 0.007 | 20.806 |
| | SCG-MA | 2564 | 0.52 | 88.759 | 6 | 0.301 | 0.048 | 0.935 | -0.713 | 0.05 | 0.34 | 57.185 |
| | SCG-MO | 1373 | 0.261 | 129.22 | 3 | 0.438 | 0.001 | 0.934 | -0.635 | 0.103 | 0.264 | 37.194 |
| | SCG-B | 868 | 0.421 | 94.476 | 3 | 0.405 | 0.0 | 0.926 | -0.411 | 0.119 | 0.226 | 3696.279 |
| | SCG-R | 1365 | 0.303 | 43.324 | 6 | 0.128 | 0.036 | 0.946 | -0.898 | 0.054 | 0.203 | 53.993 |
| | KOCG-TOP-1 | 34 | 0.941 | 5.965 | 6 | 0.9 | 0.43 | 1.0 | 0.496 | 0.576 | 0.012 | — |
| | KOCG-TOP-$r$ | 2564 | 0.892 | 6.802 | 6 | 0.043 | 0.006 | 0.779 | -0.654 | 0.035 | 0.262 | — |

Table 11: Detailed results for the **WikiPol** dataset. LSPCD (avg) and LSPCD (std) respectively indicate the mean and standard deviation across five runs of our method with different seeds.

| $k$ | | SIZE | IF | POL | K | MAC | MAO | CC+ | CC- | DENS | ISO | TIME (s) |
|---|---|---|---|---|---|---|---|---|---|---|---|---|
| 2 | LSPCD (AVG) | 599 | 0.3 | 81.985 | 2 | 0.093 | 0.034 | 0.917 | 0.87 | 0.15 | 0.109 | 57.022 |
| | LSPCD (STD) | 2 | 0.001 | 0.037 | 0 | 0.001 | 0.0 | 0.003 | 0.011 | 0.0 | 0.0 | 7.851 |
| | N2PC ($\gamma = 1$) | 472 | 0.0 | 87.547 | 1 | 0.193 | 0.0 | 0.932 | 0.0 | 0.199 | 0.101 | 171.986 |
| | N2PC ($\gamma = 1.2$) | 559 | 0.0 | 87.148 | 1 | 0.162 | 0.0 | 0.933 | 0.0 | 0.167 | 0.112 | 176.185 |
| | N2PC ($\gamma = 1.5$) | 494 | 0.022 | 86.579 | 2 | 0.092 | 0.05 | 0.932 | 0.984 | 0.188 | 0.102 | 162.781 |
| | N2PC ($\gamma = 1.7$) | 562 | 0.39 | 75.167 | 2 | 0.098 | 0.023 | 0.918 | 0.994 | 0.145 | 0.099 | 154.508 |
| | N2PC ($\gamma = 2.0$) | 1243 | 0.964 | 48.269 | 2 | 0.06 | 0.004 | 0.912 | 0.989 | 0.042 | 0.127 | 359.108 |
| | SCG-MA | 1251 | 0.014 | 82.822 | 2 | 0.035 | 0.054 | 0.924 | 0.928 | 0.072 | 0.172 | 11.484 |
| | SCG-MO | 648 | 0.007 | 88.441 | 2 | 0.071 | 0.079 | 0.928 | 1.0 | 0.147 | 0.121 | 4.041 |
| | SCG-B | 609 | 0.039 | 46.525 | 2 | 0.041 | 0.013 | 0.963 | -0.238 | 0.081 | 0.112 | 773.37 |
| | SCG-R | 7400 | 0.17 | 36.119 | 2 | 0.003 | 0.001 | 0.91 | 0.63 | 0.005 | 0.305 | 76.435 |
| | KOCG-TOP-1 | 6 | 0.792 | 3.0 | 2 | 0.75 | 0.625 | 1.0 | 1.0 | 0.6 | 0.003 | — |
| | KOCG-TOP-$r$ | 1251 | 0.988 | 1.258 | 2 | 0.024 | 0.012 | 0.322 | -0.284 | 0.035 | 0.097 | — |
| 4 | LSPCD (AVG) | 450 | 0.27 | 71.628 | 4 | 0.147 | 0.068 | 0.938 | 0.546 | 0.184 | 0.086 | 83.214 |
| | LSPCD (STD) | 23 | 0.053 | 2.015 | 0 | 0.063 | 0.032 | 0.002 | 0.249 | 0.014 | 0.003 | 26.749 |
| | SCG-MA | 2140 | 0.517 | 56.471 | 4 | 0.093 | 0.014 | 0.917 | -0.613 | 0.038 | 0.217 | 49.769 |
| | SCG-MO | 2783 | 0.305 | 39.698 | 3 | 0.283 | 0.001 | 0.895 | -0.775 | 0.026 | 0.242 | 44.39 |
| | SCG-B | 727 | 0.208 | 45.661 | 2 | 0.203 | 0.001 | 0.967 | -0.899 | 0.07 | 0.125 | 2225.353 |
| | SCG-R | 7740 | 0.144 | 33.723 | 4 | 0.002 | 0.001 | 0.916 | 0.599 | 0.005 | 0.302 | 91.037 |
| | KOCG-TOP-1 | 26 | 0.707 | 3.051 | 4 | 0.558 | 0.119 | 0.949 | 0.182 | 0.255 | 0.005 | — |
| | KOCG-TOP-$r$ | 2140 | 0.844 | 4.409 | 4 | 0.02 | 0.002 | 0.808 | -0.609 | 0.01 | 0.092 | — |
| 6 | LSPCD (AVG) | 825 | 0.536 | 58.694 | 6 | 0.191 | 0.026 | 0.94 | -0.28 | 0.093 | 0.123 | 228.675 |
| | LSPCD (STD) | 72 | 0.07 | 1.548 | 0 | 0.023 | 0.014 | 0.005 | 0.183 | 0.011 | 0.008 | 192.894 |
| | SCG-MA | 2176 | 0.415 | 57.546 | 4 | 0.259 | 0.001 | 0.919 | -0.73 | 0.037 | 0.22 | 54.104 |
| | SCG-MO | 2783 | 0.236 | 41.846 | 3 | 0.283 | 0.001 | 0.895 | -0.775 | 0.026 | 0.242 | 48.571 |
| | SCG-B | 727 | 0.161 | 45.986 | 2 | 0.203 | 0.001 | 0.967 | -0.899 | 0.07 | 0.125 | 3973.384 |
| | SCG-R | 95033 | 0.423 | 3.329 | 6 | 0.006 | 0.0 | 0.884 | -0.686 | 0.003 | 0.901 | 331.066 |
| | KOCG-TOP-1 | 83 | 0.856 | 10.135 | 6 | 0.756 | 0.029 | 0.967 | -0.658 | 0.268 | 0.015 | — |
| | KOCG-TOP-$r$ | 2176 | 0.773 | 3.585 | 6 | 0.032 | 0.002 | 0.867 | -0.578 | 0.006 | 0.077 | — |

