# OpenReview forum: "An Efficient Local Search Approach for Polarized Community Discovery in Signed Networks"
_NeurIPS.cc/2025/Conference — NeurIPS 2025 poster_

### Official Review · Reviewer_W8gu · 2025-06-26

**Clarity:** 4
**Significance:** 4
**Originality:** 3
**Rating:** 5
**Confidence:** 4

**Summary:**

The paper presents an efficient local search method for detecting polarized communities in signed networks. The authors propose a novel formulation of the Polarized Community Discovery (PCD) problem, addressing issues of community balance. The approach leverages block-coordinate Frank-Wolfe optimization and translates it into a local search procedure with provable convergence guarantees. Through extensive experiments on real-world datasets, the proposed method demonstrates superior solution quality and computational efficiency compared to state-of-the-art baselines.

**Questions:**

Q1) Could the authors clarify why a comparison with N2PC (equipped with \gamma-polarity) on synthetic networks with k = 2 and unbalanced ground-truth communities was omitted, given that both the proposed method and N2PC are specifically designed to account for community balance?

**Ethical Concerns:**

["NO or VERY MINOR ethics concerns only"]

**Final Justification:**

I was already positive about the paper before the author–reviewer interaction, as I found the contribution relevant, well-motivated, and clearly presented. The authors’ rebuttal effectively clarified the points I had raised in my review, and I found their explanations satisfactory. In addition, reading the authors’ responses to the other reviewers further reinforced my positive assessment. As no significant issues remain unresolved from my perspective, I remain convinced that the paper meets the quality standards for acceptance, given its sound methodology, clear presentation, and valuable contribution to the topic.

**Limitations:**

Yes

**Paper Formatting Concerns:**

No formatting issues.

**Quality:**

3

**Strengths And Weaknesses:**

Strengths:

S1) The paper is overall well-written

S2) The proposed approach based on Frank-Wolfe optimization appears to be novel, to the best of my knowledge. To support the claim that “in Section 4, we demonstrate across a range of real-world and synthetic datasets that the advantages of local search optimization (e.g., for CC) carry over to PCD as well,” the authors could strengthen the connection to prior work by referencing related studies on correlation clustering (CC) where local search procedures have been successfully employed [n1, n2].

[n1] An Efficient Local Search Algorithm for Correlation Clustering on Large Graphs. COCOA 2023.

[n2] In and out: Optimizing overall interaction in probabilistic graphs under clustering constraints. KDD 2020.

S3) The paper offers several valuable contributions across multiple aspects: theoretical (including a novel problem formulation and theoretical results), algorithmic (proposing algorithms to solve the introduced problem), and experimental (presenting extensive results in both the main paper and the appendix).

S4) Convergence analysis of the proposed local search algorithm.

S5) Experiments on both real-world and synthetic datasets. The selected datasets are consistent with those found in the relevant literature on PCD in signed graphs.

S6) The results confirm the significance of the proposed method, as it yields solutions with high polarity while achieving better size balance.

Weaknesses:

W1) The paper fails to compare with N2PC (equipped with \gamma-polarity) on synthetic networks with k = 2 and unbalanced ground-truth communities. Since the proposed method and N2PC are the only methods that account for community balance, such comparison seems needed. Also, I think Table 1 should report for k = 2 the results of N2PC.

Overall, I believe this is a solid piece of work in which the strengths outweigh the weaknesses, and for this reason, I recommend acceptance.

---

> ### Author Rebuttal · Authors · 2025-07-31
>
> We thank the reviewer for the thoughtful feedback/comments.
>
> > ***Q1/W1: Could the authors clarify why a comparison with N2PC (equipped with \gamma-polarity) on synthetic networks with k = 2 and unbalanced ground-truth communities was omitted, given that both the proposed method and N2PC are specifically designed to account for community balance?***
>
> The reviewer is referring to Figure 2 in Appendix G.4. This case was omitted arbitrarily, as we found that the results for $k = 4$, $k = 6$, and $k = 10$ already illustrated our point effectively. The purpose of this experiment was to demonstrate that our method can recover the ground-truth communities even when they are imbalanced in size. This highlights that our proposed objective does not impose strict balance constraints and is capable of recovering communities that differ in size.
>
> Nevertheless, we will include the results for $k = 2$ in the camera-ready version, along with the N2PC baseline, which is only applicable when $k = 2$. The results are consistent with those for $k = 4$, $k = 6$, and $k = 10$, with our method slightly outperforming N2PC.
>
> Note that we compare with N2PC in detail in the main paper, see e.g. Figure 1 and Table 2.
>
> > ***Also, I think Table 1 should report for k = 2 the results of N2PC.***
>
> The results for $k = 2$ using N2PC are presented in Table 2. We included them as a separate table to emphasize the effect of varying $\gamma$ for N2PC, and how it compares to our method. However, we can easily merge these results with Table 1 (e.g., by selecting a single representative value of $\gamma$). We will add this to the camera-ready version.
>
>
> > ***To support the claim that “in Section 4, we demonstrate across a range of real-world and synthetic datasets that the advantages of local search optimization (e.g., for CC) carry over to PCD as well,” the authors could strengthen the connection to prior work by referencing related studies on correlation clustering (CC) where local search procedures have been successfully employed [n1, n2].***
>
> Thank you for pointing this out. We will add these references as well to strengthen this claim.

---

> > ### Comment · Reviewer_W8gu · 2025-08-02
> >
> > Thank you for the detailed and thoughtful clarifications. I appreciate the authors’ explanation regarding the omission of the $k=2$ unbalanced community case in Figure 2 of Appendix G.4. While I understand that the chosen examples (with $k = 4, 6$, etc.) were intended to illustrate broader behavior and were not designed for direct comparison with N2PC—which is limited to $k=2$—I believe including the $k=2$ unbalanced case would still be valuable. As both methods explicitly address balance sensitivity, this scenario is particularly relevant. The authors mention that they have conducted such experiments (vs N2PC with unbalanced ground-truth communities) and found their method to slightly outperform N2PC, yet these results were not included in the response. While I trust their method performs as claimed, I would strongly encourage including these specific results in the camera-ready version.
> >
> > Regarding Table 1 and Table 2, I now understand the motivation for presenting N2PC separately to highlight the impact of varying $\gamma$. Nevertheless, incorporating a representative N2PC result into Table 1 (e.g., $\gamma = 1$ or $1.2$)—as suggested—would improve clarity and facilitate direct comparison at a glance.
> >
> > Thank you again for your comprehensive response. After reviewing the other reviewers’ comments and the authors’ replies, I find my evaluation confirmed and will maintain my score.

---

> ### Author Response · Authors · 2025-08-08
> **Response to W8gu**
>
> We thank the reviewer for their constructive feedback and valuable suggestions, which we will incorporate into the camera-ready version. We also kindly note that the "Mandatory Acknowledgement" has not yet been submitted, and we would appreciate it if the reviewer could submit it, if required.

---

> ### Author Response · Authors · 2025-08-09
> **Response to Reviewer W8gu**
>
> Below are the results for the special case $k=2$ as we vary imbalance, compared against N2PC, as requested by the reviewer (corresponding to Figure 2 of Appendix G.4). In this setting, all other baselines perform worse. Details of the experimental setup can be found in Figure 2 and Appendix G.4. We fix the number of nodes to $n = 1000$, the noise level to $0.4$, and the size of the non-neutral clusters to $100$.
>
> | Imbalance ρ  | LSPCD (ours)-F1_score  | N2PC-F1_score |
> |----|-------------|---------------|
> | 1  | 0.89        | 0.84          |
> | 2  | 0.87        | 0.83          |
> | 3  | 0.85        | 0.81          |
> | 4  | 0.82        | 0.81          |
> | 5  | 0.84        | 0.78          |
> | 6  | 0.82        | 0.78          |
> | 7  | 0.82        | 0.81          |
> | 8  | 0.80        | 0.75          |
>
> We observe that our method is consistent or slightly better than N2PC as the imbalance of cluster sizes increases.
>
> Apologies for the delay in posting these results, we were occupied with other experiments.

---

> > ### Comment · Reviewer_W8gu · 2025-08-09
> >
> > Thank you for providing these additional results. I suggest including them in the paper, clarifying the \gamma value used for N2PC. I remain positive about this paper.

---

### Official Review · Reviewer_hYDo · 2025-07-01

**Clarity:** 3
**Significance:** 3
**Originality:** 2
**Rating:** 3
**Confidence:** 4

**Summary:**

The submission studies the question of identifying polarized communities in signed graphs. In this setting, each community consists of two opposing groups. The main point of the submission is that in previous works, the identified communities can have opposing groups of highly imbalanced sizes. Thus, the submission proposes a novel objective function, which obtains more balanced communities.

For the novel objective, an optimization algorithm based on the block-coordinate Frank-Wolfe method is proposed. It is then shown that this is equivalent to a local search approach, which is proven to converge to a stationary point and results on the convergence speed are provided. Further, a more efficient variant of this algorithm is provided.

The proposed algorithm is evaluated on synthetic and on real-world data, and it is compared against state-of-the-art competitors.

**Questions:**

1. Regarding the motivation of the submission, it is not fully clear to me why one should be aiming to find communities which are all roughly of the same size. I do agree that obtaining just very large or very small clusters is not ideal, but why should they all have the same size?
2.  Continuing from the point above: In the experiments when defining the imbalance factor, why is the parameter $\xi$ set to 3? It appears to be somewhat arbitrary, and I wonder whether this has been done in a principled way?

**Ethical Concerns:**

["NO or VERY MINOR ethics concerns only"]

**Final Justification:**

My impression is that this is a borderline paper. In general, the paper is well-executed, giving theoretical guarantees and interesting experimental results.

On the other hand, (also after the rebuttal) my impression is that the level of innovation is not exceedingly high (w.r.t. the new problem, as well as the theoretical results) and parameters such as $\xi$ could be better explored.

**Limitations:**

Yes.

**Paper Formatting Concerns:**

None.

**Quality:**

3

**Strengths And Weaknesses:**

Strengths:
1. Identifying polarized communities in signed graphs is an interesting problem, especially as many existing methods can be quite fragile w.r.t. the solution quality they deliver. Thus, providing new methods in this area is interesting.
2. The submission is overall well-written and easy to follow.
3. Theoretical guarantees are presented, and the new method has strong mathematical foundations. In general, the submission provides good insights into the connections between continuous and discrete optimization.

Weaknesses:
1. Regarding the motivation of the submission, it is not fully clear to me why one should be aiming to find communities which are all roughly of the same size. I do agree that obtaining just very large or very small clusters is not ideal, but why should they all have the same size?
2.  Continuing from the point above: In the experiments when defining the imbalance factor, why is the parameter $\xi$ set to 3? It appears to be somewhat arbitrary, and I wonder whether this has been done in a principled way?
3. It is not surprising that local search methods converge to stationary points. Also, the way that the new objective function is derived (by penalizing too many points in the non-neutral clusters) is quite standard.

---

> ### Author Rebuttal · Authors · 2025-07-30
>
> Thank you for your insightful comments.
>
> > ***Regarding the motivation of the submission, it is not fully clear to me why one should be aiming to find communities which are all roughly of the same size. I do agree that obtaining just very large or very small clusters is not ideal, but why should they all have the same size?***
>
> **Important clarification:** We do **not** claim that all communities should be of equal size. The additive regularization term in our objective (Equation 3) is designed to discourage trivial solutions, such as those involving empty or near-empty clusters. However, it does **not** impose any strict balance constraint on cluster sizes.
>
> Our objective strikes a balance between three goals: maximizing intra-cluster similarity, minimizing inter-cluster similarity, and promoting reasonably balanced cluster sizes. As such, it may favor **unequal cluster sizes** when this leads to a better overall clustering solution according to the other criteria.
>
> This behavior is evident in our experimental results. In Table 1, we report both Polarity and the Imbalance Factor, where a value of 1 for the imbalance factor indicates perfectly balanced clusters (i.e., all clusters are of equal size). Notably, our method (LSPCD) **never attains an imbalance factor of 1**, demonstrating that it does not enforce strict balance, but instead allows some variation in cluster sizes.
>
> Additionally, Figure 2 in Appendix G.4 presents an experiment on synthetic data with **imbalanced ground-truth community sizes**. We observe that our method (LSPCD) successfully recovers the true communities in this setting. This further demonstrates that our objective does not enforce strict balance constraints and can effectively handle communities of varying sizes.
>
> Finally, Appendix E provides a detailed discussion of the benefits of discovering **reasonably balanced** communities, both from an optimization standpoint and in real-world applications.
>
> > ***Continuing from the point above: In the experiments when defining the imbalance factor, why is the parameter set to 3? It appears to be somewhat arbitrary, and I wonder whether this has been done in a principled way?***
>
> We use the **imbalance factor** as defined in \[1]. When the parameter $\xi = 1$, the measure reduces to the Shannon entropy, as noted in the original paper. In our experiments, we set $\xi = 3$ to **more strongly penalize highly imbalanced clusterings**.
>
> To illustrate the effect of $\xi$, consider the following examples:
>
> * For a perfectly balanced clustering with $k = 2$ clusters with cluster sizes $[1000, 1000]$, the imbalance factor is:
>
>   * $\xi = 1$: 1.0
>   * $\xi = 3$: 1.0
>
> * For a more imbalanced clustering with sizes $[1000, 300]$, the imbalance factor becomes:
>
>   * $\xi = 1$: 0.779
>   * $\xi = 3$: 0.548
>
> As these values show, the imbalance factor is more **sensitive to deviations from perfect balance** when $\xi = 3$, making it easier to distinguish between balanced and imbalanced solutions. This allows us to better capture and highlight differences in clustering behavior across methods.
>
> Importantly, we use the same value of $\xi$ across all methods to ensure a fair comparison. While using $\xi = 1$ leads to the **same overall conclusions**, choosing $\xi = 3$ provides a sharper distinction between solutions with different degrees of balance. We illustrate this below by showing the imbalance factor of the solutions found by our method and our main competitor SCG on seven of the real-world datasets (as Table 1 in the main paper), for both $\xi = 1$ and $\xi = 3$.
>
> We see that our method often finds more balanced solutions (larger imbalance factor), whereas our main competitor SCG finds highly imbalanced solutions (small imbalance factor). Note that in cases where their SCG achieves a larger imbalance factor, our method generally achieves a larger polarity. Please refer to Table 1 in the paper for complete results, where the trade-off between imbalance and polarity is highlighted. **The main point of the results below is that the relative ranking of the imbalance factor between the methods is consistent for** $\xi = 1$ and $\xi = 3$. We observe that the results and conclusions remain similar between $\xi = 1$ and $\xi = 3$. We will add this ablation study to the appendix of the camera-ready version.
>
> Imbalance factors for $\xi = 1$ (higher value means more balanced solution in terms of size):
>
> | k | Method       |  BTC | WikiV |  REF |   SD | WikiC |   EP | WikiP |
> | - | ------------ | ---: | ----: | ---: | ---: | ----: | ---: | ----: |
> | 2 | LSPCD (ours) | 0.84 |  0.69 | 0.88 | 0.50 |  0.93 | 0.89 |  0.56 |
> | 2 | SCG-MA       | 0.38 |  0.04 | 0.05 | 0.06 |  0.77 | 0.13 |  0.06 |
> | 2 | SCG-MO       | 0.11 |  0.04 | 0.05 | 0.04 |  0.69 | 0.13 |  0.03 |
> | 2 | SCG-B        | 1.00 |  0.12 | 0.11 | 0.15 |  0.84 | 0.14 |  0.13 |
> | 2 | SCG-R        | 0.50 |  0.39 | 0.12 | 0.22 |  0.79 | 0.41 |  0.39 |
> | 4 | LSPCD (ours) | 0.74 |  0.71 | 0.58 | 0.74 |  0.81 | 0.78 |  0.55 |
> | 4 | SCG-MA       | 0.45 |  0.60 | 0.75 | 0.51 |  0.18 | 0.50 |  0.64 |
> | 4 | SCG-MO       | 0.45 |  0.60 | 0.75 | 0.41 |  0.50 | 0.54 |  0.46 |
> | 4 | SCG-B        | 0.47 |  0.70 | 0.00 | 0.55 |  0.95 | 0.68 |  0.34 |
> | 4 | SCG-R        | 0.73 |  0.64 | 0.08 | 0.90 |  0.79 | 0.36 |  0.36 |
> | 6 | LSPCD (ours) | 0.75 |  0.70 | 0.49 | 0.66 |  0.80 | 0.71 |  0.76 |
> | 6 | SCG-MA       | 0.60 |  0.66 | 0.71 | 0.35 |  0.22 | 0.63 |  0.53 |
> | 6 | SCG-MO       | 0.61 |  0.64 | 0.78 | 0.56 |  0.50 | 0.42 |  0.36 |
> | 6 | SCG-B        | 0.64 |  0.75 | 0.00 | 0.45 |  0.83 | 0.53 |  0.26 |
> | 6 | SCG-R        | 0.58 |  0.84 | 0.44 | 0.72 |  0.87 | 0.47 |  0.51 |
>
> Imbalance factors for $\xi = 3$, used in Table 1 of main paper (higher value means more balanced solution in terms of size):
>
> | k | Method       |  BTC | WikiV |  REF |   SD | WikiC |   EP | WikiP |
> | - | ------------ | ---: | ----: | ---: | ---: | ----: | ---: | ----: |
> | 2 | LSPCD (ours) | 0.65 |  0.43 | 0.71 | 0.25 |  0.83 | 0.73 |  0.30 |
> | 2 | SCG-MA       | 0.16 |  0.01 | 0.01 | 0.01 |  0.53 | 0.04 |  0.01 |
> | 2 | SCG-MO       | 0.03 |  0.01 | 0.01 | 0.01 |  0.43 | 0.04 |  0.01 |
> | 2 | SCG-B        | 0.99 |  0.04 | 0.03 | 0.05 |  0.64 | 0.04 |  0.04 |
> | 2 | SCG-R        | 0.25 |  0.17 | 0.04 | 0.08 |  0.57 | 0.19 |  0.17 |
> | 4 | LSPCD (ours) | 0.47 |  0.52 | 0.41 | 0.54 |  0.56 | 0.58 |  0.27 |
> | 4 | SCG-MA       | 0.22 |  0.36 | 0.68 | 0.25 |  0.06 | 0.30 |  0.52 |
> | 4 | SCG-MO       | 0.22 |  0.37 | 0.70 | 0.20 |  0.24 | 0.34 |  0.30 |
> | 4 | SCG-B        | 0.23 |  0.60 | 0.00 | 0.38 |  0.86 | 0.54 |  0.21 |
> | 4 | SCG-R        | 0.52 |  0.44 | 0.02 | 0.76 |  0.66 | 0.20 |  0.14 |
> | 6 | LSPCD (ours) | 0.49 |  0.56 | 0.33 | 0.43 |  0.53 | 0.47 |  0.54 |
> | 6 | SCG-MA       | 0.46 |  0.42 | 0.62 | 0.17 |  0.07 | 0.52 |  0.42 |
> | 6 | SCG-MO       | 0.46 |  0.41 | 0.72 | 0.29 |  0.22 | 0.26 |  0.24 |
> | 6 | SCG-B        | 0.47 |  0.61 | 0.00 | 0.32 |  0.71 | 0.42 |  0.16 |
> | 6 | SCG-R        | 0.41 |  0.79 | 0.36 | 0.46 |  0.74 | 0.30 |  0.42 |
>
> > ***It is not surprising that local search methods converge to stationary points. Also, the way that the new objective function is derived (by penalizing too many points in the non-neutral clusters) is quite standard.***
>
> Penalizing large clusters to encourage the identification of neutral objects is indeed standard practice in PCD. However, we are the first to introduce this penalty via an additive regularization term (see Equation 3). This additive formulation is a key innovation that enables the development of the **first theoretically grounded and efficient local search algorithm for PCD**—a direction that would likely be infeasible under the **ratio-based objectives** used in all prior work. Our experimental results further demonstrate that this additive objective yields strong empirical performance, outperforming several baselines (*partially because we are able to employ an efficient local search based algorithm*).
>
> Moreover, our theoretical contributions go beyond standard convergence guarantees. We prove a **linear convergence rate**, making it highly efficient in practice. In particular, as discussed in the paper, the standard Frank-Wolfe algorithm is known to achieve a convergence rate of $O(1/\sqrt{t})$ for general non-concave objectives. In contrast, due to the specific structure of our objective, we prove that our method achieves a significantly faster linear convergence rate of $O(1/t)$, despite the function being both non-concave and non-multilinear.
>
> Given the prevalence and effectiveness of local search optimization methods in many learning models, including modern deep learning, our approach could even be perceived as having a higher impact and greater practicality than alternative methods.
>
> **References**
>
> [1] Pirizadeh et al. Imbalance factor: a simple new scale for measuring inter-class imbalance extent in classification problems. Knowl. Inf. Syst., 65(10):4157–4183, 2023.

---

> > ### Comment · Reviewer_hYDo · 2025-08-02
> >
> > Thank you for your response.
> >
> > I understand the effect of $\xi$, but my question was why the particular value of $\xi=3$ was chosen and not, for instance, $\xi=2$, $\xi=2.5$ or $\xi=5$. This is not addressed by the rebuttal.
> >
> > > We are the first to introduce this penalty via an additive regularization term (see Equation 3).
> >
> > I am not fully convinced by this. For instance, Tsourakakis et al. (KDD'13, see the reference below) also used an additive penalty term of $\alpha \binom{|S|}{2}$, where $S$ is the desired community, to identify more coherent densest subgraphs. This is very similar to the penalty term used in Equation (3), so I think the current phrasing is overselling this idea a bit. (Indeed, one could even argue that Equation (3) should also sum over the terms $\binom{|S_m|}{2}$ as this is the number of edges in the corresponding cliques and thus a more natural choice than $|S_m|^2$.)
> >
> > I will keep my rating for now and I will take your replies into account during the final discussion.
> >
> > Charalampos E. Tsourakakis, Francesco Bonchi, Aristides Gionis, Francesco Gullo, Maria A. Tsiarli: Denser than the densest subgraph: extracting optimal quasi-cliques with quality guarantees. KDD 2013: 104-112

---

> ### Author Response · Authors · 2025-08-03
> **Response to reviewer hYDo**
>
> Thank you for the response.
>
> Regarding the choice of $\xi$ in the imbalance factor metric, we offer the following clarifications:
>
> **i)** Evaluating unsupervised learning methods in the absence of ground truth is inherently challenging, and there is often no universally or uniquely accepted metric or criterion. As a result, it is common practice to report multiple metrics, each reflecting a different objective or perspective. In this study, for real-world datasets, we report both polarity and imbalance factor. As shown in our extensive experiments, our method is the only one that consistently performs well w.r.t. both metrics across various datasets.
>
> **ii)** Given the difficulty of evaluation in such settings, no single fixed value of $\xi$ can be considered optimal. In the absence of prior preference, a reasonable approach is to examine multiple values of $\xi$ and analyze how different methods perform under each. We experimented with a range of values and found that the conclusions remain stable. As mentioned in our rebuttal, we chose $\xi = 3$ since it provides a sharper distinction between solutions with different degrees of balance, compared to $\xi = 1$, which would have been the natural choice otherwise as it corresponds to Shannon entropy. To further demonstrate robustness, we now also report results for $\xi = 2$ and $\xi = 4$. **Across all tested values, the relative ranking of methods remains consistent, and our conclusions are unchanged** (again when SCG is more balanced, it usually yields low polarity). Results are consistent for even larger values of $\xi$.
>
> Imbalance factors for $\xi = 2$:
>
> |k|Method|BTC|WikiV|REF|SD|WikiC|EP|WikiP|
> |-|-|---:|---:|---:|---:|---:|---:|---:|
> |2|LSPCD(ours)|0.73|0.51|0.78|0.31|0.87|0.80|0.37|
> |2|SCG-MA|0.21|0.01|0.02|0.02|0.62|0.05|0.02|
> |2|SCG-MO|0.04|0.01|0.02|0.01|0.51|0.05|0.01|
> |2|SCG-B|1.00|0.05|0.04|0.06|0.72|0.06|0.05|
> |2|SCG-R|0.31|0.22|0.05|0.10|0.65|0.24|0.22|
> |2|KOCG-top-1|1.00|0.79|0.72|0.85|0.89|0.68|0.85|
> |4|LSPCD(ours)|0.56|0.57|0.46|0.61|0.65|0.65|0.34|
> |4|SCG-MA|0.28|0.44|0.71|0.31|0.08|0.36|0.55|
> |4|SCG-MO|0.27|0.45|0.72|0.25|0.30|0.40|0.35|
> |4|SCG-B|0.29|0.63|0.00|0.43|0.90|0.60|0.25|
> |4|SCG-R|0.59|0.50|0.03|0.82|0.70|0.24|0.19|
> |4|KOCG-top-1|0.93|0.85|0.72|0.86|0.30|0.94|0.78|
> |6|LSPCD(ours)|0.58|0.60|0.37|0.50|0.62|0.55|0.61|
> |6|SCG-MA|0.50|0.50|0.66|0.21|0.09|0.55|0.45|
> |6|SCG-MO|0.50|0.49|0.74|0.37|0.28|0.31|0.27|
> |6|SCG-B|0.52|0.66|0.00|0.36|0.76|0.46|0.19|
> |6|KOCG-top-1|0.94|0.97|0.95|0.93|0.62|0.96|0.89|
>
> Imbalance factors for $\xi = 4$:
>
> |k|Method|BTC|WikiV|REF|SD|WikiC|EP|WikiP|
> |-|---|---:|---:|---:|---:|---:|---:|---:|
> |2|LSPCD(ours)|0.60|0.39|0.66|0.22|0.79|0.68|0.27|
> |2|SCG-MA|0.15|0.01|0.01|0.01|0.48|0.04|0.01|
> |2|SCG-MO|0.03|0.01|0.01|0.01|0.39|0.03|0.01|
> |2|SCG-B|0.99|0.03|0.03|0.04|0.59|0.04|0.04|
> |2|SCG-R|0.22|0.15|0.03|0.07|0.52|0.17|0.15|
> |2|KOCG-top-1|1.00|0.67|0.59|0.75|0.81|0.55|0.75|
> |4|LSPCD(ours)|0.42|0.49|0.38|0.50|0.50|0.54|0.24|
> |4|SCG-MA|0.20|0.32|0.66|0.22|0.05|0.27|0.49|
> |4|SCG-MO|0.19|0.34|0.68|0.17|0.21|0.30|0.28|
> |4|SCG-B|0.21|0.58|0.00|0.35|0.82|0.51|0.19|
> |4|SCG-R|0.48|0.40|0.02|0.72|0.62|0.18|0.13|
> |4|KOCG-top-1|0.88|0.78|0.60|0.76|0.21|0.89|0.66|
> |6|LSPCD(ours)|0.45|0.55|0.31|0.40|0.48|0.43|0.49|
> |6|SCG-MA|0.43|0.38|0.60|0.15|0.06|0.50|0.40|
> |6|SCG-MO|0.44|0.37|0.71|0.26|0.19|0.24|0.22|
> |6|SCG-B|0.45|0.58|0.00|0.29|0.67|0.39|0.14|
> |6|KOCG-top-1|0.91|0.95|0.91|0.87|0.48|0.93|0.84|
>
> About the used penalty via an additive regularization, we still believe that our contribution is the first work to **investigate additive regularization for the PCD problem**. We do not argue that it is the first work to consider additive regularization for signed graph mining in general. The task and problem formulation in Tsourakakis et al. significantly differ from the PCD problem, e.g. Tsourakakis et al. focus on extracting only the densest clique and do not account for neutral objects. Therefore, Tsourakakis et al. cannot be directly applied to the PCD task without being substantially re-designed. Also, note that we sum squared cluster sizes $|S_m|^2$ to encourage solutions with balanced sizes (the squaring is what achieves this). The effect is rather different than the work of Tsourakakis et al.
>
> Furthermore, there are several other significant aspects in our work that are not present in Tsourakakis et al.: i) The way we deal with Eq. 3 is very different. ii) The connection between local search and Block-Coordinate Frank-Wolfe Optimization, as described in Sec. 3.1 and 3.2. iii) Theoretical analysis of the local search procedure and proving a convergence rate of $O(1/t)$ which is even stronger than existing convergence rates for general non-concave and non-multilinear functions (which have a convergence rate of $O(1/\sqrt{t}))$. The local search method in Tsourakakis et al. is used only as a heuristic without any in-depth or theoretical analysis. iv) The efficient variant of the local search method presented in Alg. 3.

---

> > ### Comment · Reviewer_hYDo · 2025-08-04
> >
> > Thank you for your detailed reply. I will take your responses into account during the final discussion phase among the reviewers, but decide to keep my score for now.

---

### Official Review · Reviewer_BDJW · 2025-07-01

**Clarity:** 3
**Significance:** 2
**Originality:** 3
**Rating:** 4
**Confidence:** 4

**Summary:**

This paper addresses the Polarized Community Discovery Problem in Signed Networks, which consists of partitioning the network into k clusters with high (positive) similarity within clusters, low (negative) similarity across clusters and a possibly neutral subset of nodes. The paper proposes an alternative objective (with parameters alpha and beta) to a previously applied objective (polarity) and shows that the relaxation of the objective (originally NP-bard) can be formulated as a Block-coordinate Frank-Wolfe Optimization. The FW formulation is then shown to be equivalent to a local search algorithm that can be implemented e"ciently. A theoretical analysis of the impact of the parameters alpha and beta is also provided. The experiments compare the proposed solution against four alternatives using eight datasets, showing that the baselines achieve worse results in terms of the proposed objective while the proposed solution is competitive with respect to polarity.

I have to disclose that I have reviewed a previous submission and rebuttal of this paper earlier this year. Besides clarifications, the reviews and rebuttal focused on limitations in the evaluation of the proposed approach that have been partially addressed in this new version.

**Questions:**

Q1. How can the proposed formulation be justified based on real-world applications?

Q2. Why is this paper a good match for NeurIPS?

Q3. Is the optimization problem considered in this paper hard to approximate? If not, does the proposed algorithm provide any approximation guarantee?

**Ethical Concerns:**

["NO or VERY MINOR ethics concerns only"]

**Final Justification:**

This is an interesting paper that proposes an elegant solution for a problem that I believe have more theoretical than practical interest. The paper could be improved in some aspects (stronger empirical support for the proposed objective and theoretical guarantees for the proposed algorithm). However, I believe that its strengths overcome its weaknesses. The reviews are mostly positive and I would not argue against the acceptance of this paper.

**Limitations:**

Partially. The main paper could include a paragraph discussing these aspects.

**Paper Formatting Concerns:**

NA.

**Quality:**

3

**Strengths And Weaknesses:**

**Strengths:**

- The relationship between the FW formulation and the local search algorithm is interesting.
- The paper presents many additional experiments and analysis in the appendix.
- The proposed optimizations significantly decrease the complexity of the local search heuristic.

**Weaknesses:**
- The experimental section does not provide strong support for the proposed formulation.
- The paper addresses a problem with limited interest within the ML community.
- The proposed algorithm does not provide any approximation guarantees.

**Detailed comments:**

I find this to be an interesting paper, with some innovative ideas and a few limitations regarding its practical implications. I will provide some details on the weaknesses of the paper to help the authors to develop their rebuttal and possibly improve the paper.

*Empirical support for the proposed formulation:* The formulation proposed in the paper (Equation 3) is motivated using theoretical arguments. Instead, it would be more interesting to use real-world ground-truth data to justify the proposed formulation. More specifically, how does the proposed formulation can potentially impact real-world applications in social or communication networks? Otherwise, it is hard to argue for one formulation versus another. In formulations for sparsest and normalized cuts, the balance appears as a ratio and not an additive form. Spectral graph theory supports interesting connections between these formulations and Laplacian eigenvectors. It is not clear when and why we should move to an additive balancing factor as proposed in this paper.

*Problem:* Interest in polarization in social networks peaked about an eight years ago, together with misinformation and other topics. A lot of work has been done since but it is not clear where this line of research is moving towards. Looking at the NeurIPS topics of interest, this paper is closest to unsupervised machine learning and perhaps optimization. Other venues such as WetConf, WSDM, etc. seem to be more focused on this line of research.

*Theoretical guarantees:* While the problem studied in this paper is np-hard, it is not clear if it can be approximated with any theoretical guarantee and whether the proposed algorithm achieves such a guarantee.

---

> ### Author Rebuttal · Authors · 2025-07-29
>
> We thank the reviewer for the detailed and constructive feedback.
>
> > ***Q1. How can the proposed formulation be justified based on real-world applications?***
>
> Regarding the following statement by the reviewer: “*Instead, it would be more interesting to use real-world ground-truth data to justify the proposed formulation*.”
>
> This appears to imply that existing (signed) graph clustering objectives were derived from explicit analyses of patterns in real-world ground-truth data, rather than being motivated by good empirical clustering performance across datasets. However, it is not evident that this is the case. As the reviewer notes, many existing objectives, such as (signed) ratio cut, (signed) ratio association, and (signed) normalized cut, are connected to the spectral properties of various signed Laplacian matrices (as detailed in [1]). But it is not clear that spectral properties of a specific, arbitrarily chosen Laplacian matrix are inherently more aligned with real-world structure than alternative formulations.
>
> For example, consider the seminal work on PCD by Bonchi et al. [2]. Their formulation is initially based on correlation clustering, which seeks to maximize intra-cluster similarity and minimize inter-cluster similarity. They observe that this objective does not naturally encourage the formation of neutral objects. To solve this, they propose a modified version which normalizes the objective by cluster size to promote neutral objects (as we discuss in detail in Section 2.2). Crucially, their formulation is not derived from real-world ground-truth data, but from a theoretical adjustment motivated by desired clustering behavior. The same is true for other PCD papers that we cite.
>
> In our work, we adopt a similar high-level concept and approach, but propose to encourage balanced communities and neutral objects via an additive regularization term rather than a ratio-based adjustment. Our motivation is algorithmic: the additive formulation enables the development of the first (theoretically proven) efficient local search algorithm for PCD, **which would likely be infeasible with ratio-based objectives**. In addition, our experimental results demonstrate that this additive objective achieves strong performance compared to alternatives (*partially because we are able to employ an efficient local search based algorithm*). It is possible that our objective is connected to spectral properties of some Laplacian matrix, but we leave that analysis for future work.
>
> > ***Q2. Why is this paper a good match for NeurIPS?***
>
> Polarized Community Discovery (PCD) is a fundamental problem with numerous applications, particularly in the analysis of social networks, as discussed in Section 1 (see also the survey in [10]). Notably, two of our strongest competitors have been published in top-tier machine learning venues such as *Machine Learning Journal* (MLJ) and *NeurIPS* [3, 4].
>
> Beyond PCD, the broader task of clustering in signed networks (or signed graph mining more generally) has attracted considerable attention, with many contributions appearing in leading machine learning conferences and journals, see for example [5–10].
>
> Our approach can also be extended to deep learning settings, where pairwise similarities are derived from complex neural architectures, such as large language models. This direction has been explored, for instance, in the context of correlation clustering of signed networks [5, 11].
>
> > ***Q3. Is the optimization problem considered in this paper hard to approximate? If not, does the proposed algorithm provide any approximation guarantee?***
>
> Correlation clustering, a problem closely related to Polarized Community Discovery (PCD), is known to be **APX-hard**, and therefore **hard to approximate** within arbitrary precision [13]. Given the structural similarity between the two problems, it is likely that PCD is also APX-hard. In our work, we prove that our problem (Problem 2 in the paper) is **NP-hard** by reducing from correlation clustering, and a similar reduction likely implies **APX-hardness** as well.
>
> While **constant-factor approximation algorithms** exist for correlation clustering — for example, using simple pivot-based methods \[13] — these algorithms are often **outperformed in practice** by more sophisticated approaches based on **local search**, as shown in detail for example by [12]. However, these local search methods **lack known approximation guarantees** for the general case with arbitrary number of clusters $k$, in both correlation clustering and PCD.
>
> Similarly, our local search-based approach does not come with theoretical approximation guarantees due to the algorithm's complexity. Instead, we focus on proving strong convergence guarantees and demonstrate that our method consistently outperforms standard baselines in empirical evaluations. In particular, as discussed in the paper, the standard Frank-Wolfe algorithm is known to achieve a convergence rate of $O(1/\sqrt{t})$ for general non-concave objectives. In contrast, due to the specific structure of our objective, we prove that our method achieves a significantly faster linear convergence rate of $O(1/t)$, despite the function being both non-concave and non-multilinear. Given the prevalence and effectiveness of local search optimization methods in many learning models, including modern deep learning, our approach could even be perceived as having a higher impact and greater practicality than the methods with approximation guarantees.
>
> **References**
>
> [1] Chiang et al. Scalable Clustering of Signed Networks Using Balance Normalized Cut. CIKM 2012.
>
> [2] Bonchi et al. Discovering Polarized Communities in Signed Networks. CIKM 2019.
>
> [3] Tzeng et al. Discovering conflicting groups in signed networks. NeurIPS 2020.
>
> [4] Gullo et al. Neural discovery of balance-aware polarized communities. Mach. Learn., 113(9):6611–6644, 2024
>
> [5] Silwal et al. Kwikbucks: correlation clustering with cheap-weak and expensive-strong signals. ICLR 2023.
>
> [6] Kuroki et al. Query-Efficient Correlation Clustering with Noisy Oracle. NeurIPS 2024.
>
> [7] Bressan et al. Correlation Clustering with Adaptive Similarity Queries. NeurIPS 2019.
>
> [8] Mazumdar et al. Clustering with noisy queries. NeurIPS 2017.
>
> [9] Cucuringu et al. SPONGE: A generalized eigenproblem for clustering signed networks. AISTATS 2019.
>
> [10] Tang et al. A Survey of Signed Network Mining in Social Media. ACM Comput. Surv., 49(3):42:1–42:37, 2016.
>
> [11] Tipirneni et al. Context-Aware Clustering using Large Language Models. Preprint 2024.
>
> [12] Thiel et al. A non-convex optimization approach to correlation clustering. AAAI 2019.
>
> [13] Bansal et al. Correlation clustering. Mach. Learn., 56:89–113, 2004.

---

> > ### Comment · Reviewer_BDJW · 2025-08-03
> > **Response to the rebuttal**
> >
> > Thanks to the authors for their response. The answers address my questions partially. I believe this paper has relevant some contributions and its strengths overcome some of the weaknesses I have listed. I understand that Q1 and Q3 are hard to address directly due to the lack of real labelled datasets and theory that is specific to Polarized Community Discovery, respectively. I also do agree that Correlation Clustering has attracted more attention from the ML community than the related problem studied in this paper. I believe that the current reviews (mostly positive) reflect the contributions of the paper well and would not argue against the acceptance of this paper.

---

### Official Review · Reviewer_Tg7f · 2025-07-03

**Clarity:** 3
**Significance:** 2
**Originality:** 3
**Rating:** 4
**Confidence:** 4

**Summary:**

The paper proposes a polarized community discovery method for signed graphs, aiming to address imbalanced clustering issues while maintaining good polarity of the discovered cohesive communities. Experiments on real and synthetic graphs were conducted to verify the performance of the method.

**Questions:**

Q1: The paper claims that scalability is one of the main contributions of the proposed method. However, the largest dataset used in the paper has only 2 million edges, which is not convincing to me. It would be better to include larger real datasets in the main paper to showcase the efficiency of the proposed method.

Q2: It would be beneficial to demonstrate performance by comparing with related works using more sophisticated polarized community models, e.g., k-core, k-truss, and cliques. These methods can easily be extended to support top-k community retrieval. For example:

[1] Effective and efficient core computation in signed networks, Information Sciences 2023

[2] Maximum Balanced (k,ϵ)-Bitruss Detection in Signed Bipartite Graph, VLDB 2024

[3] Clique Identification in Signed Graphs: A Balance Theory Based Model, TKDE 2023

[4] Identifying Large Structural Balanced Cliques in Signed Graphs, TKDE 2024

**Ethical Concerns:**

["NO or VERY MINOR ethics concerns only"]

**Final Justification:**

Thank you for the rebuttal. I have read all the reviews, rebuttals, and the subsequent discussion. I will keep my current score.

**Limitations:**

Yes

**Quality:**

3

**Strengths And Weaknesses:**

S1: The problem studied in the paper is well-motivated and highly practical for real applications.

S2: The paper is clearly written and easy to follow.

S3: Detailed proofs and analyses are provided in the appendices.

W1: It would be better to include performance comparisons with the most recent methods. Among all compared algorithms, only one was recently proposed (in 2024); all others were proposed before 2020.

W2: The experiments could be enhanced by including larger datasets.

W3: The related works are not comprehensively surveyed.

---

> ### Author Rebuttal · Authors · 2025-07-29
>
> Thank you for the thoughtful feedback.
>
> > ***Q1/W2:  The paper claims that scalability is one of the main contributions of the proposed method. However, the largest dataset used in the paper has only 2 million edges, which is not convincing to me. It would be better to include larger real datasets in the main paper to showcase the efficiency of the proposed method.***
>
> We use the same datasets as the most relevant prior work on polarized community discovery (PCD), which we cite (e.g., [5,6,7,8]). These datasets vary widely in size, ranging from 6K nodes with 214K edges to 116K nodes with 2M edges. Across these datasets, our method remains competitive with baselines in terms of runtime (see Appendix G.8), demonstrating that our local search approach scales comparably to the best existing methods for PCD (while outperforming them in clustering quality). Furthermore, our proposed local search algorithm (Algorithm 3) is substantially faster than naive local search (see runtime comparison in Figure 3).
>
> > ***Q2/W1: It would be beneficial to demonstrate performance by comparing with related works using more sophisticated polarized community models, e.g., k-core, k-truss, and cliques. These methods can easily be extended to support top-k community retrieval.***
>
> We aim to find $k$ polarized communities in a signed graph, allowing for the presence of **neutral** objects (which we refer to as PCD). Among existing works on PCD, only [5,7] explicitly support an arbitrary number of clusters $k$, making them the only directly relevant prior work. We also include [7] in our comparisons, as it is the state-of-the-art for the special case of $k = 2$ clusters (there are a number of works for this special case, which we cite).
>
> Following [7], we additionally evaluate two state-of-the-art signed graph clustering methods (SPONGE and BNC). However, these methods do **not** explicitly account for neutral objects. To enable a fair comparison, we adapt them using heuristic extensions to allow neutral objects (as in [7]). Nonetheless, our experiments show that these heuristics result in poor performance. This supports our conclusion that signed clustering methods which do not explicitly model neutrality are not effective in settings where neutral objects are present.
>
> The works cited by the reviewer (i.e., [1,2,3,4]) also do **not** explicitly handle neutral objects and would likely perform similarly poorly in our setting. Moreover, while some of the ideas presented in these papers can be used to detect clusters/communities, none of these works focus specifically on community or cluster detection. We now provide a detailed comparison of the four mentioned works:
>
> * [1]: The paper looks for a *single* maximal ($p$,$n$)-core, i.e., one subgraph in which every vertex has at least $p$ positive and fewer than $n$ negative intra‑edge, whereas PCD must *simultaneously* produce $k$ **disjoint communities plus a neutral set**. In our setting, the clusters/communities are detected automatically and do not need to set such possibly non-trivial hyper parameters. Since their objective is to *grow one cohesive subgraph*, not to partition the whole graph, there is no straightforward way to adapt their algorithms into a clustering baseline. It is not clear how one would extract exactly $k$ clusters, and neutral objects also are not explicitly handled.
>
> * [2]: Their algorithm looks for one (or several) dense balanced bitrusses in a **bipartite signed network**, defined by butterfly motifs and two disjoint node types, whereas PCD aims to partition a general (possibly **non‑bipartite**) signed graph into exactly $k$ communities plus a neutral set. It is not clear how one would extract exactly $k$ clusters from their method, and neutral objects are not explicitly handled. They mention **Community Search**, but this is about finding one community that contains a set of query nodes.
>
> * [3]: This work differs from ours, as it focuses on clique enumeration under the following assumptions: (i) each clique must contain at least $k$ nodes (note: this is a different $k$ compared to out paper, where $k$ denotes number of clusters), and (ii) cliques must not include any triangle with either two positive and one negative edge, or with all three edges negative. It is worth noting that triangles with two positive and one negative edge are particularly common in the presence of noisy edge weights (including all real-world and synthetic datasets in our experiments). Therefore, it is reasonable to allow them to be included in the discovered communities. Notably, our method is highly robust to such noise. Moreover, in our setting we do not impose any strict conditions on the size of the clusters.
>
> * [4]: This method also differs significantly from our setting: it aims to compute a maximal clique $C$ that includes $C_L$​ and $C_R$​, such that all negative edges lie between $C_L$​ and $C_R$​, and both $C_L$​ and $C_R$​ are larger than a predefined threshold $\tau$. However, we are looking for coherent and positively well-connected clusters as well as a neutral set.
>
> In short, none of the methods can be included as baselines for signed clustering with neutral objects without their methods being significantly re-designed.
>
> > ***W3: The related works are not comprehensively surveyed.***
>
> We have included all works directly relevant to PCD in our discussion (see Section 1). As noted, the papers cited by the reviewer ([1,2,3,4]) are not directly related to our task. They also do not cite any literature on polarized community discovery with neutral objects, nor are they cited by any such works. For these reasons, we do not consider them relevant for direct comparison. However, we will include a brief discussion in the introduction (Section 1) about the connection to the suggested papers and the scope of our study in the camera-ready version.
>
> **References**
>
> [5] Chu et al. Finding gangs in war from signed networks. SIGKDD 2016.
>
> [6] Bonchi et al. Discovering polarized communities in signed networks. CIKM 2019.
>
> [7] Tzeng et al. Discovering conflicting groups in signed networks. NeurIPS 2020.
>
> [8] Gullo et al. Neural discovery of balance-aware polarized communities. *Mach. Learn.*, 113(9):6611–6644, 2024

---

> ### Comment · Reviewer_Tg7f · 2025-08-05
>
> The author could refer to the recent papers on signed graphs to find more larger datasets to enhance the evaluation of the proposed tehcniques.

---

> ### Author Response · Authors · 2025-08-05
> **Response to reviewer Tg7f**
>
> We thank the reviewer for their comment. As noted in our rebuttal, we use the same datasets as the most relevant prior work on PCD, including eight real-world datasets with diverse sizes, characteristics, and properties, as well as four synthetic datasets generated using the m-SSBM model (12 datasets in total). For each synthetic dataset, we evaluate performance across multiple noise levels.
>
> We understand that in general including more datasets may often enhance the experimental studies. However, it is unclear whether other datasets from the signed graph mining literature, selected arbitrarily, would be suitable for evaluating PCD, as many signed networks lack well-defined community structures. Therefore, we kindly ask the reviewer to let us know if they have any specific dataset in mind.
>
> We use the same real-world datasets as our most relevant competitor, SCG [7]. Furthermore, a recent work on PCD by Gullo et al. (2024) [8], which proposes the N2PC method included in our comparisons, also relies on largely the same set of datasets. Notably, none of the datasets used by Gullo et al. are larger than those employed in our study.

---

> ### Author Response · Authors · 2025-08-08
> **Response to reviewer Tg7f**
>
> As detailed in the rebuttal, our experiments demonstrate that our proposed local search method (LSPCD) is competitive with state-of-the-art baselines in runtime, while achieving superior clustering quality, on both real-world and synthetic datasets commonly used in prior PCD studies.
>
> To further highlight the scalability of our approach, we present additional experiments on large synthetic datasets generated using the m-SSBM model described in the paper, containing up to 250K nodes and 150 million edges. Our method is capable of handling even larger datasets; the only constraint was the memory required to store the matrix of pairwise relations on the machine used for our experiments, a limitation that could be easily overcome by using machines with larger memory capacities. We compare LSPCD with the strongest baseline, SCG with min-angle rounding (SCG-MA), and report the F1 score (ground-truth cluster recovery) and runtime in seconds. Other baselines yield lower F1 scores, and most are also slower in runtime. We fix noise level to $0.45$, $k = 6$ and size of non-neutral clusters to $500$.
>
> | # Nodes | # Edges  | LSPCD(ours)-F1_Score | SCG-MA-F1_Score | LSPCD(ours)-Runtime(s) | SCG-MA-Runtime(s) |
> |--------:|----------|-------------|--------------|----------------|-----------------|
> |  50,000 | 9M       | 0.99986     | 0.83484      | 33.50          | 218.04          |
> |  75,000 | 17M      | 0.99986     | 0.83182      | 35.21          | 298.93          |
> | 100,000 | 28M      | 0.99986     | 0.87673      | 47.11          | 507.96          |
> | 125,000 | 42M      | 0.99986     | 0.83446      | 61.04          | 732.75          |
> | 150,000 | 60M      | 0.99986     | 0.83315      | 84.31          | 866.92          |
> | 200,000 | 100M     | 0.99986     | 0.83363      | 112.08         | 1437.89         |
> | 250,000 | 150M     | 0.99986     | 0.83350      | 114.77         | 2030.98         |
>
> Our method consistently outperforms SCG in terms of F1 score while also being more efficient, demonstrating its ability to scale to larger graphs. This experiment will be added to the appendix of the camera-ready version.

---

### Note · Authors · 2025-08-12

Below are our final remarks.

* **Reviewer Tg7f** requested experiments on larger datasets to further demonstrate the scalability of our local search method. Our paper already evaluates on both synthetic and real-world datasets standard in prior PCD work (see rebuttal to Reviewer Tg7f for citations), covering a range of sizes and characteristics. Results show our method is competitive with existing PCD baselines in runtime while consistently outperforming them in clustering quality. Figure 3 further demonstrates that it scales substantially better than a naive local search variant.

  To our knowledge, there are no commonly used real-world benchmark datasets for PCD **larger** than those used in our main experiments.  In the rebuttal, we reported results on larger datasets generated by the m-SSBM model, ranging from 10 million to 150 million edges, confirming that our method scales to substantially larger graphs. We will include them in the camera-ready version.

* **Reviewer BDJW** asked about approximation guarantees. To our knowledge, none are known for local search in PCD or the related correlation clustering problem, due to the difficulty of analyzing local search compared to simpler methods with known guarantees. Nevertheless, it performs strongly in practice, usually outperforming such simpler methods. Instead of approximation bounds, we prove a **linear convergence rate**, demonstrating its efficiency.

* **Reviewer hYDo** asked about the hyperparameter $\xi$ used in the imbalance-factor metric, which quantifies the degree of imbalance of a solution. We used $\xi = 3$ to sharpen distinctions between different balance levels. In the rebuttal, we verified robustness by varying $\xi \in \\{1,2,3,4\\}$; demonstrating that the key findings of the paper remain unchanged. We will add this result to the camera-ready version.

* **Reviewer W8gu** suggested adding results for $k = 2$ comparing to N2PC in Figure 2 of Appendix G.4, which studies imbalanced ground-truth communities. We included these results in our response and will add them to the camera-ready version. The reported results are for $\gamma = 1$, the default setting for the N2PC method. As expected, larger $\gamma$ values, which encourage more balanced communities for N2PC, performed worse in this setting due to the imbalance in the ground-truth communities. The reviewer also recommended merging Tables 1 and 2 by adding N2PC results for a representative $\gamma$; we will make this change too.

---

### Decision · Program_Chairs · 2025-09-17

**Decision:**

Accept (poster)

**Comment:**

This paper addresses the problem of discovering polarized communities in signed graphs. The authors propose a novel objective function with additive regularization and solve it using a block-coordinate Frank-Wolfe optimization approach. The paper is clearly written, with strong theoretical contributions and extensive experimental evaluation. Although the reviewers concerned insufficient comparisons with recent methods and limited dataset sizes in the experiments, the proposed formulation and its connection to an efficient local search algorithm via Frank-Wolfe optimization are both interesting and valuable. Therefore, I recommend acceptance of this paper.